# Personalized Adapter for Large Meteorology Model on Devices: Towards Weather Foundation Models

**Shengchao Chen♦, Guodong Long♦, Jing Jiang♦, and Chengqi Zhang♠**

♦Australian Artificial Intelligence Institute, University of Technology Sydney
♠Department of Data Science and AI, The Hong Kong Polytechnic University
shengchao.chen.uts@gmail.com, {guodong.long, jing.jiang}@uts.edu.au
chengqi.zhang@polyu.edu.hk

## Abstract

This paper demonstrates that pre-trained language models (PLMs) are strong foundation models for on-device meteorological variables modeling. We present LM-WEATHER, a generic approach to taming PLMs, that have learned massive sequential knowledge from the universe of natural language databases, to acquire an immediate capability to obtain highly customized models for heterogeneous meteorological data on devices while keeping high efficiency. Concretely, we introduce a lightweight personalized adapter into PLMs and endows it with weather pattern awareness. During communication between clients and the server, low-rank-based transmission is performed to effectively fuse the global knowledge among devices while maintaining high communication efficiency and ensuring privacy. Experiments on real-wold dataset show that LM-WEATHER outperforms the state-of-the-art results by a large margin across various tasks (*e.g.*, forecasting and imputation at different scales). We provide extensive and in-depth analyses experiments, which verify that LM-WEATHER can (1) indeed leverage sequential knowledge from natural language to accurately handle meteorological sequence, (2) allows each devices obtain highly customized models under significant heterogeneity, and (3) generalize under data-limited and out-of-distribution (OOD) scenarios. Code available on `https://github.com/shengchaochen82/LM-Weather`.

## 1 Introduction

Accurately modeling weather variation pattern from large amount of meteorological variables sequences is increasingly vital for providing efficient weather analysis support for disaster warning. Recently, the promise of learning to understand weather pattern from data via deep learning (DL) has led to an ongoing paradigm shift apart from the long-established physics-based methods [1, 2].

Mining potential patterns from meteorological sequences that collected from different regions, including forecasting and imputation, is one of the most important problems in meteorology. Significant progress has been made by several latest time series approaches [1, 3, 4]. These approaches formulate meteorological variable modeling as an end-to-end spatio-temporal learning problem. This overlooks the reality that ground weather devices distributed globally gather vast amounts of data quickly. The sheer volume of data, coupled with limited network capacity, necessitates local processing on the devices, making centralised learning challenging [5]. On-device intelligence enables edge devices to compute independently, offering a primary solution to the problem.

Federated Learning (FL) [7] is a promising on-device intelligence implementation that collaboratively train a uniform model across devices without exchanging raw data. However, the model often under-perform due to data heterogeneity among clients. Personalized FL (PFL) provides new insights for on-device intelligence that allows each device obtains customized models for providing personalized

38th Conference on Neural Information Processing Systems (NeurIPS 2024).

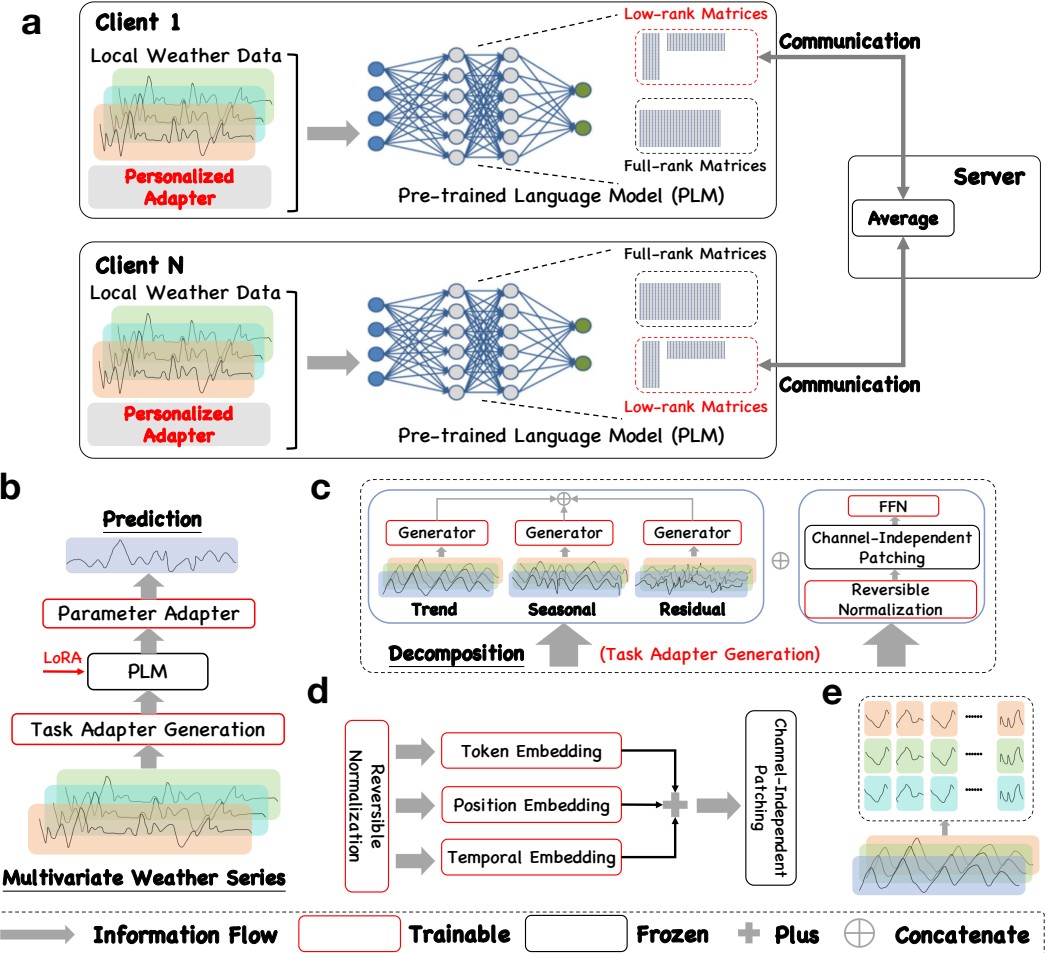

Figure 1: *Framework Overview*. **(a)** Schematic of LM-WEATHER, each client using *personalized adapter* to endow the PLM for local weather awareness, only low-rank matrices are transmitted to enhance efficiency during communication; **(b)** Brief structure of PLM on each client, detailed architecture can be found in Appendix; **(c)** Task Adapter Generation, the multivariate weather series input splits into two paths. The first path isolates the trend, seasonal, and residual elements, which each go through independent generator to produce specific adapters; **(d)** Architecture of the generator for each decomposed element; **(e)** Schematic diagram of Channel-Independent Patching [6].

insights [8, 9]. Albeit PFL methods showing revolutionized capability in this field, we argue that the current advancements are not necessarily at their best in on-device meteorological variable modeling as three major obstacles remain and hinder further progress:

(i) **Challenge of Heterogeneity.** Weather data's heterogeneity, unlike that of images or text, arises mainly from the unique characteristics of data collected by weather devices in various regions, such as tropical or arid areas. Furthermore, sensor malfunctions or extreme events can lead to collection disruptions or inconsistent missing data, which significantly increase the differences in data distribution across devices.

(ii) **Underperformed Shallow Network Structures.** The vast and varied data gathered by weather devices challenge simpler neural network models to generalize effectively. Furthermore, the frequent updates of weather data (hourly or by the minute) require neural models on devices to train and infer more often. This demand is hard to meet with deeper models that, while more performant, are also more resource-intensive.

(iii) **Resource-constrained Weather Devices.** From a computation perspective, weather devices cannot afford of training complex neural models from scratch, especially for foundation models [4]. From a communication perspective, transmitting complete model during the

aggregation phase in FL/PFL significantly increases communication overhead, which is impractical for real-time weather modeling.

Therefore, a compact foundational model (FM) is crucial for personalized on-device weather modeling. Yet, there's a gap in FMs for observational data. Models trained on large-scale simulation data struggle in practical applications because of notable differences in data formats and parameter scales [1, 4].

Inspired by the impressive progress of large language models (LLMs) in natural language processing, recent literature in time series analysis research has also demonstrated that pre-trained LMs provide excellent performance over dedicated models for time series analysis with tuning [10] or reprogramming [11]. This comprehensive and thorough sequence knowledge from language models can be effortlessly transferred across domains without large-scale parameter tuning. Thus, an exciting research question naturally arises:

> **Question:** *Since PLMs are powerful sequence modelers, can we leverage PLMs as foundation models to achieve personalized on-device meteorological variable modeling?*

In this paper, we show that pre-trained language models (PLMs) can as outstanding foundation models that tuned on each device with low cost can achieve personalized on-device weather pattern modeling. We propose LM-WEATHER, a generic approach to taming PLMs to understand heterogeneity on-device weather data. As shown in **Fig.1a**, we conduct a local tuning on an uniform PLM (e.g., GPT2), where lightweight *personalized adapters* are implanted to endow PLMs with weather pattern awareness by decomposing weather sequence to implicit knowledge (e.g., seasonal, trend). During communication between client and server, fewer parameters are shared globally while locally retained adapters are enforced to resist heterogeneity and facilitate privacy-assured fusion of global knowledge.

We highlight our contributions and findings as follows:

- We introduce LM-WEATHER, a generic approach that transforms Pre-trained Language Models as the foundation model to customized on-device meteorological variable modeling via *personalized adapter*. LM-WEATHER yields preferable meteorological variable sequences modeling, while being parameter-, communication-, and data-efficient.

- We collect and compile four real-world versatile datasets for on-device meteorological variable modeling across regions. As opposed to simulated datasets such as ERA5 [12], our datasets are all real-time observations. These datasets based on real-world practice and challenging, provide a pioneer in the field of on-device meteorological variable modeling.

- Experiments show that LM-WEATHER advances the state-of-the-art methods by a large margin across various setting while keeping **3.7%** of parameters communication. LM-WEATHER also demonstrates superior communication efficiency in the context of meteorological variable modeling, beating FL baselines tailored to reduce communication overhead.

- In particular, we find that LM-WEATHER can accurately handle structurally non-deterministic sequences (e.g., differences in time or variable dimensions across devices) thanks to the learned sequences knowledge from pre-trained LMs. We also find that LM-WEATHER can indeed be spatio-temporal sequences sensitive, thereby better modeling the weather pattern specificity of those high distribution similarity.

- We find that LM-WEATHER can work well in data-limited environments across various few-shot settings. We further evaluate zero-shot generalizability of LM-WEATHER in modeling complex weather patterns of unseen data, including different group of datasets and other devices, and observe superb performance.

We highlight that the goal of this study is not to compete but instead to complement current on-device meteorological variables modeling framework. Today's climate foundation models are typically trained from scratch, utilizing exceptionally large datasets (nearly 100TB [4, 13]) and incurring substantial computational costs [1]. We hope that LM-WEATHER offers a cost-effective alternative for modeling meteorological variables on-device, thereby enabling accurate regional weather trend analysis. In addition, the dataset we complied can be the important resource to provide exploring chances for this field, facilitating future research.

## 2 Preliminaries

### 2.1 On-device Meteorological Variable (Sequence) Modeling

The on-device meteorological variable (sequence) modeling challenge involves predicting future sequences from past observations for forecasting or predicting missing values for imputation on each device. While traditional physics-based approach this as a complex problem of solving multilevel atmospheric equations [14], recent deep learning techniques have shown significant potential in uncovering patterns for better weather prediction [4, 2].

**Problem Formulation**    On-device meteorological variable modeling can be formulated as an end-to-end sequence-to-sequence learning problem for each device without exchange raw data. Formally, a parameterized local model for $i$-th device $\mathcal{M}_\theta^i$ is tasked with predicting the weather sequence,

$$\mathcal{M}_\theta^i : \mathcal{X}_i \to \hat{\mathcal{X}}_i \tag{1}$$

where the $\mathcal{X}_i \in \mathbb{R}^{L \times C}$ and $\hat{\mathcal{X}}_i \in \mathbb{R}^{L' \times C'}$ denote the input and output sequences on $i$-device, $L$ and $L'$ is the input length and output length, $C$ and $C'$ is the number of input and output variable. Note that the $L' \to L$ when performing imputation. The local learning objective on each device is to find the model parameter $\theta$ that minimize the distance between $\hat{\mathcal{X}}_i$ and $\mathcal{X}_i$ given sufficient weather sequence data. The overall optimization objective is based on FedAvg,

$$F(\theta) := \arg\min \sum_{i=1}^{N} \frac{n_i}{n} F_i(\theta_i | \{D_i\}), \tag{2}$$

where $n_i$ and $n$ is the number of samples held by the $i$-th client and all clients[1], respectively, $F(\theta | \{D\})$ denotes the local objective function, $\{D\}$ is the local data.

### 2.2 Language Models in Time Series

Language models (LMs) trained on large-scale sequence data have shown extraordinary advances and led to a significant paradigm shift in NLP, boosting machines in understanding human languages (BERT/MLM-style) and synthesizing human-like text (GPT/CLM-style [15]). Analogies between time series and human languages have long been noted [16]. Recent advancements in time series analysis have demonstrated the effectiveness of PLMs in modeling time series [17, 11]. Although some of those have shown that PLMs can beat time series-specific models in updating a minor fraction of parameters [18]. As such, it is exciting to expect cutting-edge techniques of language modeling can tackle weather variables sequence-related problems rather than considering train climate foundation models [4, 1] from scratch that are heavy and expensive, and are trained from simulated data.

## 3 Taming PLMs for On-device Meteorological Variable (Sequence) Modeling

**Overview**    We proposed a generic framework named LM-WEATHER that encouraging PLMs to yield accurate prediction while keeping high efficiency for each device. The architecture is illustrated in **Fig. 1**. To endow PLMs with weather pattern awareness, we introduce a lightweight *personalized adapter* into PLMs (e.g., GPT2 [15]) such that the emergent ability of sequence modeling that transferred from text into weather is activated. To achieve cross-domain knowledge transfer with minimal effort while maintaining the sequence modeling capabilities of PLMs as intact as possible, we introduce lightweight operations in it enables both clients and servers to achieve a good trade-off between performance and efficiency (e.g., computation and communication).

### 3.1 Local Training on Each Device

Our LM-WEATHER refines PLMs for personalized weather sequence modeling on heterogeneous devices using a modular, plug-and-play architecture. Specifically, we introduce *personalized adapter* consists of (1) *Task Adapter* from latent weather knowledge and (2) *Parameter Adapter* that converts representation from the PLM into into weather forecasts. In addition, we employ lightweight operations in local training to boost computational efficiency.

---

[1]The words 'client' and 'device' have the same meaning in our paper.

**Task Adapter.**  To provide PLMs with richer effective information to activate their sequence modeling capabilities in the target knowledge domain, similar to text-based prompts in language to LLMs in NLP, we constructed task adapters by decomposing the input weather sequences into multimodal latent statistical information,

$$\mathcal{X}^k_{\text{Trend}} + \mathcal{X}^k_{\text{Seasonal}} + \mathcal{X}^k_{\text{Residual}} = \texttt{Decomp}(\mathcal{X}^k), \tag{3}$$

where $\mathcal{X}^k \in \mathbb{R}^{L \times 1}$ denote the $k$-th variable in weather sequence $\mathcal{X} \in \mathbb{R}^{L \times C}$, the trend component $\mathcal{X}_{\text{Trend}}$ and the seasonal component $\mathcal{X}_{\text{Seasonal}}$ captures the underlying long-term weather pattern and encapsulates the repeating short-term weather cycles, respectively. Furthermore, the residual component $\mathcal{X}_{\text{Residual}}$ represents the remainder of the sequence after the trend and seasonality have been extracted. Note that $\mathcal{X}_{\text{Trend}}$, $\mathcal{X}_{\text{Seasonal}}$, and $\mathcal{X}_{\text{Residual}}$ have the same shape as $\mathcal{X}$. This decomposition explicitly enables the identification of unusual observation and shifts in seasonal patterns or trends. The $\mathcal{X}_{\text{Trend}}$, $\mathcal{X}_{\text{Seasonal}}$, $\mathcal{X}_{\text{Residual}}$ are used to generate *Task Adapter* via an unified generator as **Fig. 1c & Fig. 1d** that consisting of Token Embedding, Position Embedding, and Temporal Embedding. Specially, we use one-dimensional convolution operation to map each each specific sample $\mathcal{X}^k$ while keeping raw shape to generate Token Adapter $\boldsymbol{P}_{\text{TO}}$. Additionally, we use a trainable lookup table to map each point's explicit position in the entire sequence, to generate Position Adapter $\boldsymbol{P}_{\text{PO}}$. Furthermore, we separately encode different time attributes such as minutes, hours, days, weeks, and months, via trainable parameters to dynamically model complex temporal shifts, to generate Temporal Adapter $\boldsymbol{P}_{\text{TE}}$. Finally, for each decomposition components, corresponding generated adapters can be obtained by aggregating Token Adapter $\boldsymbol{P}_{\text{TO}} \in \mathbb{R}^{L \times C}$, Position Adapter $\boldsymbol{P}_{\text{PO}} \in \mathbb{R}^{L \times C}$, and Temporal Adapter $\boldsymbol{P}_{\text{TE}} \in \mathbb{R}^{L \times C}$ as $\boldsymbol{P}_d = \boldsymbol{P}^d_{\text{TO}} + \boldsymbol{P}^d_{\text{PO}} + \boldsymbol{P}^d_{\text{TE}}$, where $d \in \{\text{Trend}, \text{Seasonal}, \text{Residual}\}$, this means that we can obtain $\boldsymbol{P}_{\text{Trend}}, \boldsymbol{P}_{\text{Seasonal}}, \boldsymbol{P}_{\text{Residual}}$. Details about the generator in **Appendix B.5**.

**Lightweight Operations.**  To enhance the PLMs' ability to represent complex inputs while reducing the computational burden to adapt to low-resource devices, we introduce lightweight operations, which includes channel-independent patching (CIP, **Fig. 1e**) [6] for input and efficient tuning of parameters for PLMs. Among them, CIP splits the multivariate sequence into separate univariate sequences, each processed by a single model with length $L_p$. This approach outperforms the original method of mixing channels by treating the variables as independent. It enables the model to capture channel interactions indirectly through shared weights, leading to improved performance without directly modeling the complexity of multiple data channels. The total number of inputs patches is $P = \frac{(T - L_p)}{S} + 2$, where $S$ denotes the horizontal sliding stride. Given these patches $\mathcal{X}^i_P \in \mathbb{R}^{P \times L_p}$, we use rearrange operation and a trainable FFN embed them as $\hat{\mathcal{X}}^i_P \in \mathbb{R}^{P \times d_m}$, where $d_m$ is dimensions created by the FFN. We also introduce a low-rank adaptation (LoRA) [19] inside PLMs aiming at language modeling for lightweight fine-tuning of attention layers to achieve cross-modal/-domain knowledge transfer from text sequences to weather sequences with minimal effort.

**Parameter Adapter.**  To adapt PLM outputs for downstream weather sequence modeling, we introduce *Parameter Adapter*, a simple FFN with a single linear layer positioned after the PLM. This adapter transforms the PLM's output to match the prediction horizon, formalized as follows:

$$\hat{\mathcal{X}} = \text{FFN}(\mathcal{M}_\theta(\text{Concat}[\hat{\boldsymbol{P}}_{\text{Trend}}, \hat{\boldsymbol{P}}_{\text{Seasonal}}, \hat{\boldsymbol{P}}_{\text{Residual}}, \hat{\mathcal{X}}])), \tag{4}$$

where the $\hat{\boldsymbol{P}}_{\text{Trend}}, \hat{\boldsymbol{P}}_{\text{Seasonal}}, \hat{\boldsymbol{P}}_{\text{Residual}}$, and $\hat{\mathcal{X}}$ are obtained from CIP based on $\boldsymbol{P}_{\text{Trend}}, \boldsymbol{P}_{\text{Seasonal}}, \boldsymbol{P}_{\text{Residual}}$, and $\mathcal{X}$. The key objectives are twofold: (1) to enrich the PLM's cross-modal representations by incorporating task-specific knowledge, and (2) to enhance the PLM's output accuracy while preserving its inherent knowledge through the integration of weather data for cross-domain knowledge transfer.

## 3.2   High-efficiency Communication Between Clients and Server

To avoid data silos and counteract the performance disparities caused by data heterogeneity while ensuring efficient communication, we update personalized adapters locally and share low-rank parameters globally in each round. Specifically, the local PLM $\mathcal{M}_\theta$ can be formulated as below:

$$\mathcal{M}_\theta \rightarrow \mathcal{M}_{\theta,t}(Communication) + \mathcal{M}_{\theta,f}(Locally) \tag{5}$$

where $\mathcal{M}_{\theta,t}$ denotes the trainable parameter from the low-rank matrices of *query* and *value* in attention modules, $\mathcal{M}_{\theta,f}$ is the frozen parameter (mainly the PLM backbone) and other trainable ones (primarily for the personalized adapter). During client-server communication, only $\mathcal{M}_{\theta,t}$ is transmitted and averaged using FedAvg [7]. At the start of the next training round, the updated $\mathcal{M}_{\theta,t}$ is broadcast to clients for further updates. Privacy is further protected by sending minimal parameters.

# 4 Main Theorems

**Theorem 4.1** (**Decomposition Rationality from Time Series**). *Given a weather series* $\mathcal{X} = \mathcal{X}_{Trend,t} + \mathcal{X}_{Seasonal,t} + \mathcal{X}_{Residual,t}$, $t \in [t_1, t_n]$. *Let* $\boldsymbol{E} = \{e_1, e_2, ..., e_n\}$ *denotes a set of orthogonal bases. Lets* $\boldsymbol{E}_{Seasonal} \subseteq \boldsymbol{E}$ *denote the subset of* $\boldsymbol{E}$ *on which* $\mathcal{X}_{Seasonal,t}$ *has non-zero eigenvalues and* $\boldsymbol{E}_{Trend} \subseteq \boldsymbol{E}$ *denote the subset of* $\boldsymbol{E}$ *on which* $\mathcal{X}_{Trend,t}$ *has non-zero eigenvalues. If* $\mathcal{X}_{Trend,t}$ *and* $\mathcal{X}_{Seasonal,t}$ *are not orthogonal, i.e.,* $\sum_{i=1}^{n} \mathcal{X}_{Trend,t}^i \mathcal{X}_{Seasonal,t}^i \neq 0$, *then* $\boldsymbol{E}_{Trend} \bigcap \boldsymbol{E}_{Seasonal} \neq 0$, *i.e.,* $\boldsymbol{E}$ *can not disentangle the two signals onto two disjoint set of bases.*

**Theorem 4.2** (**Exchange Low-Rank Matrices Ensures Privacy**). *Given a on-device weather modeling framework based on federated learning that gloabl optimization object is* $\boldsymbol{F}(\theta) = \sum_{n}^{i=1} p_i f(\{D_i\}; \theta)$, *where* $f(x; \theta)$ *is the loss function of* $i$-*th client,* $\{D_i\}$ *is dataset of* $i$-*th client, and* $p_i$ *and* $\theta$ *denote the data distribution weight of client* $i$ *and the model parameters, respectively. Given that the parameters* $\theta$ *of the PLM* $\mathcal{M}_\theta$ *broadcasted by the server consist of two parts: a frozen part* $\mathcal{M}_{\theta,f}$ *and a trainable part* $\mathcal{M}_{\theta,t}$, *interacting only the low-rank matrix parameters* $\mathcal{M}_{\theta,l} \subset \mathcal{M}_{\theta,t}$ *is a subset of trainable part* $\mathcal{M}_{\theta,t}$ *during each round ensures privacy.*

# 5 Experiments

In this section, we first present the real-world datasets that we have collected and compiled for on-device meteorological variable modeling, and second, we evaluate LM-WEATHER on these datasets, which involves normal scenario, a data-limited few-shot scenario, and a zero-shot scenario with no training data (OOD). Please refer to **Appendix** for more detailed information about proposed datasets and additional results of all evaluations (e.g., full results, additional findings & experiments).

## 5.1 Datasets

Despite the proliferation of reanalysis data aimed at building frameworks for global climate analysis, these datasets often struggle to model regional weather trend due to: (1) they depend on numerous simulations of atmospheric equations, introducing biases inconsistent with real observations, and (2) they face challenges in refining their scale to suit specific regional applications. Hence, we collected real observational data from various weather stations across different regions. We then organized this data into two series, each comprising two distinct datasets, to underscore the heterogeneity inherent in real-world settings. For detailed information on these datasets, please see the **Appendix B.1**.

**On-device Weather Series 1# (ODW1).** The dataset gathered from 15 ground weather stations across China, Japan, and South Korea, encompasses over 20 variables. It has been divided into two subsets: **ODW1T** has a heterogeneous time span, meaning the data collection start and end times vary by location. and **ODW1V** extends **ODW1T** by adding variability in the observed variables; while one variable remains constant at each station, the others vary.

**On-device Weather Series 2# (ODW2).** This dataset consists of data from 36 weather observation stations in the United States, Canada, and Israel, covering 5 different variables with a temporal resolution of 1 hour. Following the dataset setting of **ODW1**, the dataset was also subdivided into two different dataset, including **ODW2T** and **ODW2V**.

## 5.2 Setup

**Baseline.** Since our framework is based on a language model, we compare with DL-based SOTA time series models, including Transformer-based methods: Transformer [20], Informer [3], Reformer [21], Pyraformer [22], iTransformer [23], and PatchTST [6], and recent competitive models: GPT4TS [17], DLinear [24] and LightTS [25]. Note that our setting is FL-based, so we place them in FL and rename them FL-(*baseline*) like FL-Transformer, etc., and all aggregation methods used in above models is FedAvg [7]. In addition, we report a variants of LM-WEATHER, LM-WEATHER-AVE that based on FedAvg without personalization. Detailed information are in **Appendix B.2**.

**Basic Setup.** We focus on on-device meteorological variable forecasting and imputation tasks. For forecasting, we create scenarios for predicting a single variable (multivariate-univariate) and for predicting all variables (multivariate-multivariate). The main text only includes multivariate-to-multivariate forecasting results due to page constraints. For multivariate-to-univariate forecasts, refer

to the Appendix E. In imputation, we use sequence lengths of $\{96, 192, 336, 720\}$ and apply three different masking probabilities $\{25\%, 35\%, 50\%\}$ to represent missing data. The main manuscript shows imputation results for a 50% masking ratio. For more details on the setup, please refer to **Appendix B.3**. All our experiments are repeat five times and we report the averaged results.

## 5.3 Main Results

In this section, we evaluate LM-WEATHER and baseline methods on four on-device meteorological variable modeling datasets in general experiments to validate its effectiveness.

**Setups & Results of Forecasting Tasks.** Input length $L_b$ is fixed to 192, and we use four different prediction horizons $L_f \in \{96, 192, 336, 720\}$. Evaluation metrics include mean absolute error (MAE) and root square mean error (RMSE). The brief results is shown in **Tab. 1**, where our LM-WEATHER outperforms all baselines in most cases and significantly so to the majority of them. Particularly noteworthy is the comparison with GPT4TS that involves fine-tuning PLMs, where LM-WEATHER has an average **9.8%** improvement over FL-GPT4TS (MAE reported), and even the variant LM-WEATHER-AVE has an average **4%** improvement over FL-GPT4TS. In addition, LM-WEATHER shows significant average performance gains of **11.2%** and **19%** w.r.t. MAE relative to other SOTA such as FL-DLinear and FL-PatchTST.

Table 1: Results under on-device meteorological variable forecasting task (multivariate-to-multivariate). A lower value indicates better performance. **Bold**: the best, Underline: the second best. **Complete results can be found at Appendix E due to page limitation.**

| Method | | LM-WEATHER-AVE | | LM-WEATHER | | FL-GPT4TS | | FL-Reformer | | FL-Pyraformer | | FL-DLinear | | FL-PatchTST | | FL-iTransformer | | FL-LightTS | | FL-Transformer | | FL-Informer | |
|---|---|---|---|---|---|---|---|---|---|---|---|---|---|---|---|---|---|---|---|---|---|---|---|
| Dataset | Length | MAE | RMSE | MAE | RMSE | MAE | RMSE | MAE | RMSE | MAE | RMSE | MAE | RMSE | MAE | RMSE | MAE | RMSE | MAE | RMSE | MAE | RMSE | MAE | RMSE |
| ODW1T | 96 | 44.1 | 74.8 | **42.3** | **71.1** | 46.3 | 78.5 | 70.7 | 92.9 | 67.2 | 86.1 | 49.7 | 78.6 | 45.0 | 77.0 | 48.4 | 80.2 | 54.8 | 85.6 | 50.7 | 82.1 | 51.9 | 83.2 |
| | 192 | 46.3 | 77.5 | **44.4** | **73.6** | 48.6 | 81.3 | 75.1 | 98.3 | 70.0 | 90.9 | 52.3 | 81.8 | 47.3 | 79.8 | 51.8 | 84.3 | 59.5 | 90.6 | 52.1 | 84.0 | 52.9 | 84.6 |
| | 336 | 47.9 | 79.3 | **45.8** | **75.2** | 50.3 | 83.2 | 79.8 | 100.5 | 74.1 | 92.8 | 53.9 | 83.7 | 49.0 | 81.7 | 54.5 | 87.3 | 64.0 | 94.6 | 52.9 | 85.2 | 53.5 | 85.6 |
| | 720 | 51.8 | 83.0 | **49.2** | **78.5** | 54.4 | 87.2 | 87.1 | 102.9 | 80.5 | 95.2 | 57.2 | 87.3 | 53.3 | 85.6 | 60.1 | 93.1 | 72.4 | 102.7 | 55.4 | 87.6 | 55.3 | 87.4 |
| | Avg. | 47.5 | 78.7 | **45.4** | **74.6** | 49.9 | 82.5 | 78.2 | 98.7 | 73.0 | 91.3 | 53.3 | 82.8 | 48.6 | 81.0 | 53.7 | 63.7 | 62.7 | 93.4 | 52.8 | 84.7 | 53.4 | 85.2 |
| ODW1V | 96 | 42.7 | 69.5 | **42.3** | 69.6 | 44.0 | 71.4 | 42.9 | 67.8 | 57.7 | **67.2** | 46.4 | 73.3 | 44.3 | 69.6 | 56.8 | 76.8 | 48.0 | 75.1 | 67.0 | 89.4 | 59.0 | 80.3 |
| | 192 | 45.5 | 72.6 | **44.4** | 71.7 | 47.0 | 75.8 | 48.4 | 75.4 | 59.2 | **69.4** | 47.9 | 75.1 | 46.8 | 72.1 | 55.0 | 75.0 | 49.1 | 79.2 | 69.9 | 93.0 | 61.2 | 82.8 |
| | 336 | 47.2 | 74.3 | **46.0** | 72.4 | 48.8 | 77.7 | 51.0 | 77.0 | 63.4 | 73.3 | 49.1 | 76.9 | 48.5 | 74.8 | 62.4 | 83.7 | 50.8 | 77.9 | 71.4 | 94.8 | 63.7 | 85.8 |
| | 720 | 51.2 | 78.2 | **49.7** | **74.0** | 53.3 | 81.7 | 54.5 | 82.3 | 67.3 | 76.1 | 52.5 | 80.3 | 54.3 | 79.1 | 72.1 | 96.2 | 54.7 | 82.7 | 76.2 | 87.3 | 64.8 | 91.8 |
| | Avg. | 46.6 | 73.6 | **45.6** | 71.9 | 48.3 | 76.7 | 49.2 | 75.6 | 61.9 | **71.5** | 49.0 | 76.4 | 48.5 | 73.9 | 58.1 | 85.0 | 50.7 | 78.7 | 71.1 | 91.1 | 63.1 | 85.2 |
| ODW2T | 96 | 64.3 | 88.2 | **62.8** | 85.5 | 66.8 | 91.7 | 100.3 | 126.3 | 95.0 | 120.3 | 67.9 | 84.7 | 70.2 | 88.1 | 68.6 | 86.5 | 68.4 | 85.4 | 85.0 | 103.0 | 84.7 | 102.7 |
| | 192 | 67.7 | 91.5 | **66.2** | 89.1 | 71.1 | 96.1 | 102.1 | 130.3 | 99.9 | 125.8 | 71.4 | 88.1 | 72.2 | 90.7 | 71.1 | 88.9 | 71.9 | 88.9 | 85.0 | 103.0 | 84.9 | 102.8 |
| | 336 | 69.5 | 93.7 | **67.9** | 91.1 | 72.9 | 98.4 | 104.2 | 130.0 | 102.0 | 128.5 | 73.0 | 89.5 | 73.0 | 91.9 | 71.8 | 89.6 | 73.7 | 90.5 | 82.6 | 100.5 | 84.8 | 102.9 |
| | 720 | 72.6 | 97.3 | **70.7** | 94.6 | 76.2 | 101.2 | 107.3 | 134.2 | 104.2 | 131.4 | 76.1 | 92.9 | 75.1 | 93.3 | 72.9 | 91.0 | 76.7 | 93.7 | 84.1 | 105.1 | 85.4 | 103.8 |
| | Avg. | 68.5 | 92.7 | **66.9** | 90.1 | 71.8 | 96.9 | 103.5 | 130.2 | 100.3 | 126.5 | 72.1 | **88.8** | 72.6 | 91.0 | 71.1 | 89.0 | 72.7 | 89.6 | 84.2 | 102.9 | 84.9 | 103.1 |
| ODW2V | 96 | 76.8 | 99.7 | **65.1** | 88.4 | 78.5 | 102.7 | 89.6 | 112.7 | 89.1 | 112.5 | 74.8 | 96.8 | 76.3 | 99.9 | 73.5 | 97.7 | 92.2 | 117.7 | 77.0 | 100.1 | 77.4 | 100.4 |
| | 192 | 77.9 | 100.8 | **68.3** | 91.4 | 79.7 | 103.8 | 90.5 | 114.2 | 96.4 | 120.1 | 76.6 | 98.9 | 79.9 | 103.3 | 78.8 | 103.6 | 100.5 | 128.1 | 78.3 | 101.8 | 78.0 | 101.1 |
| | 336 | 78.5 | 101.5 | **69.9** | 93.0 | 80.3 | 104.5 | 94.2 | 119.3 | 98.4 | 122.2 | 77.6 | 100.2 | 81.8 | 105.3 | 82.1 | 107.5 | 105.5 | 134.4 | 79.4 | 103.3 | 78.7 | 102.0 |
| | 720 | 79.9 | 103.6 | **72.9** | 96.5 | 82.0 | 106.7 | 97.4 | 120.4 | 100.5 | 125.0 | 79.6 | 103.0 | 86.2 | 100.2 | 86.2 | 112.7 | 111.0 | 141.3 | 86.1 | 112.3 | 81.3 | 105.6 |
| | Avg. | 78.3 | 101.4 | **69.0** | 92.3 | 80.1 | 104.4 | 92.9 | 116.6 | 96.1 | 120.0 | 77.2 | 99.7 | 81.1 | 102.2 | 80.2 | 105.4 | 102.3 | 130.4 | 80.2 | 104.4 | 78.8 | 102.2 |
| 1st Count | | 0 | | **29** | | 0 | | 0 | | 4 | | 0 | | 0 | | 0 | | 0 | | 0 | | 0 | |

**Setups & Results of Imputation Tasks.** Our brief results are in **Tab. 2**, where LM-WEATHER consistently surpasses all baselines, outperforming FL-GPT4TS by **5.7%**. LM-WEATHER remains competitive even when compared with the SOTA, FL-PatchTST, FL-LightTS, and FL-DLinear.

Table 2: Results under on-device meteorological variable imputation task, where random masking ratio is 50%. A lower value indicates better performance. **Bold**: the best, Underline: the second best. **Complete results can be found at Appendix E due to page limitation.**

| Method | | LM-WEATHER-AVE | | LM-WEATHER | | FL-GPT4TS | | FL-Reformer | | FL-Pyraformer | | FL-DLinear | | FL-PatchTST | | FL-iTransformer | | FL-LightTS | | FL-Transformer | | FL-Informer | |
|---|---|---|---|---|---|---|---|---|---|---|---|---|---|---|---|---|---|---|---|---|---|---|---|
| Dataset | Length | MAE | RMSE | MAE | RMSE | MAE | RMSE | MAE | RMSE | MAE | RMSE | MAE | RMSE | MAE | RMSE | MAE | RMSE | MAE | RMSE | MAE | RMSE | MAE | RMSE |
| ODW1T | 96 | 22.4 | 43.5 | **21.7** | **41.8** | 23.3 | 45.2 | 63.7 | 88.4 | 62.2 | 85.9 | 29.2 | 50.8 | 28.9 | 54.6 | 22.8 | 44.5 | 24.4 | 43.7 | 58.3 | 82.8 | 70.8 | 99.6 |
| | 192 | 23.4 | 43.7 | **22.6** | **42.0** | 24.6 | 45.9 | 67.2 | 91.2 | 65.5 | 88.5 | 28.7 | 50.2 | 47.5 | 77.3 | 23.8 | 44.1 | 25.7 | 45.3 | 57.3 | 82.4 | 66.3 | 92.1 |
| | 336 | 24.1 | 44.1 | **23.2** | **42.4** | 25.3 | 46.3 | 70.4 | 93.4 | 68.5 | 90.6 | 28.3 | 49.4 | 48.6 | 77.0 | 27.2 | 47.7 | 26.9 | 46.6 | 58.4 | 83.5 | 36.9 | 55.3 |
| | 720 | 26.0 | 45.1 | **24.9** | **43.3** | 27.3 | 47.4 | 77.9 | 96.8 | 75.8 | 93.9 | 28.0 | 49.0 | 56.6 | 85.1 | 36.5 | 56.2 | 27.2 | 47.4 | 56.6 | 80.4 | 71.7 | 96.7 |
| | Avg. | 24.0 | 44.1 | **23.1** | **42.4** | 25.1 | 46.2 | 69.8 | 92.5 | 68.0 | 89.7 | 28.5 | 49.9 | 45.4 | 73.5 | 27.6 | 48.2 | 26.1 | 45.7 | 57.6 | 82.3 | 61.4 | 85.9 |
| ODW1V | 96 | 42.1 | 62.0 | 41.1 | 60.4 | 42.9 | 63.8 | 43.8 | 64.9 | 42.3 | **53.0** | 43.0 | 63.0 | 53.6 | 77.1 | 38.7 | 58.2 | 41.5 | 61.5 | 37.8 | 56.9 | 41.1 | 59.2 |
| | 192 | 43.9 | 64.5 | 42.8 | 62.8 | 45.6 | 66.9 | 45.8 | 67.6 | 44.7 | 56.2 | 49.3 | 71.2 | 57.5 | 81.5 | 49.3 | 68.9 | **41.9** | 62.0 | 44.1 | **57.4** | 48.8 | 66.8 |
| | 336 | 45.7 | 66.6 | **44.6** | 64.9 | 47.5 | 69.2 | 47.6 | 69.8 | 54.6 | 65.7 | 53.4 | 76.6 | 60.7 | 85.0 | 60.0 | 79.8 | 47.3 | 64.6 | 48.5 | 68.0 | 50.2 | 67.1 |
| | 720 | 47.5 | 68.7 | **46.3** | 66.9 | 49.4 | 71.4 | 49.6 | 72.0 | 59.2 | 73.5 | 56.8 | 80.7 | 63.3 | 87.4 | 61.6 | 80.4 | 52.5 | 72.9 | 52.7 | 70.1 | 60.3 | 77.2 |
| | Avg. | 44.8 | 65.5 | **43.7** | 63.8 | 46.4 | 67.8 | 46.7 | 68.6 | 50.2 | 62.1 | 50.6 | 72.9 | 58.8 | 82.7 | 52.4 | 71.8 | 45.8 | 65.3 | 45.8 | **63.1** | 50.1 | 67.6 |
| ODW2T | 96 | 38.0 | 56.6 | **36.9** | 54.9 | 39.1 | 58.3 | 50.3 | 70.3 | 95.4 | 120.8 | 40.8 | 60.0 | 38.4 | 58.6 | 39.1 | 58.3 | 38.8 | 57.8 | 65.5 | 86.6 | 51.7 | 72.0 |
| | 192 | 38.3 | 56.6 | **37.2** | 54.9 | 39.8 | 58.9 | 52.1 | 74.2 | 96.2 | 122.3 | 42.9 | 62.7 | 66.7 | 87.8 | 39.4 | 58.3 | 39.5 | 58.4 | 71.4 | 92.8 | 55.0 | 75.7 |
| | 336 | 43.5 | 65.5 | **42.2** | **63.5** | 44.8 | 68.1 | 56.6 | 78.9 | 97.8 | 125.5 | 46.0 | 67.7 | 68.7 | 90.1 | 44.8 | 67.5 | 47.8 | 65.3 | 66.8 | 88.8 | 51.5 | 72.8 |
| | 720 | 47.9 | 68.8 | **46.5** | **66.7** | 49.8 | 71.5 | 64.3 | 87.7 | 99.1 | 129.9 | 52.8 | 76.1 | 70.4 | 93.5 | 49.3 | 71.0 | 48.0 | 68.0 | 67.4 | 89.2 | 51.5 | 73.0 |
| | Avg. | 41.9 | 61.9 | **38.8** | 61.7 | 43.4 | 64.2 | 55.8 | 77.8 | 97.1 | 124.6 | 45.6 | 66.6 | 61.1 | 82.5 | 43.3 | 62.4 | 43.5 | 62.4 | 67.8 | 89.4 | 52.4 | 73.4 |
| ODW2V | 96 | 28.1 | 45.3 | **27.5** | **44.0** | 28.4 | 45.8 | 50.3 | 70.3 | 53.2 | 72.4 | 72.1 | 92.0 | 39.8 | 58.4 | 72.7 | 94.7 | 96.4 | 123.5 | 52.7 | 73.2 | 54.8 | 76.9 |
| | 192 | 28.6 | 45.3 | **28.0** | **44.0** | 29.2 | 46.1 | 51.0 | 71.1 | 46.1 | 65.2 | 75.7 | 95.9 | 44.9 | 63.7 | 79.1 | 102.0 | 98.6 | 125.8 | 53.9 | 74.7 | 56.2 | 78.8 |
| | 336 | 33.7 | 49.8 | **32.7** | **48.4** | 34.9 | 51.8 | 54.2 | 76.6 | 74.2 | 97.3 | 77.3 | 97.8 | 50.9 | 70.1 | 82.6 | 106.1 | 101.2 | 128.8 | 54.4 | 75.4 | 56.8 | 79.7 |
| | 720 | 37.1 | 53.1 | **36.0** | **51.5** | 39.3 | 56.3 | 59.4 | 81.7 | 82.4 | 100.9 | 77.1 | 97.3 | 59.2 | 79.3 | 83.0 | 106.0 | 98.5 | 124.3 | 55.4 | 77.5 | 56.4 | 78.6 |
| | Avg. | 31.9 | 48.4 | **31.1** | **47.0** | 33.0 | 50.0 | 53.7 | 74.9 | 64.0 | 84.0 | 75.5 | 95.8 | 48.7 | 67.9 | 79.4 | 102.2 | 98.7 | 125.6 | 54.1 | 75.2 | 56.0 | 78.5 |
| 1st Count | | 0 | | **30** | | 0 | | 0 | | 0 | | 0 | | 0 | | 0 | | 0 | | 0 | | 0 | |

## 5.4 Few-Shot Learning Experiments

PLMs have demonstrated remarkable few-shot learning capabilities [26]. In this subsection, we assess whether LM-WEATHER retains this ability in both forecasting and imputation tasks, based on FL for resource-constrained on-device weather modeling environments.

**Setups and Results of Forecasting & Imputation.** For both forecasting and imputation tasks, we evaluate the few-shot learning capability in scenarios using limited data, specifically, we use training ratios of 5% and 15% (Our full few-shot learning results (training ratio of 5% and 15%) can be found at **Appendix E.2**). The brief 5% few-shot learning results on forecasting and imputation tasks are depicted in **Tab. 3** and **Tab. 4**, respectively. LM-WEATHER remarkably excels over all baseline methods, and we attribute this to the successful cross-domain knowledge activation in our local dual fine-tuning for the PLM. In addition, our LM-WEATHER's communication mechanism also reduces the impact of data heterogeneity on performance, which is reflected in the fact that LM-WEATHER has an average **14.7%** and **20%** improvement relative to LM-WEATHER-ave, in the forecasting and imputation, respectively. In relation to recent SOTA methods such as FL-PatchTST, FL-LightTS, and FL-DLinear, our LM-WEATHER enhancements surpass **78%**, **14.3%**, and **72.8%** for forecasting, and **102.1%**, **122.1%**, and **96.35%** for imputation. This means that heterogeneity poses challenge to baseline and they struggle to understand weather patterns with limited data. Moreover, it implies that LM-WEATHER can effectively achieve cross-domain knowledge transfer to PLMs. This benfits from the personalized adapter we integrated into the PLM, coupled with lightweight operations.

Table 3: Few-Shot learning results on forecasting task (5% training data). A lower value indicates better performance. **Bold**: the best, Underline: the second best, '-' denotes insufficient data. **Complete results can be found at Appendix E.2**.

| Method | | LM-WEATHER-ave | | LM-WEATHER | | FL-GPT4TS | | FL-Reformer | | FL-Pyraformer | | FL-Dlinear | | FL-PatchTST | | FL-iTransformer | | FL-Lights | | FL-Transformer | | FL-Informer | |
|---|---|---|---|---|---|---|---|---|---|---|---|---|---|---|---|---|---|---|---|---|---|---|---|
| Metrics | Length | MAE | RMSE | MAE | RMSE | MAE | RMSE | MAE | RMSE | MAE | RMSE | MAE | RMSE | MAE | RMSE | MAE | RMSE | MAE | RMSE | MAE | RMSE | MAE | RMSE |
| ODW1T | 96 | 88.1 | 95.1 | 87.3 | **93.9** | 91.6 | 100.9 | 166.9 | 296.0 | 173.6 | 299.2 | 92.4 | 187.5 | **85.1** | 182.7 | 103.3 | 204.8 | 185.8 | 328.1 | 93.7 | 193.5 | 91.1 | 190.0 |
| | 192 | 90.2 | 98.4 | **89.6** | **96.5** | 95.8 | 104.6 | 166.9 | 297.3 | 176.0 | 303.0 | 94.4 | 192.4 | 90.7 | 191.6 | 106.7 | 210.7 | 188.1 | 336.0 | 96.8 | 200.1 | 93.5 | 195.9 |
| | 336 | 94.2 | 101.7 | **92.2** | **99.7** | 100.0 | 108.2 | 168.9 | 297.5 | 177.6 | 303.0 | 95.9 | 193.2 | 96.5 | 197.4 | 108.7 | 211.8 | 188.4 | 334.6 | 100.2 | 203.3 | 99.3 | 201.0 |
| | 720 | - | - | - | - | - | - | - | - | - | - | - | - | - | - | - | - | - | - | - | - | - | - |
| | Avg. | 90.8 | **98.4** | 89.7 | 96.7 | 95.8 | 104.6 | 167.7 | 296.9 | 175.7 | 301.7 | 94.2 | 191.0 | **90.7** | 190.6 | 106.3 | 209.1 | 187.5 | 333.0 | 96.9 | 199.0 | 94.6 | 195.6 |
| ODW1V | 96 | 79.6 | 104.3 | **75.7** | **98.1** | 82.0 | 108.5 | 101.5 | 130.2 | 81.6 | 107.5 | 98.8 | 127.4 | 327.6 | 392.4 | 135.0 | 168.3 | 111.0 | 141.6 | 116.6 | 155.8 | 111.5 | 144.9 |
| | 192 | 87.8 | 115.5 | **82.5** | **108.4** | 90.8 | 120.3 | 107.0 | 136.7 | 90.2 | 118.9 | 110.4 | 141.6 | 334.4 | 403.4 | 145.4 | 180.2 | 117.5 | 149.2 | 123.4 | 164.1 | 116.0 | 152.4 |
| | 336 | 103.9 | 133.4 | **98.7** | **125.4** | 107.0 | 138.7 | 113.2 | 142.4 | 106.1 | 137.5 | 120.0 | 153.2 | 341.6 | 413.7 | 122.1 | 153.5 | 126.3 | 159.7 | 133.6 | 161.3 | 123.2 | 167.4 |
| | 720 | - | - | - | - | - | - | - | - | - | - | - | - | - | - | - | - | - | - | - | - | - | - |
| | Avg. | 90.4 | 117.7 | **85.6** | **110.6** | 93.3 | 122.5 | 107.2 | 136.4 | 92.6 | 121.3 | 109.7 | 140.7 | 334.5 | 403.2 | 134.2 | 167.3 | 118.3 | 150.2 | 124.5 | 160.4 | 116.9 | 154.9 |
| ODW2T | 96 | 111.0 | 159.4 | 99.0 | 135.5 | 127.9 | 178.2 | 158.3 | 241.2 | 173.3 | 247.1 | 107.1 | 152.8 | 101.2 | 147.9 | 115.9 | 166.3 | 183.6 | 273.3 | 142.3 | 199.6 | 158.8 | 201.3 |
| | 192/336/720 | - | - | - | - | - | - | - | - | - | - | - | - | - | - | - | - | - | - | - | - | - | - |
| | Avg. | 111.0 | 159.4 | **99.0** | **135.5** | 127.9 | 178.2 | 158.3 | 241.2 | 173.3 | 247.1 | 107.1 | 152.8 | 101.2 | 147.9 | 115.9 | 166.3 | 183.6 | 273.3 | 142.3 | 199.6 | 158.8 | 201.3 |
| ODW2V | 96 | 105.3 | 135.7 | 96.8 | 122.1 | 110.2 | 155.4 | 151.5 | 190.7 | 150.5 | 189.3 | 112.2 | 141.2 | 115.5 | 145.8 | 110.2 | 143.4 | 162.1 | 212.5 | 106.4 | 136.8 | 149.6 | 188.2 |
| | 192/336/720 | - | - | - | - | - | - | - | - | - | - | - | - | - | - | - | - | - | - | - | - | - | - |
| | Avg. | 105.3 | 135.7 | **96.8** | **122.1** | 110.2 | 155.4 | 151.5 | 190.7 | 150.5 | 189.3 | 112.2 | 141.2 | 115.5 | 145.8 | 110.2 | 143.4 | 162.1 | 212.5 | 106.4 | 136.8 | 149.6 | 188.2 |
| $1^{st}$ Count | | | 1 | | 16 | | 0 | | 0 | | 0 | | 0 | | 2 | | 0 | | 0 | | 0 | | 0 |

Table 4: Few-Shot learning results on imputation task (5% training data), where random masking ratio is 50%. A lower value indicates better performance. **Bold**: the best, Underline: the second best, '-' denotes insufficient data. **Appendix E.2 shows our full results.**

| Method | | LM-WEATHER-ave | | LM-WEATHER | | FL-GPT4TS | | FL-Reformer | | FL-Pyraformer | | FL-DLinear | | FL-PatchTST | | FL-iTransformer | | FL-LightTS | | FL-Transformer | | FL-Informer | |
|---|---|---|---|---|---|---|---|---|---|---|---|---|---|---|---|---|---|---|---|---|---|---|---|
| Ratio | Length | MAE | RMSE | MAE | RMSE | MAE | RMSE | MAE | RMSE | MAE | RMSE | MAE | RMSE | MAE | RMSE | MAE | RMSE | MAE | RMSE | MAE | RMSE | MAE | RMSE |
| ODW1T | 96 | 61.2 | 121.2 | **59.9** | **120.8** | 62.4 | 138.6 | 147.4 | 261.3 | 149.5 | 256.4 | 110.0 | 209.1 | 64.2 | 147.0 | 119.0 | 228.5 | 173.0 | 310.8 | 140.8 | 260.0 | 143.9 | 264.7 |
| | 192 | 69.1 | 130.2 | 64.7 | **127.7** | 67.3 | 145.2 | 151.3 | 267.8 | 152.0 | 258.1 | 110.1 | 203.4 | 74.1 | 155.1 | 120.9 | 223.0 | 172.2 | 301.4 | 149.1 | 262.4 | 150.3 | 264.2 |
| | 336/720 | - | - | - | - | - | - | - | - | - | - | - | - | - | - | - | - | - | - | - | - | - | - |
| | Avg. | 65.2 | 125.7 | **62.3** | **124.4** | 64.9 | 141.9 | 149.4 | 264.6 | 150.8 | 257.3 | 110.0 | 206.2 | 69.2 | 151.0 | 120.0 | 225.7 | 172.6 | 306.1 | 145.0 | 261.2 | 147.1 | 264.4 |
| ODW1V | 96 | 62.2 | 134.1 | 62.8 | 135.5 | 67.3 | **131.2** | 103.9 | 189.8 | **61.5** | 132.6 | 112.2 | 208.4 | 161.1 | 281.5 | 117.6 | 219.5 | 119.6 | 223.5 | 98.0 | 198.5 | 94.2 | 188.1 |
| | 192 | 71.4 | 140.5 | 72.2 | 142.1 | 74.6 | 152.7 | 103.3 | 182.4 | **70.6** | **138.8** | 113.3 | 200.6 | 160.5 | 272.4 | 124.5 | 218.9 | 122.7 | 217.8 | 101.8 | 191.3 | 96.7 | 181.5 |
| | 336/720 | - | - | - | - | - | - | - | - | - | - | - | - | - | - | - | - | - | - | - | - | - | - |
| | Avg. | 66.8 | 137.3 | 67.5 | 138.8 | 71.0 | 142.0 | 103.6 | 186.1 | **66.1** | **135.7** | 112.7 | 204.5 | 160.8 | 276.9 | 121.0 | 219.2 | 121.2 | 220.7 | 101.8 | 194.9 | 95.5 | 184.8 |
| ODW2T | 96 | 102.5 | 156.3 | **99.4** | **151.6** | 112.0 | 157.2 | 116.2 | 161.3 | 124.9 | 165.6 | 123.7 | 178.3 | 173.0 | 256.7 | 127.3 | 190.6 | 133.8 | 200.3 | 124.3 | 187.5 | 105.7 | 161.1 |
| | 192/336/720 | - | - | - | - | - | - | - | - | - | - | - | - | - | - | - | - | - | - | - | - | - | - |
| | Avg. | 102.5 | 156.3 | **99.4** | **151.6** | 112.0 | 157.2 | 116.2 | 161.3 | 124.9 | 165.6 | 123.7 | 178.3 | 173 | 256.7 | 127.3 | 190.6 | 133.8 | 200 | 124.3 | 188 | 105.7 | 161.0 |
| ODW2V | 96 | 42.4 | 62.9 | **35.7** | 112.1 | 56.4 | 77.3 | 106.8 | 135.5 | 70.8 | **95.5** | 113.3 | 148.0 | 153.8 | 199.5 | 101.8 | 136.4 | 106.1 | 142.2 | 100.1 | 134.6 | 89.8 | 119.0 |
| | 192/336/720 | - | - | - | - | - | - | - | - | - | - | - | - | - | - | - | - | - | - | - | - | - | - |
| | Avg. | 42.4 | 62.9 | **35.7** | 112.1 | 56.4 | 77.3 | 106.8 | 135.5 | 70.8 | **95.5** | 113.3 | 148.0 | 153.8 | 199.5 | 101.8 | 136.4 | 106.1 | 142.2 | 100.1 | 134.6 | 89.8 | 119.0 |
| $1^{st}$ Count | | | 0 | | 12 | | 1 | | 0 | | 6 | | 0 | | 0 | | 0 | | 0 | | 0 | | 0 |

## 5.5 Zero-Shot Learning (Out of Distribution Modeling) Experiments

Beyond few-shot learning, PLMs hold potential as effective zero-shot reasoners. We evaluate the zero-shot learning capabilities of LM-WEATHER within the framework of cross-domain adaption. Specifically, we examine how well a method performs on a dataset when it is optimized on another dataset, where the model has not encountered any data samples from the original dataset. We use forecasting/imputation protocol and evaluate on various cross-domain scenarios. Note that we choose LM-WEATHER-AVE rather than LM-WEATHER for

Table 5: Results on Zero-Shot Learning (ave. MAE on forecasting/imputation tasks report). **Bold**: the best, Underline: the second best, ⇔: domain transferring between datasets.

| Setting | LM-WEATHER-AVE | FL-GPT4TS | FL-DLinear | FL-PatchTST |
|---|---|---|---|---|
| **1T ⇔ 1V** | 54.2/48.9 | 59.4/**48.4** | **50.2**/54.9 | 67.4/69.5 |
| **1T ⇔ 2V** | 92.1/**33.2** | **89.9**/34.2 | 99.4/80.4 | 96.7/56.4 |
| **1T ⇔ 2T** | **80.4**/48.4 | 87.4/53.5 | 94.8/67.2 | 86.5/71.1 |
| **2T ⇔ 2V** | **84.9**/**33.3** | 88.2/36.4 | 106.5/99.1 | 92.1/55.5 |
| **2T ⇔ 1V** | **57.7**/49.6 | 58.3/**47.5** | 69.1/75.3 | 74.2/71.1 |
| **2T ⇔ 1T** | **59.5**/25.5 | 63.1/27.1 | 78.3/57.1 | 61.2/38.6 |
| **1V ⇔ 2V** | **90.1**/36.9 | 96.7/**27.2** | 114.2/101.2 | 104.7/59.2 |
| **1V ⇔ 2T** | **79.3**/46.1 | 84.5/47.5 | 96.9/71.2 | 89.7/77.4 |
| **1V ⇔ 1T** | **51.2**/25.8 | 53.8/27.2 | 67.7/54.0 | 56.4/32.7 |
| **2V ⇔ 1V** | **56.0**/51.8 | 58.5/54.2 | 70.4/74.2 | 72.7/69.9 |
| **2V ⇔ 1T** | **59.5**/29.6 | 63.1/**30.9** | 72.9/59.8 | 60.4/39.9 |
| **2V ⇔ 2T** | **72.1**/44.3 | 76.9/**41.2** | 87.4/66.7 | 80.5/65.7 |
| $1^{st}$ Count | **18** | 5 | 1 | 0 |

comparison due to it can obtain an unified model for zero-shot experiments whereas LM-WEATHER is obtain multiple personalized models. The results are in **Tab. 5**. LM-WEATHER-AVE consistently outperforms the most competitive baselines by a large margin, over **14.2%** and **14.2%** w.r.t the second-best in MAE reduction, in forecasting and imputation, respectively. We attribute this to our personalized adapter that we implant in PLMs being better at activating the PLM's knowledge transfer and domain-adaption capabilities in a resource-efficient manner when modeling weather variables.

## 5.6 Framework Analysis Experiments

We demonstrate the effectiveness of LM-WEATHER through experiments focused on ablation studies, computational/communication comparison, and robustness evaluation. For detailed results and further analysis, please refer to the **Appendix D** and **Appendix E**.

**Ablation Study.** Follow the setting of main experiments, we report our brief ablation results in **Tab. 6**, please refer to **Appendix E.3** for full results. The results indicate a notable drop in performance when we omit the weather decomposition components (LM-WEATHER-A/B/C/D). Additionally, keeping the decomposition term but removing the associated generator leads to a 14.5% average performance decline. This suggests that our personalized adapter effectively leverages the PLM's modeling of weather data. Conversely, when we alter the personalized approach by changing the shared low-rank matrix to other trainable parameters (LM-WEATHER-F), we observe a significant performance drop and increased communication costs. Furthermore, moving from LoRA to fully fine-tuning the attention parameters results in a slight performance gain but incurs over four times the parameter count and a massive increase in communication overhead, which is inefficient for us. These outcomes highlight the benefits of the personalized adapter.

Table 6: Ablation results on forecasting (multivariate to multivariate) and imputation (50% masking ratio, **OWD1T** dataset). A lower value indicates better performance. **Bold**: the best, Underline: the second best, ↓ and ↑ denote performance degradation and performance improvement, respectively.

| Method | Task | | Ablation Perspective | | Ave. Variations | | Params.# | |
|---|---|---|---|---|---|---|---|---|
| | Forecasting | Imputation | Model Component | Personalized Method | Forecasting | Imputation | Train.# | Comm.# |
| LM-WEATHER | 45.4/74.6 | 23.1/40.0 | Original | Original | - | - | 10.38 M | 0.38 M |
| LM-WEATHER-A | 50.8/87.6 | 26.0/47.7 | *wo* Decomposition | Original | ↓11.8% | ↓12.6% | 10.38 M | 0.38 M |
| LM-WEATHER-B | 50.9/85.6 | 25.4/47.1 | *wo* Trend Component | Original | ↓12.1% | ↓10.0% | 10.37 M | 0.38 M |
| LM-WEATHER-C | 50.1/83.6 | 25.0/46.1 | *wo* Seasonal Component | Original | ↓10.3% | ↓8.2% | 10.37 M | 0.38 M |
| LM-WEATHER-D | 49.3/81.7 | 24.4/45.6 | *wo* Residual Component | Original | ↓8.6% | ↓5.6% | 10.37 M | 0.38 M |
| LM-WEATHER-E | 53.8/95.6 | 25.5/47.0 | *wo* Prompt Generator | Original | ↓18.5% | ↓10.4% | 10.36 M | 0.38 M |
| LM-WEATHER-F | 49.4/82.3 | 28.1/52.0 | Original | *w* LoRA, Local: Low-Rank Matrix, Global: the rest of trainable param. | ↓8.8% | ↓21.6% | 10.38 M | 10.00 M |
| LM-WEATHER-G | 43.2/71.4 | 22.4/39.1 | Original | *wo* LoRA, Local: Attention Param. Global: Attention Param | ↑5.1% | ↑3.1% | 52.01 M | 41.99 M |
| LM-WEATHER-H | **42.7/71.2** | **22.2/39.3** | Original | *wo* LoRA, Local: Attention Param. Global: the rest of trainable param. | ↑6.3% | ↑4.1% | 52.01 M | 10.00 M |

**Parameter Comparison.** The results are shown in **Tab. 7**. LM-WEATHER ensures top while only communicate about **3.7%** of the trainable parameters, compared to the baseline that communicates the full model parameters. When compared with competitive methods, FL-DLinear and FL-LightTS, LM-WEATHER's communication overhead is just **35.9%** and **22.6%** of theirs, respectively, highlighting LM-WEATHER's superior communication efficiency.

Table 7: Experiment results on parameter comparison (ave. MAE/RMSE report), **Bold**: the best.

| Method | Task | | Params.# | | |
|---|---|---|---|---|---|
| | Forecasting | Imputation | Train. | Comm. | Ratio |
| LM-WEATHER | **45.4/74.6** | **23.1/42.2** | 10.38 M | 0.38 M | 3.70% |
| FL-GPT4TS | 49.9/82.5 | 25.1/46.2 | 12.42 M | 12.42 M | 100% |
| FL-Reformer | 78.2/98.7 | 69.8/92.5 | 19.74 M | 19.74 M | 100% |
| FL-Pyraformer | 73.0/91.3 | 68.0/89.7 | 153.32 M | 153.32 M | 100% |
| FL-DLinear | 63.3/82.8 | 28.5/49.9 | 1.06 M | 1.06 M | 100% |
| FL-PatchTST | 48.6/81.0 | 45.4/73.5 | 74.74 M | 74.74 M | 100% |
| FL-Itransformer | 53.7/63.7 | 27.6/48.2 | 26.74 M | 26.74 M | 100% |
| FL-LightTS | 62.7/93.4 | 26.1/45.7 | 1.68 M | 1.68 M | 100% |
| FL-Transformer | 52.8/84.7 | 57.6/82.3 | 45.55 M | 45.55 M | 100% |
| FL-Informer | 53.4/85.2 | 61.4/85.9 | 52.31 M | 52.31 M | 100% |

**Communication Efficiency.** To further validate the excellent communication efficiency of LM-WEATHER, we introduce quantitative comparisons by including FL methods tailored to improve communication efficiency (FedKD [27], FedPAQ [28], FedBF [29], FedAP [29], PromptFL [30]) as baselines[2]. The results is shown in **Table 8**, which demonstrate our LM-WEATHER achieves a significant improvement in communication efficiency while maintaining excellent performance. Additionally, LM-WEATHER significantly outperforms baseline in terms of both communication efficiency and performance across different tasks. Even when compared to lightweight baselines

---

[2]Due to scenario and model differences, we modified these baselines for LM-WEATHER by applying solely their strategies to improve communication efficiency, as detailed in **Appendix B.2**

(i.e., FL-LightTS/DLinear), LM-WEATHER continues to outperform them. This underscores LM-WEATHER's superiority in both communication efficiency and performance.

Table 8: Comparison of LM-WEATHER and baseline that tailored to improve communication efficiency in terms of forecasting (multivariate-multivariate)/imputation (50% masking rate) performance as well as communication efficiency, with × denotes the improvement in communication efficiency relative to the standard line (LM-WEATHER-Ave), MAE/RMSE report. **Bold**: the best.

| Method | Forecasting | Imputation | Train. | Comm. Params. | Comm. |
|---|---|---|---|---|---|
| FL-Pyraformer | 73.0/91.3 | 68.0/89.7 | 153.32 M | 153.32 M | 0.07× |
| FL-PatchTST | 48.6/81.0 | 45.4/73.5 | 74.74 M | 74.74 M | 0.14× |
| FL-LightTS | 62.7/93.4 | 26.1/45.7 | 1.68 M | 1.68 M | 6.2× |
| FL-DLinear | 63.3/82.8 | 28.5/49.9 | 1.06 M | 1.06 M | 9.8× |
| LM-WEATHER-Ave | 47.5/78.7 | 24.0/44.1 | 10.38 M | 10.38 M | 1× |
| LM-WEATHER (Ours) | **45.4/74.6** | **23.1/42.4** | 10.38 M | **0.38 M** | **27.3×** |
| LM-WEATHER (w FedKD) | 49.6/76.2 | 27.5/43.6 | 10.38 M | 1.68 M | 6.2× |
| LM-WEATHER (w FedPer) | 52.1/79.0 | 25.1/44.3 | 10.38 M | 8.46 M | 1.2× |
| LM-WEATHER (w FedBF) | 46.2/78.1 | 23.7/44.0 | 10.49 M | 10.49 M | 0.9× |
| LM-WEATHER (w FedAP) | 47.4/79.2 | 24.3/44.7 | 10.38 M | 9.6 M | 1.1× |
| LM-WEATHER (w PromptFL) | 46.0/78.4 | 23.8/45.1 | 10.38 M | 8.4 M | 1.2× |

**Robustness to Number of Devices.** To evaluate LM-WEATHER's robustness against device count variations, we assessed the percentage change in performance relative to the default device number. Our results (**Tab. 9**) reveals that LM-WEATHER maintains robustness across different device counts due to several factors: **(1)** Increasing device numbers during training typically yields slight performance improvements within a stable range, applicable in both regular and few-shot scenarios. **(2)** Additional devices can sometimes impair performance due to imbalances in data distribution, highlighting non-proportional gains. **(3)** Adding more devices increases communication overhead, which may not justify minor improvements, especially in resource-limited settings. These findings underscore LM-WEATHER's relative resilience to device count variations and its ability to strike an optimal balance between performance enhancement and communication overhead.

Table 9: Results of LM-WEATHER under forecasting (multivariate-multivariate) and imputation (50% masking rate) at different device participation rates $[0.1, 0.3, 0.5, 0.7, 0.9]$, $\uparrow/\downarrow$ implies an increase/decrease in performance relative to the original setting (0.1), MAE/RMSE report, where 15% represents the proportion of data on each client involved in training.

| Dataset | Rate / Devices | Normal | | Few-Shot (15%) | |
|---|---|---|---|---|---|
| | | Forecasting | Imputation | Forecasting | Imputation |
| **ODW1T** | 0.1 (2/round) | 44.4/73.6 | 22.6/42.0 | 64.7/100.4 | 40.2/68.2 |
| | 0.3 (5/round) | 43.7/72.5 (↑ 1.55) | 24.2/43.7 (↓ 5.55) | 63.4/99.7 (↑ 1.40) | 41.4/68.7 (↓ 1.85) |
| | 0.5 (8/round) | 42.9/72.0 (↑ 2.85) | 21.0/42.1 (↑ 3.90) | 63.7/99.2 (↑ 1.40) | 42.3/68.5 (↓ 2.8) |
| | 0.7 (11/round) | 43.9/74.1 (↑ 0.25) | 21.8/41.2 (↑ 2.80) | 64.5/101.0 (↑ 0.10) | 39.5/66.7 (↑ 2.00) |
| | 1.0 (16/round) | 44.2/74.0 (0 -) | 21.3/41.6 (↑ 3.10) | 63.6/100.2 (↑ 0.95) | 40.4/68.0 (↓ 0.1) |
| **ODW2T** | 0.1 (4/round) | 66.2/89.1 | 37.2/54.9 | 89.7/131.8 | 77.2/112.6 |
| | 0.3 (9/round) | 68.2/89.7 (↓ 1.85) | 36.5/53.1 (↑ 2.65) | 90.2/132.5 (↓ 0.55) | 75.4/110.3 (↑ 2.25) |
| | 0.5 (18/round) | 65.4/89.2 (↑ 0.55) | 36.7/53.4 (↑ 2.05) | 89.1/131.4 (↑ 0.50) | 76.5/111.2 (↑ 1.10) |
| | 0.7 (25/round) | 65.7/88.8 (↑ 0.90) | 36.1/53.9 (↑ 2.45) | 88.9/130.9 (↑ 0.80) | 76.9/112.3 (↑ 0.35) |
| | 1.0 (36/round) | 65.9/89.0 (↑ 0.25) | 36.9/55.0 (↑ 0.30) | 89.1/130.7 (↑ 0.75) | 76.7/112.1 (↑ 0.50) |

## 6 Conclusion and Future Works

This paper demonstrate that pre-trained language models (PLMs) are strong foundation models for personalized on-device meteorological variable modeling. We propose LM-WEATHER, a generic framework to taming PLMs to acquire highly customized models for heterogeneous meteorological data on devices while keeping high efficiency. Concretely, we introduce a lightweight personalize adapter into PLMs and endow it with weather pattern awareness. Experiments on real-world datasets demonstrate that LM-WEATHER outperforms the SOTA results by a large margin across various tasks. In addition, extensive analyses indicate that LM-WEATHER can (1) effectively achieve cross-domain knowledge transfers, (2) render device with highly customized model while keeping high efficiency, and (3) generalize under few-shot and zero-shot scenario. In future work, we plan to extend LM-WEATHER to multimodal weather data (text/image/time-series) and to finer scales.

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

# APPENDIX: Personalized Adapter for Large Meteorology Model on Devices: Towards Weather Foundation Models

The appendix includes missing information from our main text, including: Appendix A More Related Work; Appendix B Experimental Details; Appendix C Theorems and Proof; Appendix D Additional Finding & Experiment & Discussion; Appendix E Full Experimental Results and Appendix F Additional Statements.

## Appendix A    More Related Work

In this section, we will discuss in detail advances relevant to our work, which include weather variable modeling, personalized federated learning, universal time series learning, and large language models (LLMs) in time series.

**From Meteorological Variable Modeling to Weather Forecasting.**    Weather conditions play a crucial role in sectors such as transportation, tourism, and agriculture. Meteorological factors, including temperature, humidity, and precipitation, provide essential support and historical insights that enable researchers to analyze weather trends. For decades, Numerical Weather Prediction (NWP) [14] has been the prevalent method, employing physical models to simulate and forecast atmospheric dynamics. However, the accuracy of NWP can be compromised by the uncertainty of initial conditions in differential equations [31, 32], particularly in complex atmospheric processes, and it requires significant computational resources [5, 33, 34].

The recent exponential growth in weather data has prompted a shift from traditional physics-based methods to data-driven approaches using machine learning (ML) and deep learning (DL), which bypass physical constraints in meteorological variables [5]. DL strategies, with their deeper representational capabilities, generally surpass ML methods. Various deep network architectures have been employed to perform extensive weather modeling using large-scale reanalysis data [1, 4, 35, 36, 37]. Yet, these methods tend to focus on global weather patterns, often overlooking the specifics of regional weather variables, and thus fail to offer detailed regional analyses. Moreover, these models require extensive datasets and substantial computational resources—for example, some need to train on 192 NVIDIA Tesla V100 GPUs for 16 days [4]. Additionally, prevailing models assume centralized data storage, which contrasts with the decentralized data collection from diverse ground weather stations. Our research addresses these challenges by focusing on regional meteorological variables in low-resource settings, aiming to provide reliable analytical support for weather pattern modeling and understanding.

**Personalized Federated Learning.**    Federated learning (FL) [7] is a distributed learning paradigm that facilitates the collaborative training of models without exposing data from each participant. Personalized FL (PFL) aims to train a personalized model for each client. Existing PFLs are based on various techniques. Refs. [38, 39, 40] add a regularization term that benefits decomposing the personalized model optimization from global model learning. Refs. [8, 41] share part of the model and keep personalized layers private to achieve personalization. Ref. [42] enables a more flexible personalization by adaptive weighted aggregation. Ref. [43] study PFL from a Model-Agnostic Meta-Learning where a meta-model is learned to generate the initialized local model for each client. This paper tackles on-device meteorological variable modeling from PFL perspective.

**Universal Time Series Learning.**    On-device meteorological variable modeling addresses time series analysis of complex weather patterns on diverse, low-resource devices. We have expanded this to include task-specific time series learning. Recent advancements have enhanced Transformer [44] for time series forecasting by integrating signal processing techniques such as patching [6], exponential smoothing [45], decomposition [24], and frequency analysis [46]. Among them, PatchTST [6] improves the accuracy of long-term forecasting compared to other Transformer models. ETSFormer [47] applies principles of power series smoothing within the Transformer framework to boost efficiency. Similarly, FEDformer [46] merges the Transformer with seasonal & trend decomposition, offering improved performance and efficiency. Autoformer [45] leverages sequence periodicity for better dependency discovery and representation, excelling in both efficiency and accuracy.

While these methods excel in efficiency and accuracy, they are typically tailored for narrow-range forecasting on select classical time series datasets. Real-world weather data, however, often displays more complex patterns and interconnected variable relationships. Furthermore, weather modeling extends beyond forecasting, rendering these methods less effective for weather sequences. To improve modeling for intricate weather sequences, models need the flexibility to adjust to complex distributions and various tasks with minimal training. The ideal weather models would capture weather patterns accurately, facilitating knowledge transfer, such as between regions. However, creating versatile weather models remains a challenging endeavor. Recent studies have started to examine the potential of large-scale climate models [1, 4], utilizing simulated datasets advances. Yet, their generalizability is hindered by data differences, complex architectures, and the vast number of model parameters.

**LLMs in Time Series.**    Large language models (LLMs) have spurred advances in natural language processing (NLP). Although time series modeling hasn't seen similar leaps, the impressive capabilities of LLMs have led to their use in this field. In general, pre-trained LLMs are often fine-tuned or reprogrammed to model

time series [18, 11, 10, 17]. Among them, PROMPTCAST [16] and HEALTHLEARNER [26] treat time series as "text sequences," inputting them directly into LLMs and using prompts for forecasting. [17] dencodes time series as embeddings for LLM output, showing LLMs' strength in time series analysis. LLM4TS [18] uses a two-stage fine-tuning approach to adapt LLMs to time series data. TEMPO [10] breaks down time series features to leverage LLMs in prediction tasks, while TIME-LLM [11] fine-tunes LLMs with multimodal data, integrating relevant text prompts for efficient analysis. However, these approaches focus on centralized time series modeling and overlook the complexities of real-world distributed settings. Weather data, in particular, has unique challenges like heterogeneity from geographic factors and privacy concerns, making central training methods both risky and difficult.

## Appendix B   Experimental Details

### B.1   Datasets

Despite the proliferation of reanalysis data aimed at building frameworks for global climate analysis, these datasets often struggle to model regional weather trend due to: (1) they depend on numerous simulations of atmospheric equations, introducing biases inconsistent with real observations, and (2) they face challenges in refining their scale to suit specific regional applications. Hence, we collected real observational data from various weather stations across different regions. We then organized this data into two series, each comprising two distinct datasets, to underscore the heterogeneity inherent in real-world settings.

**On-device Weather Series 1# (ODW1).**   The dataset gathered from 15 ground weather stations across China, Japan, and South Korea, encompasses over 20 variables[3]. It has been divided into two subsets: **ODW1T** has a heterogeneous time span, meaning the data collection start and end times vary by location. and **ODW1V** extends **ODW1T** by adding variability in the observed variables; while one variable remains constant at each station, the others vary. The temporal resolution of the dataset is 1h. Details are presented in **Tab. 10** and **Tab. 11**.

Table 10: Details about **ODW1T** dataset, where **Start** and **End** indicate the respective beginning and ending timestamps of data collected at a specific weather station, **Samples** denotes the count of weather sequence samples gathered at that station, and **Variables** refers to the weather variables included in the data from each station (For the full names of these variables, please refer to **Tab. 14**).

| Weather Station | Start (UTC+0) | End (UTC+0) | Num. of Samples # | Variables |
|---|---|---|---|---|
| Hua-Nan | 06/11/2018 16:00 | 03/19/2020 00:00 | 230,280 | |
| E-Min | 01/23/2019 16:00 | 06/07/2024 04:00 | 240,280 | |
| Huhehaote | 09/02/1028 01:00 | 06/01/2020 07:00 | 306,400 | |
| Yin-Chuan | 10/31/2018 20:00 | 05/06/2020 07:00 | 265,220 | |
| Shen-Yang | 04/09/2018 01:00 | 08/30/2019 06:00 | 243,980 | |
| Hai-Dian | 05/29/2017 16:00 | 12/05/2018 11:00 | 266,340 | |
| Xin-Du | 01/29/2020 08:00 | 06/29/2021 00:00 | 248,040 | |
| Lin-Zhi | 10/13/2019 08:00 | 03/30/2021 09:00 | 256,380 | ap, t, mxt, mnt, dt, rh, wvp, p1, p2, p3, p4, p5, wd, ws, mwd, mws, st, hv1, hv2, vv |
| Kun-Ming | 08/28/2018 07:00 | 06/13/2020 22:00 | 314,740 | |
| Wu-Han | 05/20/2019 01:00 | 02/17/2021 21:00 | 247,160 | |
| Tokyo | 07/17/2017 23:00 | 01/27/2019 14:00 | 248,180 | |
| Nagoya | 05/10/2017 15:00 | 02/14/2019 17:00 | 309,680 | |
| Hiroshima | 05/27/2019 10:00 | 11/01/2020 03:00 | 251,420 | |
| Seoul | 04/28/2017 04:00 | 11/20/2018 00:00 | 274,040 | |
| Busan | 02/24/1029 18:00 | 07/17/2020 17:00 | 244,340 | |

**On-device Weather Series 2# (ODW2).**   This dataset consists of data from 36 weather observation stations in the United States, Canada, and Israel, covering 5 different variables with a temporal resolution of 1 hour. Following the dataset setting of **ODW1**, the dataset was also subdivided into two different dataset, including **ODW2T** and **ODW2V**. Detailed information are presented in **Tab. 12** and **Tab. 13**.

**Remark.**   Four standard steps were performed during the collection and compilation of these dataset, as shown below:

[1] **Collection of Raw Meteorological Data.** Raw data collection represents the foundational and initial step in constructing our dataset. We procure open-source raw data from various national meteorological centers and data repositories, including the National Meteorological Science Data Center of China[4], Korea Meteorological Administration[5], Global Surface Meteorological Observations

---

[3]We treat the value of each meteorological variable at each timestamp as an independent sample uniformly.

[4]https://data.cma.cn/

[5]https://www.kma.go.kr/

Table 11: Details about **ODW1V** dataset, where **Start** and **End** indicate the respective beginning and ending timestamps of data collected at a specific weather station, **Num. of Samples #** denotes the count of weather sequence samples gathered at that station, and **Fixed Variables** refers to the shared variables among different weather stations, and **Other Variables** is the remain weather variables in each weather station (For the full names of these variables, please refer to **Tab. 14**).

| Weather Station | Start (UTC+0) | End (UTC+0) | Num. of Samples # | Fixed Variable | Other Variables |
|---|---|---|---|---|---|
| Hua-Nan | 06/11/2018 16:00 | 03/19/2020 00:00 | 69,084 | | ws, p4, p1, p5, p2 |
| E-Min | 01/23/2019 16:00 | 06/07/2024 04:00 | 72,072 | | p1, mwd, p2, wvp, ws |
| Huhehaote | 09/02/1028 01:00 | 06/01/2020 07:00 | 91,920 | | vv, p3, dt, mwd, p4 |
| Yin-Chuan | 10/31/2018 20:00 | 05/06/2020 07:00 | 79,566 | | ws, dt, mnt, p3, p2 |
| Shen-Yang | 04/09/2018 01:00 | 08/30/2019 06:00 | 73,194 | | wd, st, hv1, mwd, vv |
| Hai-Dian | 05/29/2017 16:00 | 12/05/2018 11:00 | 79,902 | | p5, rh, ap, mxt, mwd |
| Xin-Du | 01/29/2020 08:00 | 06/29/2021 00:00 | 74,412 | | p1, hv1, wvp, mxt, p5 |
| Lin-Zhi | 10/13/2019 08:00 | 03/30/2021 09:00 | 76,914 | Temperature | ws, vv, p1, hv1, p5 |
| Kun-Ming | 08/28/2018 07:00 | 06/13/2020 22:00 | 94,422 | | hv2, mws, mnt, p2, mwd |
| Wu-Han | 05/20/2019 01:00 | 02/17/2021 21:00 | 92,148 | | st, ws, p3, p5, rh |
| Tokyo | 07/17/2017 23:00 | 01/27/2019 14:00 | 80,454 | | p5, st, hv1, ws, hv2 |
| Nagoya | 05/10/2017 15:00 | 02/14/2019 17:00 | 92,904 | | mwd, p1, mws, mnt, st |
| Hiroshima | 05/27/2019 10:00 | 11/01/2020 03:00 | 75,426 | | p4, mwd, wd, hv1, dt |
| Seoul | 04/28/2017 04:00 | 11/20/2018 00:00 | 82,212 | | hv1, mwd, vv, rh, p4 |
| Busan | 02/24/1029 18:00 | 07/17/2020 17:00 | 73,302 | | p4, p3, wvp, p1, vv |

Table 12: Details about **ODW2T** dataset, where **Start** and **End** indicate the respective beginning and ending timestamps of data collected at a specific weather station, **Num. of Samples #** denotes the count of weather sequence samples gathered at that station, and **Variables** refers to the weather variables included in the data from each station (For meaning of variables, please refer to **Tab. 14**).

| Weather Station | Start (UTC+0) | End (UTC+0) | Num. of Samples # | Variables |
|---|---|---|---|---|
| Albuquerque | 02/26/2016 01:00 | 11/16/2016 19:00 | 31,780 | |
| Atlanta | 03/24/2013 21:00 | 12/05/2013 18:00 | 30,715 | |
| Beersheba | 05/06/2014 06:00 | 03/26/2015 18:00 | 35,350 | |
| Boston | 04/06/2015 10:00 | 03/23/2016 00:00 | 35,120 | |
| Charlotte | 08/10/2013 09:00 | 04/24/2014 14:00 | 30,875 | |
| Chicago | 06/09/2015 17:00 | 03/22/2016 10:00 | 34,415 | |
| Dallas | 04/11/2013 00:00 | 08/26/2014 11:00 | 35,463 | |
| Denver | 06/23/2015 22:00 | 05/18/2016 02:00 | 39,510 | |
| Detroit | 10/30/2012 06:00 | 08/22/2013 22:00 | 35,610 | |
| Eilat | 11/24/2012 19:00 | 08/02/2013 05:00 | 30,060 | |
| Haifa | 05/18/2013 21:00 | 04/16/2014 14:00 | 39,915 | |
| Houston | 12/19/2013 02:00 | 11/13/2014 02:00 | 39,490 | |
| Indianapoils | 01/10/2016 23:00 | 10/07/2016 04:00 | 32,435 | |
| Jacksonvile | 11/11/2012 16:00 | 09/28/2013 01:00 | 38,455 | |
| Jerusalem | 04/22/2015 03:00 | 02/28/2016 01:00 | 37,440 | |
| Kansas City | 06/03/2015 12:00 | 03/08/2016 21:00 | 33,535 | |
| Las Veges | 11/25/2014 09:00 | 10/19/2015 15:00 | 39,400 | |
| Los Angeles | 07/04/2013 08:00 | 04/28/2014 18:00 | 35,820 | |
| Miami | 02/10/2014 10:00 | 12/05/2014 10:00 | 35,770 | h, p, t, wd, ws |
| Minneapolis | 10/17/2013 03:00 | 08/31/2014 15:00 | 38,230 | |
| Montreal | 08/14/2015 07:00 | 04/26/2016 06:00 | 30,725 | |
| Nahariyya | 12/03/2013 02:00 | 09/18/2014 01:00 | 34,685 | |
| Nashville | 02/08/2017 03:00 | 11/13/2017 15:00 | 33,430 | |
| New York | 09/27/2013 20:00 | 06/14/2014 15:00 | 31,185 | |
| Philadelphia | 12/08/2012 05:00 | 09/13/2013 20:00 | 33,565 | |
| Phoenix | 07/25/2013 15:00 | 06/05/2014 09:00 | 37,780 | |
| Pittsburgh | 10/16/2015 06:00 | 07/19/2016 16:00 | 33,300 | |
| Portland | 10/12/2013 12:00 | 06/27/2014 18:00 | 31,000 | |
| Saint Louis | 09/16/2014 13:00 | 07/21/2015 06:00 | 36,935 | |
| San Antonio | 03/15/2014 01:00 | 12/03/2014 20:00 | 31,665 | |
| San Diego | 07/02/2015 11:00 | 03/18/2016 00:00 | 31,155 | |
| San Francisco | 05/10/2013 06:00 | 01/21/2014 12:00 | 30,760 | |
| Seattle | 08/06/2014 22:00 | 06/28/2015 05:00 | 39,045 | |
| Tel Aviv District | 09/23/2013 06:00 | 06/27/2014 18:00 | 33,310 | |
| Toronto | 12/06/2016 16:00 | 10/19/2017 15:00 | 38,045 | |
| Vancouver | 08/26/2015 10:00 | 06/02/2016 20:00 | 33,780 | |

Historical Dataset [6], Canadian Meteorological Data Center [7], and World Weather Data Repository

---

[6]`https://k.data.cma.cn/mekb/?r=dataService/cdcindex&datacode=A.0020.0002.S001`

Table 13: Details about **ODW2V** dataset, where **Start** and **End** indicate the respective beginning and ending timestamps of data collected at a specific weather station, **Num. of Samples #** denotes the count of weather sequence samples gathered at that station, and **Fixed Variables** refers to the shared variables among different weather stations, and **Other Variables** is the remain weather variables in each weather station (For the full names of these variables, please refer to **Tab. 14**).

| Weather Station | Start (UTC+0) | End (UTC+0) | Num. of Samples # | Fixed Variable | Other Variables |
|---|---|---|---|---|---|
| Albuquerque | 02/26/2016 01:00 | 11/16/2016 19:00 | 19,068 | | ws, wd |
| Atlanta | 03/24/2013 21:00 | 12/05/2013 18:00 | 18,429 | | wd,p |
| Beersheba | 05/06/2014 06:00 | 03/26/2015 18:00 | 21,210 | | t, ws |
| Boston | 04/06/2015 10:00 | 03/23/2016 00:00 | 21,072 | | ws, wd |
| Charlotte | 08/10/2013 09:00 | 04/24/2014 14:00 | 18,525 | | t, p |
| Chicago | 06/09/2015 17:00 | 03/22/2016 10:00 | 20,649 | | t, p |
| Dallas | 04/11/2013 00:00 | 08/26/2014 11:00 | 21,279 | | p, wd |
| Denver | 06/23/2015 22:00 | 05/18/2016 02:00 | 23,706 | | t, p |
| Detroit | 10/30/2012 06:00 | 08/22/2013 22:00 | 21,366 | | p, t |
| Eilat | 11/24/2012 19:00 | 08/02/2013 05:00 | 18,036 | | ws, p |
| Haifa | 05/18/2013 21:00 | 04/16/2014 14:00 | 23,949 | | ws, t |
| Houston | 12/19/2013 02:00 | 11/13/2014 02:00 | 23,694 | | p, wd |
| Indianapoils | 01/10/2016 23:00 | 10/07/2016 04:00 | 19,461 | | t, p |
| Jacksonvile | 11/11/2012 16:00 | 09/28/2013 01:00 | 23,073 | | ws, t |
| Jerusalem | 04/22/2015 03:00 | 02/28/2016 01:00 | 22,464 | | ws, p |
| Kansas City | 06/03/2015 12:00 | 03/08/2016 21:00 | 20,121 | | wd, ws |
| Las Veges | 11/25/2014 09:00 | 10/19/2015 15:00 | 23,640 | | p, t |
| Los Angeles | 07/04/2013 08:00 | 04/28/2014 18:00 | 21,492 | | ws, t |
| Miami | 02/10/2014 10:00 | 12/05/2014 10:00 | 21,462 | Humidity | wd, ws |
| Minneapolis | 10/17/2013 03:00 | 08/31/2014 15:00 | 22,938 | | t, p |
| Montreal | 08/14/2015 07:00 | 04/26/2016 06:00 | 18,435 | | wd, p |
| Nahariyya | 12/03/2013 02:00 | 09/18/2014 01:00 | 20,811 | | p, ws |
| Nashville | 02/08/2017 03:00 | 11/13/2017 15:00 | 20,058 | | wd, ws |
| New York | 09/27/2013 20:00 | 06/14/2014 15:00 | 18,711 | | t, p |
| Philadelphia | 12/08/2012 05:00 | 09/13/2013 20:00 | 20,139 | | ws, wd, h |
| Phoenix | 07/25/2013 15:00 | 06/05/2014 09:00 | 22,668 | | ws, wd |
| Pittsburgh | 10/16/2015 06:00 | 07/19/2016 16:00 | 19,980 | | wd, ws |
| Portland | 10/12/2013 12:00 | 06/27/2014 18:00 | 18,600 | | ws, p |
| Saint Louis | 09/16/2014 13:00 | 07/21/2015 06:00 | 22,161 | | t, wd |
| San Antonio | 03/15/2014 01:00 | 12/03/2014 20:00 | 18,999 | | t, wd |
| San Diego | 07/02/2015 11:00 | 03/18/2016 00:00 | 18,693 | | t, p |
| San Francisco | 05/10/2013 06:00 | 01/21/2014 12:00 | 18,456 | | p, t |
| Seattle | 08/06/2014 22:00 | 06/28/2015 05:00 | 23,427 | | t, wd |
| Tel Aviv District | 09/23/2013 06:00 | 06/27/2014 18:00 | 19,986 | | t, p |
| Toronto | 12/06/2016 16:00 | 10/19/2017 15:00 | 22,827 | | wd, p |
| Vancouver | 08/26/2015 10:00 | 06/02/2016 20:00 | 20,268 | | wd, ws |

from Kaggle [8]. This process ensures that the collected weather data from these sources are consistent in terms of temporal resolution and variable dimensions. All raw data are open-source and can be freely utilized or modified.

[2] **Selection of Critical Meteorological Variables.** To support personalized on-device meteorological variable modeling and enhance regional weather forecasting reliability, we selected twenty representative meteorological variables. These variables, including temperature, barometric pressure, relative humidity, and precipitation, were chosen based on their significant impact on weather conditions. Detailed definitions, physical descriptions, and units of these selected variables are provided in **Table 14**.

[3] **Ensuring Completion of Meteorological Time Series.** In this step, we primarily focus on ensuring the completeness of weather time series data collected from ground weather stations. Incomplete weather time series can generate unreliable predictions, potentially leading to significant unforeseen losses. Most ground weather stations are susceptible to unpredictable events such as power outages and equipment damage, which may result in data gaps. To enhance dataset completeness, we meticulously examined the raw data for missing values across various timestamps and employed a linear interpolation strategy to fill these gaps.

[4] **Handling of Outliers.** Outliers are common in weather time series data. We distinguish between factual outliers, typically caused by extreme weather events (e.g., heavy rainfall, typhoons, thunderstorms), and non-factual outliers, often due to observational device anomalies or sensor malfunctions

---

[7] https://weather.gc.ca/
[8] https://www.kaggle.com/datasets

Table 14: Abbreviations, full names and corresponding units of the different variables in our proposed datasets.

| Abbreviation | Full name | Unit |
| --- | --- | --- |
| ap | Air Pressure | $hpa$ |
| t | Air Temperature | $^\circ C$ |
| mxt | Maximum Temperature | $^\circ C$ |
| mnt | Minimum Temperature | $^\circ C$ |
| dt | Dewpoint Temperature | $^\circ C$ |
| rh | Relative Humidity | $\%$ |
| wvp | Water Vapor Pressure | $hpa$ |
| p1 | Precipitation in 1h | $mm$ |
| p2 | Precipitation in 3h | $mm$ |
| p3 | Precipitation in 6h | $mm$ |
| p4 | Precipitation in 12h | $mm$ |
| p5 | Precipitation in 24h | $mm$ |
| wd | Wind Direction | $^\circ C$ |
| ws | Wind Speed | $ms^{-1}$ |
| mwd | Maximum Wind Direction | $^\circ$ |
| st | Land Surface Temperature | $^\circ C$ |
| hv1 | Horizontal Visibility in 1 min | $m$ |
| hv2 | Horizontal Visibility in 10 min | $m$ |
| vv | Vertical Visibility | $m$ |

at weather stations. We identify significant deviations in a weather variable—for instance, a sudden increase from an average rainfall of 2 mm to 200 mm—as outliers. These are manually corrected; initially, the values are set to zero and then replaced using linear interpolation, reflecting the gradual nature of weather phenomena.

**Visualisation.**   We hope to deepen the reader's understanding of the datasets we have collected and compiled by providing standard visualizations. Considering the overall size of the datasets and the large number of meteorological variables, we have provided visualisations of representative variables here for reference. The visualisation of OWD1 is shown in Fig. 2. Due to the number of devices involved in the OWD2 dataset, we have divided it into two consecutive images for presentation, as shown in Fig 3 and Fig. 4.

## B.2   Baselines

We compare with state-of-the-art time series analysis models and put them into Federated Learning environments, including Transformer-based methods like Transformer [44], Informer [3], Reformer [21], Pyraformer [22], iTransformer [23], and PatchTST [6], and recent competitive models including GPT4TS [17], DLinear [24] and LightTS [48], detailed information about baselines is below:

- **Transformer.** [44] This model uses a self-attention mechanism, popular for time series prediction tasks, to efficiently and accurately learn relationships within a sequence and contextual information.

- **Informer.** [3] An optimized Transformer-based model for long-range time series prediction. It uses ProbSparse self-attention for efficiency, processes long inputs effectively, and employs a fast prediction decoder.

- **Reformer.** [21] This model improves Transformer efficiency by using locality-sensitive hashing for attention and reversible residual layers. It offers better memory efficiency and speed for lengthy sequences without sacrificing performance.

- **Pyraformer.** [22] It features hierarchical pyramidal attention modules with binary trees to capture temporal dependencies across different ranges efficiently, both in time and memory complexity.

- **iTransformer.** [23] The iTransformer adds attention and feedforward networks to the inverse dimension. It embeds time points as variable tokens, using attention to capture multivariate correlations and feedforward networks for nonlinear representation of each token.

- **PatchTST.** [6] This method divides the time series into patches at the subseries level for input to the Transformer. Each channel holds a univariate time series, sharing the same embedding and Transformer weights across all series.

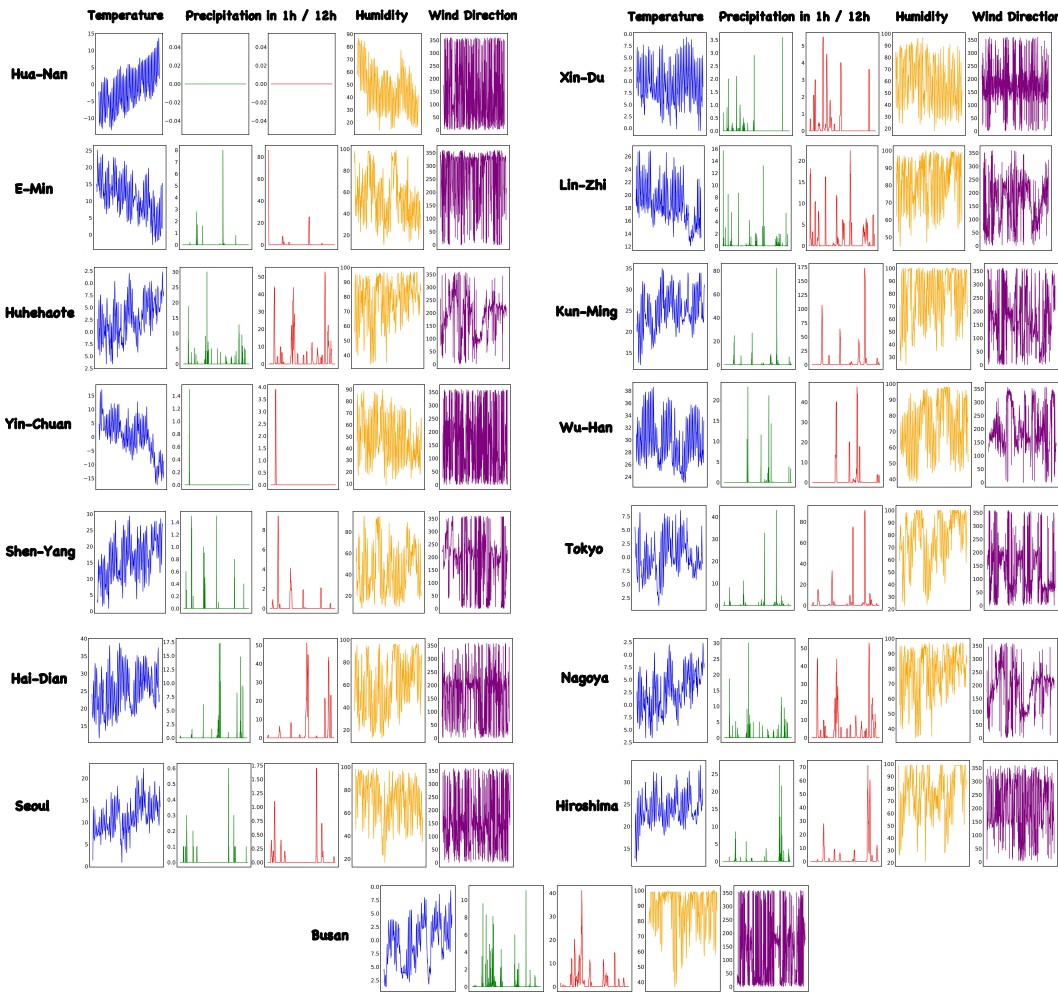

Figure 2: Visualisation of partial variables in ODW1 dataset, where we have selected the first 1,000 time points for presentation. The data distribution from different ground weather stations exhibit significant heterogeneity, and even though the trends of some variables may be similar, there are serious differences in magnitudes. The selected variables are, from left to right, temperature, precipitation in 1-hour/12-hour, humidity, and wind direction.

- **DLinear.** [24] DLinear integrates decomposition schemes from Autoformer and FEDformer with linear layers to model time series data tables. It effectively summarizes trend and seasonal components, enhancing performance on datasets rich in trends.

- **LightTS.** [48] A lightweight structure based on a simple MLP. It utilizes two downsampling strategies—spaced and sequential sampling—on the MLP structure, capitalizing on the fact that downsampled time series generally maintain most of their original information.

- **GPT4TS.** [17] This model is designed for time series analysis across various scenarios, achieved by fine-tuning a pre-trained language model, specifically GPT2, for the time series domain. It's important to note that for a fair comparison, our baseline setup differs from the original publication's configuration. Instead of using the first six layers of GPT2 as the backbone, we align with our approach and utilize only the first five layers.

In addition, pre-trained language models (PLMs) are the key component of our `LM-Weather`, we use different PLMs as the backbone to demonstrate the PLM can as the strong weather foundation model for on-device weather modeling. We use GPT-2 as the default setting, and BERT [49], LLaMA [50] as the alternatives.

- **BERT.** [49] BERT, short for Bidirectional Encoder Representations from Transformers, is a deep learning model that uses the Transformer architecture. It understands the context of words by analyzing

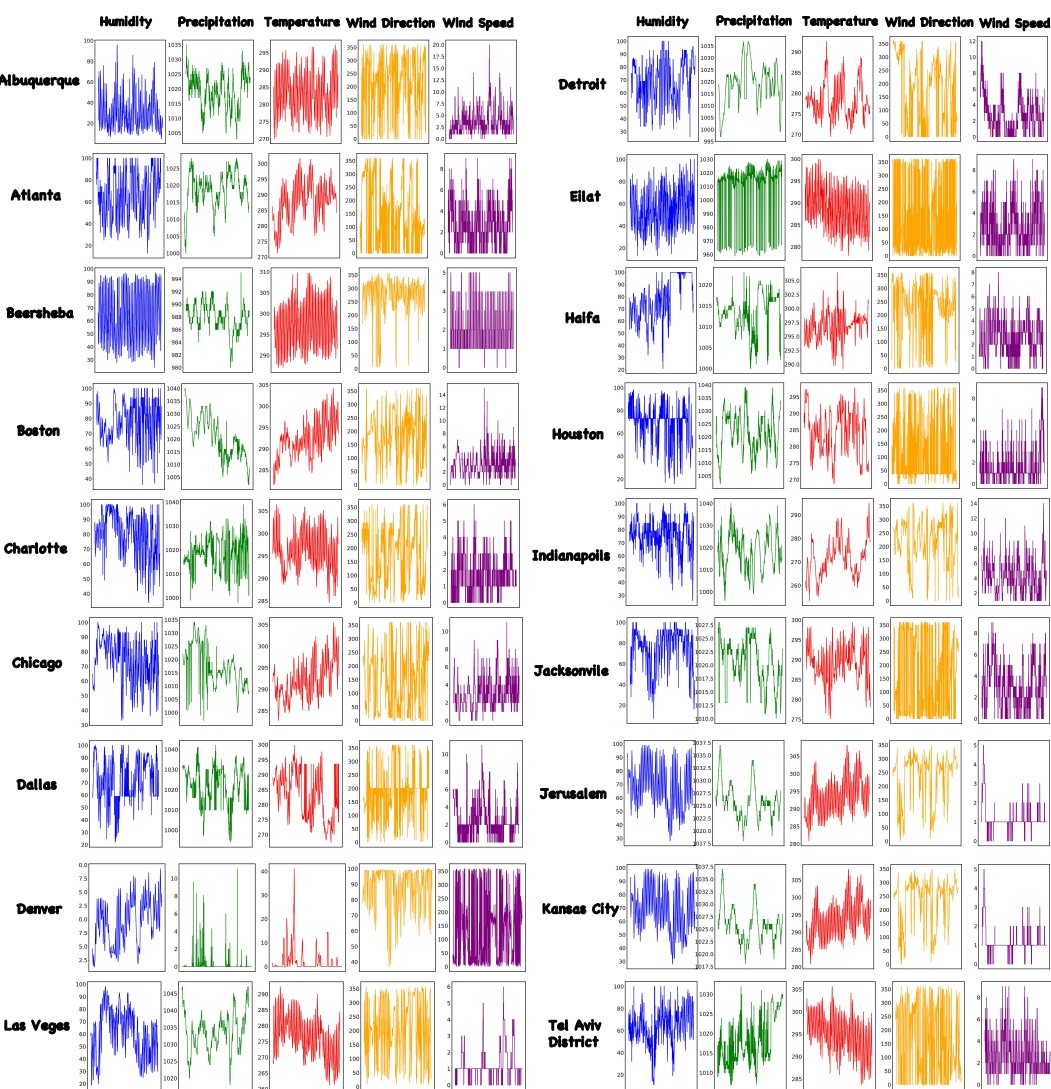

Figure 3: Visualisation of partial variables in ODW2 dataset, where we have selected the first 1,000 time points for presentation. The data distribution from different ground weather stations exhibit significant heterogeneity, and even though the trends of some variables may be similar, there are serious differences in magnitudes. The selected variables are, from left to right, humidity, precipitation, temperature, wind direction, and wind speed.

text in both directions. When used as a baseline for evaluation, we only employ the first 5 layers of the pre-trained BERT.

- **GPT-2.** [15] Developed by OpenAI, GPT-2 is a language model that can generate coherent and diverse text based on a given prompt. In our research, we utilize the first 5 layers of the pre-trained GPT-2-base.

- **LLaMa.** [50] LLaMa stands for Large Language Model Meta AI and is a series of cutting-edge language models with sizes ranging from 7B to 65B parameters. They offer top-notch performance with less computational power and resources. In our research, we utilize the first 4 layers of the 3B LLaMa model.

A brief description of these FL methods tailored to improve communication efficiency is as follows.

- **FedKD**: This parameter-efficient PFL method integrates knowledge distillation within a single client and employs a parameter aggregation strategy using Singular Value Decomposition (SVD). For the purposes of this section, which focuses solely on comparing communication efficiency, we incorporate only the SVD-based client-server communication strategies into LM-WEATHER as a baseline.

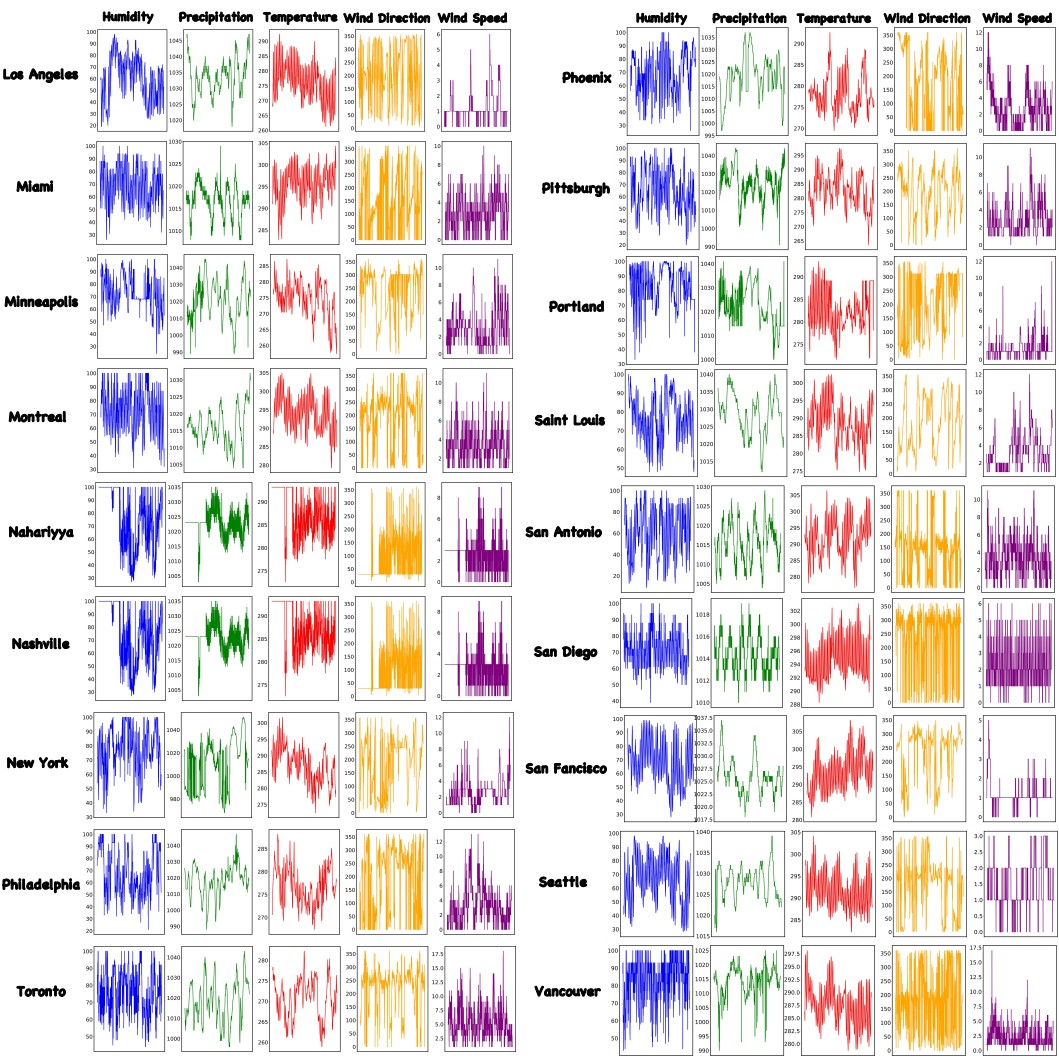

Figure 4: (**Figure 3** continued) Visualisation of partial variables in ODW2 dataset, where we have selected the first 1,000 time points for presentation. The selected variables are, from left to right, humidity, precipitation, temperature, wind direction, and wind speed.

- **FedPer**: This PFL approach maintains a personalized layer while sharing the remaining base layers during communication. This enhances communication efficiency by transmitting only a portion of the parameters.

- **FedBF**: This fine-tuning method enhances parameter efficiency by sharing only the biases of the local model during global aggregation, thereby reducing communication overhead. To integrate this method into LM-WEATHER as a baseline, we adjusted all biases in LM-WEATHER to be unfrozen.

- **FedAP**: A parameter-efficient fine-tuning method in FL, which involves sharing only adapters during global aggregation.

- **PromptFL**: This parameter-efficient FL method enables participants to cooperatively train lightweight prompts without sharing the entire model, significantly accelerating both local training and global aggregation. In our experiments, we treat the adapter generated on clients as the prompt to facilitate the incorporation of this baseline.

## B.3   Task Setups

We evaluate our proposed LM-WEATHER using four distinct on-device weather modeling datasets, each with tailored settings for various tasks. The specific task settings for these datasets are detailed in **Tab. 15**.

Additionally, the specific tasks and scenarios for the on-device weather forecasting/imputation vary by dataset, as outlined in **Tab. 16**.

Table 15: Task setup for different datasets during the evaluation. Note that for the imputation task there are actually no historical observations, but rather they are performed on a single long sequence.

| Dataset | Task | Historical Observation Horizon | Prediction Horizon | Random Masking Ratio |
|---|---|---|---|---|
| ODW1T | Forecasting | 192 | | N |
| | Imputation | Consistent with the prediction horizon | | $\{25\%, 35\%, 50\%\}$ |
| ODW1V | Forecasting | 192 | | N |
| | Imputation | Consistent with the prediction horizon | | $\{25\%, 35\%, 50\%\}$ |
| ODW2T | Forecasting | 192 | $\{96, 192, 336, 720\}$ | N |
| | Imputation | Consistent with the prediction horizon | | $\{25\%, 35\%, 50\%\}$ |
| ODW2V | Forecasting | 192 | | N |
| | Imputation | Consistent with the prediction horizon | | $\{25\%, 35\%, 50\%\}$ |

Table 16: Summary of framework evaluation scenarios for various datasets. **Scenario 1/2/3/4** (in forecasting) refers to multivariate to univariate forecasting, where all historical variables are used to predict a single future variable. **All** represents multivariate to multivariate forecasting, meaning all variables predict all others. The symbol "-" indicates a non-existent scenario for that dataset. **Scenario 1/2/3** (in imputation) indicates different masking ratios for the original weather sequences.

| Dataset | Forecasting | | | | | Imputation | | |
|---|---|---|---|---|---|---|---|---|
| | Scenario 1 | Scenario 2 | Scenario 3 | Scenario 4 | Scenario 5 | Scenario 1 | Scenario 2 | Scenario 3 |
| ODW1T | Temperature | Humidity | Wind Speed | Surface Temperature | All | | | |
| ODW1V | Temperature | - | - | - | All | 25% | 35% | 50% |
| ODW2T | Temperature | Humidity | - | - | All | | | |
| ODW2V | Humidity | - | - | - | All | | | |

## B.4 Implementation

We mainly follow the experimental configurations across all baselines within a unified evaluation pipeline in `https://github.com/thuml/Time-Series-Library` for fair comparison. Specially, we use GPT-2-base as the default backbone model unless state otherwise. All our experiments are repeat five times and we report the averaged results. Our detailed model configurations are in Appendix B.8. All the algorithm implementations and designs in this study are based on Py torch and the algorithms are run on two RTX3090 GPUs 24GB.

## B.5 Technical Details

**Reversible Normalization.**    In time series analysis, statistical properties like mean and variance often shift over time, indicating distributional changes in the data. To address this, we've incorporated Reversible Normalization (RevIn) [51] into our LM-WEATHER. Specifically, we've integrated RevIn into our Task Adapter Generation. This introduces two dynamic factors that adaptively normalize segments of the meteorological variable sequence $\mathcal{X}$, or their decomposed components (Trend $\mathcal{X}_{\text{Trend}}$, Seasonal $\mathcal{X}_{\text{Seasonal}}$, Residual $\mathcal{X}_{\text{Residual}}$), enhancing the accuracy of meteorological variable modeling. Specifically, for the trend component of $\mathcal{X}$, *i.e.,*, $\mathcal{X}_{\text{Trend}}$, its transformed value $\mathcal{X}'_{\text{Trend}}$ can be given by:

$$\mathcal{X}'_{\text{Trend}} = \gamma_T \left( \mathcal{X}_{\text{Trend}} - \frac{\mathbb{E}[\mathcal{X}_{\text{Trend}}]}{\sqrt{\text{Var}\left[\mathcal{X}_{\text{Trend}}\right] + \epsilon_T}} \right) + \beta_T \tag{6}$$

where $\mathbb{E}[\mathcal{X}_{\text{Trend}}]$ and $\text{Var}\left[\mathcal{X}_{\text{Trend}}\right]$ are the instance-specific mean and variance, respectively. $\gamma_T$ and $\beta_T$ are the trainable parameters for this component. This transformation is also applied to both the seasonal and residual components.

**Pre-trained Language Model (PLM).**    In LM-WEATHER, we do not change the main architecture of the PLM, but use the parameter-efficient fine-tuning (PEFT) strategy to avoid large-scale parameter variations to ensure high efficiency on resource-constrained weather devices, and in this way, to achieve more reliable cross-domain knowledge transfer. Specifically, we introduce LoRA in the local PLM, which allows only **1.5%** of the PLM parameters to be trained while the rest remain frozen, as shown in Fig. Note that LoRA is only applied to the query and value of each Attention in the PLM, and the resulting low-rank matrices are used for global sharing between the client and the server.

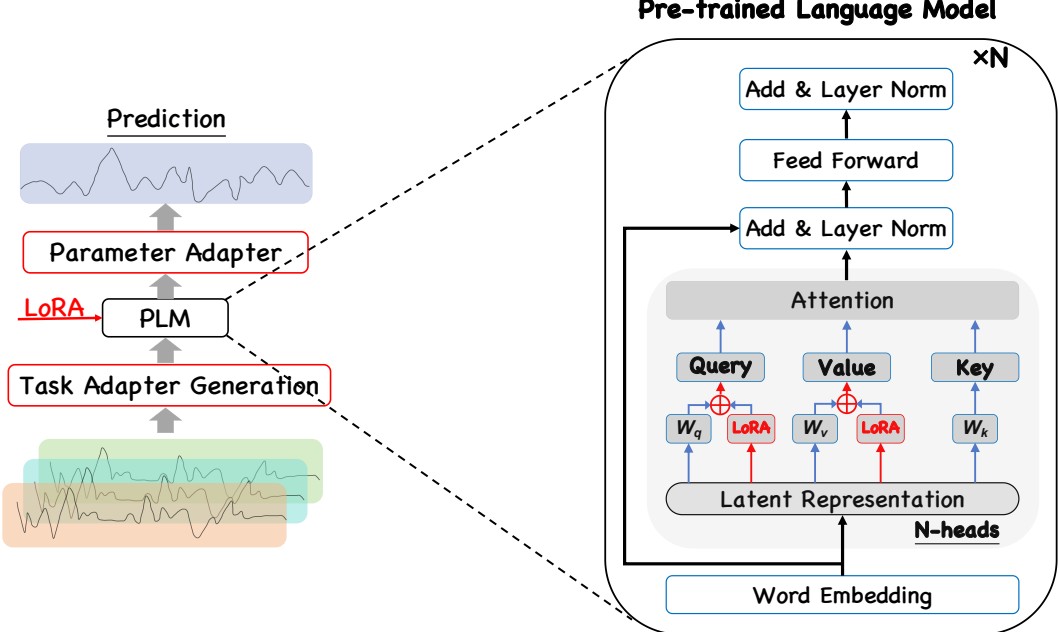

Figure 5: Schematic diagram of the PLM in LM-WEATHER, where we introduced LoRA to the PLM, to achieve more reliable cross-domain knowledge transfer while at the same time ensuring greater efficiency in adapting to low-resource weather devices.

**Low-Rank Adaption (LoRA).** To achieve more reliable cross-domain knowledge transfer (i.e., from natural language to complex weather sequences) while guaranteeing higher efficiency, we introduce LoRA [19], a parameter-efficient fine-tuning method for large language models, into PLM. Specifically, LoRA is applied to the *Query* and *Value* of each Attention layer by creating low-rank matrices for two pre-trained parameters $W_q$ and $W_k$:

$$\text{QUERY} = W_q\mathcal{X} + A_qB_q, \quad \text{VALUE} = W_v\mathcal{X} + A_vB_v, \tag{7}$$

where $\mathcal{X}$ denote the latent representation from input weather sequences through PLM's word embedding layer, $A_q \in \mathbb{R}^{d \times r}$ and $B_q \in \mathbb{R}^{r \times d}$ are low-rank matrices created from $W_q \in \mathbb{R}^{d \times d}$, $A_v \in \mathbb{R}^{d \times r}$ and $B_v \in \mathbb{R}^{r \times d}$ are low-rank matrices created from $W_v \in \mathbb{R}^{d \times d}$, $d$ is the number of dimensions, $r$ is the rank, and $r \ll d$. It's important to note that only the low-rank matrices $A_q, B_q, A_v, B_v$ are trainable; the others remain fixed during training. Initially, $A_q$ and $A_v$ are set with random Gaussian values, and $B_q$ and $B_v$ start as zero at the beginning of training.

**Task Adapter Generation.** The $\mathcal{X}_{\text{Trend}}, \mathcal{X}_{\text{Seasonal}}, \mathcal{X}_{\text{Residual}}$ obtained from decomposition are used to generate *Task Adapter* via an unified generator as **Fig. 1B** that consisting of Token Embedding, Position Embedding, and Temporal Embedding. Specially, we use one-dimensional convolution operation to map each each specific sample $\mathcal{X}^k \in \mathbb{R}^{T \times 1}$ while keeping raw shape to generate TOKEN ADAPTER $\boldsymbol{P}_{\text{TO}} \in \mathbb{R}^{T \times C}$, as

$$\boldsymbol{P}_{\text{TO}}^k = \text{CONV1D}(\mathcal{X}^k), \quad \boldsymbol{P}_{\text{TO}} = \text{CONV1D}(\mathcal{X}) \tag{8}$$

Additionally, we use a trainable lookup table to map each point's explicit position in the entire sequence, to generate POSITION ADAPTER $\boldsymbol{P}_{\text{PO}} \in \mathbb{R}^{T \times C}$, as:

$$\boldsymbol{P}_{\text{PO}} = \boldsymbol{E}(\text{INDEX}(\mathcal{X})), \tag{9}$$

where $\boldsymbol{E}(\cdot)$ is the trainable lookup table, and INDEX($\cdot$) is a function that achieve the indices of each point's locations of weather sequence $\mathcal{X}$. Furthermore, we separately encode different time attributes such as minutes, hours, days, weeks, and months, using trainable parameters to dynamically model complex temporal shifts, to generate TEMPORAL ADAPTER $\boldsymbol{P}_{\text{TE}}$, as

$$\boldsymbol{P}_{\text{TE}} = \sum_{\alpha \in \{\text{mins,hours,days,weeks,months}\}} \boldsymbol{E}_\alpha(\mathcal{X}) \tag{10}$$

where $\alpha$ represents different temporal attributes, $\boldsymbol{E}_\alpha$ denotes the trainable lookup table for each temporal attributes. Finally, for each decomposition components, corresponding generated adapters can be obtained by aggregating Token Adapter $\boldsymbol{P}_{\text{TO}} \in \mathbb{R}^{L \times C}$, Position Adapter $\boldsymbol{P}_{\text{PO}} \in \mathbb{R}^{L \times C}$, and Temporal Adapter $\boldsymbol{P}_{\text{TE}} \in \mathbb{R}^{L \times C}$ as $\boldsymbol{P}_d = \boldsymbol{P}_{\text{TO}}^d + \boldsymbol{P}_{\text{PO}}^d + \boldsymbol{P}_{\text{TE}}^d$, where $d \in \{\text{Trend, Seasonal, Residual}\}$, this means that we can obtain $\boldsymbol{P}_{\text{Trend}}, \boldsymbol{P}_{\text{Seasonal}}, \boldsymbol{P}_{\text{Residual}}$.

## B.6 Theoretical Insights on Personalized Adapter

The effectiveness and superiority of our proposed Personalized Adapter have been demonstrated in sufficient ablation studies (please refer to **Table 6** in the main text). Here, we will discuss theoretical insights that further supports the effectiveness of Personalized Adapter. Personalized Adapter can effectively capture potential pattern in meteorological variable time series, which comprises both the Task Adapter and the FFN-based Parameter Adapter. Specifically, our focus primarily revolves around the Task Adapter active in extracting representations, which including Token/Positional/Temporal Embedding for transforming meteorological variable time series.

Let the weather sequence be $\mathbf{X} = (\mathbf{x}_1, \mathbf{x}_2, \cdots, \mathbf{x}_T)$, where $\mathbf{x}_t \in \mathbb{R}^d$ Is the observed value with $d$ variables at $t$ moment. Let $\mathcal{X}$ represent the function space to which $\mathbf{x}_t$ of a weather sequence belongs, and $\mathcal{Z}$ denote the function space to which the implicit representation $\mathbf{z}_t$ belongs. Token Embedding can be interpreted as a mapping $f_\theta : \mathcal{X} \to \mathcal{Z}$. According to the Kolmogorov-Arnold representation theorem, for any continuous function $f \in C([0,1]^d)$, there exist $2d + 1$ continuous functions $\phi_q \in C([0,1])$ and $\psi_q \in C([0,1])$ such:

$$f(\mathbf{x}) = \sum_{q=0}^{2d} \phi_q(\sum_{p=1}^{d} \psi_q(x_p)),$$

which means Token Embedding can construct a high-dimensional nonlinear mapping from multiple one-dimensional functions, capturing complex patterns within weather sequences. Positional Embedding introduces a vector $\mathbf{p}_t$ for each step, enabling the model to differentiate between observations at different time steps. For any two steps $t_1$ and $t_2$, their position vectors $\mathbf{p}_{t_1}$ and $\mathbf{p}_{t_2}$ satisfy:

$$||\mathbf{p}_{t_1} - \mathbf{p}_{t_2}||_2 = \sqrt{2k(1 - \cos\frac{2\pi(t_1 - t_2)}{10000^{1/k}})}.$$

The growing distance between $t_1$ and $t_2$ with an increasing time gap mirrors the relative positioning of time steps, aiding the model in grasping temporal dependencies. Additionally, the sine-cosine function's periodicity resonates with weather data's cyclical behavior, helping the model to learn from these recurrent patterns.

Finally, consider the role of Temporal Embedding from the view of matrix decomposition. Suppose the temporal matrix $\mathbf{T}$ has a rank of $r$, it can be decomposed as $\mathbf{z}_t \otimes \mathbf{T} = \sum_{i=1}^{r}(\mathbf{z}_t \otimes \mathbf{u}_i)\mathbf{v}_i^T$. Temporal Embedding transforms the original sequence by scaling and rotating it to represent different interaction patterns. Using singular value decomposition, the top $r$ singular vectors distill the core structure of the time-based matrix. This allows Temporal Embedding to intuitively learn a compact representation of weather sequences, highlighting the primary interactions between variables. In optimizing Personalized Adapter within LM-WEATHER, the focus lies solely on Personalized Adapter and the attention layer influenced by LoRA during local updates. As Personalized Adapter undergoes solely local updates while sharing low-rank matrices globally, akin to layer-wise optimization in PLM, the efficacy of its optimization process can be theoretically substantiated by the theoretical analysis provided in [52].

## B.7 Evaluation Metrics

For evaluation metrics, as [34], we utilize the mean absolute error (MAE) and root mean square error (RMSE) for both forecasting and imputation. The calculation of these metrics are as follows:

$$\text{MAE} = \frac{1}{T}\sum_{i=1}^{T}|\mathbf{Y}_i - \hat{\mathbf{Y}}_i|, \qquad \text{RMSE} = \sqrt{\frac{1}{T}\sum_{i=1}^{T}(\mathbf{Y}_i - \hat{\mathbf{Y}}_i)^2}, \qquad (11)$$

where $T$ denotes the number of data points (*i.e.*, prediction horizon in our cases), $\mathbf{Y}_i$ and $\hat{\mathbf{Y}}_i$ are the $i$-th ground truth and prediction where $i \in \{1, ..., T\}$.

## B.8 Model Configurations

The configurations of our LM-WEATHER for different tasks and datasets are summarized in **Tab. 17**. We consistently use the AdamW [53] optimizer in all experiments.

# Appendix C  Theorems and Proofs

**Theorem C.1** (**Decomposition Rationality from Time Series**). *Given a weather time series $\mathcal{X} = \mathcal{X}_{Trend,t} + \mathcal{X}_{Seasonal,t} + \mathcal{X}_{Residual,t}$, $t \in [t_1, t_n]$. Let $\mathbf{E} = \{e_1, e_2, ..., e_n\}$ denotes a set of orthogonal bases. Lets $\mathbf{E}_{Seasonal} \subseteq \mathbf{E}$ denote the subset of $\mathbf{E}$ on which $\mathcal{X}_{Seasonal,t}$ has non-zero eigenvalues and $\mathbf{E}_{Trend} \subseteq \mathbf{E}$ denote the subset of $\mathbf{E}$ on which $\mathcal{X}_{Trend,t}$ has non-zero eigenvalues. If $\mathcal{X}_{Trend,t}$ and $\mathcal{X}_{Seasonal,t}$ are not orthogonal, i.e., $\sum_{i=1}^{n} \mathcal{X}_{Trend,t}^i \mathcal{X}_{Seasonal,t}^i \neq 0$, then $\mathbf{E}_{Trend} \bigcap \mathbf{E}_{Seasonal} \neq 0$, i.e., $\mathbf{E}$ can not disentangle the two signals onto two disjoint set of bases.*

Table 17: An overview of the experimental configuration for LM-WEATHER. LR is the initial learning rate, (FS) denotes the few-shot learning setting.

| Task-Dataset / Configuration | Model Hyperparameter | | | | Traning Process | | | | | |
| --- | --- | --- | --- | --- | --- | --- | --- | --- | --- | --- |
| | Backbone (PLM) Layers | Input Length | Patch Dim | Heads | LR | Loss | Batch Size | Local Epochs | Communication Round | participation rate |
| Forecasting - ODW1T | 5 | 192 | 16 | 8 | 0.005 | MSE | 128 | 20 | 50 | 0.1 |
| Forecasting - ODW1V | 5 | 192 | 16 | 8 | 0.005 | MSE | 128 | 20 | 50 | 0.1 |
| Forecasting - ODW2T | 5 | 192 | 16 | 8 | 0.005 | MSE | 256 | 20 | 50 | 0.1 |
| Forecasting - ODW2V | 5 | 192 | 16 | 8 | 0.005 | MSE | 256 | 20 | 50 | 0.1 |
| Imputation - ODW1T | 5 | 96, 192, 336, 720 | 16 | 8 | 0.005 | MSE | 128 | 20 | 50 | 0.1 |
| Imputation - ODW1V | 5 | 96, 192, 336, 720 | 16 | 8 | 0.005 | MSE | 128 | 20 | 50 | 0.1 |
| Imputation - ODW2T | 5 | 96, 192, 336, 720 | 16 | 8 | 0.005 | MSE | 256 | 20 | 50 | 0.1 |
| Imputation - ODW2V | 5 | 96, 192, 336, 720 | 16 | 8 | 0.005 | MSE | 256 | 20 | 50 | 0.1 |
| Forecasting - ODW1T (FS) | 5 | 192 | 16 | 8 | 0.005 | MSE | 128 | 20 | 50 | 0.1 |
| Forecasting - ODW1V (FS) | 5 | 192 | 16 | 8 | 0.005 | MSE | 128 | 20 | 50 | 0.1 |
| Forecasting - ODW2T (FS) | 5 | 192 | 16 | 8 | 0.005 | MSE | 256 | 20 | 50 | 0.1 |
| Forecasting - ODW2V (FS) | 5 | 192 | 16 | 8 | 0.005 | MSE | 48 | 20 | 50 | 0.1 |
| Imputation - ODW1T (FS) | 5 | 96, 192, 336, 720 | 16 | 8 | 0.005 | MSE | 128 | 20 | 50 | 0.1 |
| Imputation - ODW1V (FS) | 5 | 96, 192, 336, 720 | 16 | 8 | 0.005 | MSE | 128 | 20 | 50 | 0.1 |
| Imputation - ODW2T (FS) | 5 | 96, 192, 336, 720 | 16 | 8 | 0.005 | MSE | 256 | 20 | 50 | 0.1 |
| Imputation - ODW2V (FS) | 5 | 96, 192, 336, 720 | 16 | 8 | 0.005 | MSE | 48 | 20 | 50 | 0.1 |

*Proof.* We decompose $\mathcal{X}_{\text{Seasonal},t}$ and $\mathcal{X}_{\text{Trend},t}$ onto $\boldsymbol{E}$ and acquire that $\mathcal{X}_{\text{Seasonal},t} = \sum a_i e_i$ and $\mathcal{X}_{\text{Trend},t} = \sum b_i e_i$. Then it is obvious that $e_i \in \mathcal{X}_{\text{Seasonal}} \Leftrightarrow a_i \neq 0$ and $e_i \in \mathcal{X}_{\text{Trend}} \Leftrightarrow b_i \neq 0$. Now, let us consider the inner product of $\mathcal{X}_{\text{Seasonal},t}$ and $\mathcal{X}_{\text{Trend},t}$:

$$\sum_{i=1}^{n} \mathcal{X}_{\text{Trend},t}^{i} \mathcal{X}_{\text{Seasonal},t}^{i} = \mathcal{X}_{\text{Trend},t} \mathcal{X}_{\text{Seasonal},t}$$
$$= (\sum a_i e_i) \cdot (\sum b_i e_i) = \sum_{i,j} a_i b_j e_i e_j \tag{12}$$

Note that $\sum_{i=1}^{n} \mathcal{X}_{\text{Trend},t}^{i} \mathcal{X}_{\text{Seasonal},t}^{i} = 0$. Thus, there must be at least one $i$ such that $a_i \neq 0$ and $b_i \neq 0$. Thus. $e_i \in \boldsymbol{E}_{\text{Seasonal}}$ and $e_i \in \boldsymbol{E}_{\text{Trend}}$, in other words, $\boldsymbol{E}_{\text{Trend}} \cap \boldsymbol{E}_{\text{Seasonal}} \neq 0$. The theorem demonstrates that if $\mathcal{X}_{\text{Trend},t}$ and $\mathcal{X}_{\text{Seasonal},t}$ are not orthogonal, orthogonal bases that separate $\mathcal{X}_{\text{Trend},t}$ and $\mathcal{X}_{\text{Seasonal},t}$ into two distinct sets cannot exist. Typically, periodic and non-periodic signals are not orthogonal because the periodic signal has a discrete spectrum, while the non-periodic signal has a continuous one, leading to potential overlaps at non-zero frequencies. Principal Component Analysis (PCA) seeks to find orthogonal bases in data, but it cannot split these two signals into separate bases. Citing Theorem 1 from [17], we understand that self-attentive mechanisms in pre-trained large models function similarly to PCA. Thus, without manual intervention, the self-attentive mechanism is unable to automatically divide a time series into trend and seasonal components. □

**Theorem C.2** (Exchange Low-Rank Matrices Ensures Privacy: Parameter Interaction Perspective). *Given a on-device weather modeling framework based on federated learning that global optimization object is $\boldsymbol{F}(\theta) = \sum_{n}^{i=1} p_i f(\{D_i\}; \theta)$, where $f(x; \theta)$ is the loss function of $i$-th client, $\{D_i\}$ is dataset of $i$-th client, and $p_i$ and $\theta$ denote the data distribution weight of client $i$ and the model parameters, respectively. Given that the parameters $\theta$ of the PLM $\mathcal{M}_\theta$ broadcasted by the server consist of two parts: a frozen part $\mathcal{M}_{\theta,f}$ and a trainable part $\mathcal{M}_{\theta,t}$, interacting only the low-rank matrix parameters $\mathcal{M}_{\theta,l} \subset \mathcal{M}_{\theta,t}$ is a subset of trainable part $\mathcal{M}_{\theta,t}$ during each round ensures privacy.*

*Proof.* We assume that $f(x; \theta)$ is a convex function with respect to $\theta$, i.e., for any $\theta_1$ and $\theta_2$ and $\lambda \in [0, 1]$, we have
$$f(x; \lambda\theta_1 + (1 - \lambda)\theta_2) \leq \lambda f(x; \theta_1) + (1 - \lambda)f(x; \theta_2). \tag{13}$$

Since only low-rank matrices parameter $\mathcal{M}_{\theta,l}$ parameterized by $\theta_l$ is exchanged, we can convert $\theta$ to $\theta' = [\theta'_l, \theta_o]$, where $\theta'_l$ is the embedding parameter after the server update. Since we only update on $\theta_l$, $\theta_o$ remains unchanged. Thus, data privacy can be ensured, as $\theta_o$ contains parameters that reveal user-specific information. Furthermore, the low-rank matrices applied to the PLM $\mathcal{M}_\theta$ using LoRA are initialized with a random Gaussian distribution and all-zero values, respectively, before training. This global information sharing approach also helps to enhance privacy. □

**Theorem C.3** (Exchange Low-Rank Matrices Ensures Privacy: Model Indistinguishability Perspective). *Our LM-Weather can hide key features of local model when sharing low-rank parameters, even external attacker gains access to a shared low-rank update, it is difficult to reconstruct or differentiate between the original models of different participants. Support client $i$ and client $j$ get two different local model due to updated on heterogeneity weather time series, as $\mathcal{M}_i$ and $\mathcal{M}_j$ parametered by $\theta_i$ and $\theta_j$, $L(\mathcal{M})$ denotes the low-rank matrix generated by model $\mathcal{M}$, where $B \in \mathbb{R}^{d \times r}$ and $A \in \mathbb{R}^{r \times d}$, and $r \ll d$, and define $\boldsymbol{Adv}$ as the adversary. Conditions exist such that for any polynomial-time attacker $\boldsymbol{Adv}$, $|Pr[\boldsymbol{Adv}(L(\mathcal{M}_i)) = 1] - Pr[\boldsymbol{Adv}(L(\mathcal{M}_j)) = 1]| \leq \epsilon$ holds,*

*where $\epsilon$ is a small positive number, which implies that even if the attacker acquires a shared low-rank update, it is difficult to reconstruct or distinguish between the original models of the different participants.*

*Proof.* The goal of LoRA is to find the low-rank approximation: $min||\theta - \theta_0 - BA||_F$, where $\theta_0$ is the initial weight, $||\cdot||_F$ is the Frobenius paradigm. Consider two models $\mathcal{M}_i$ and $\mathcal{M}_j$, whose weights differ by $\Delta\theta = \theta_i - \theta_j$. The corresponding LoRA matrix is:

$$L_i = B_i A_i \approx \Delta\theta_i = \theta_i - \theta_0,$$
$$L_j = B_j A_j \approx \Delta\theta_j = \theta_j - \theta_0. \tag{14}$$

According to matrix approximation theory, for the best approximation with rank $r$, the upper bound on the error is:

$$||\Delta\theta_i - L_i||F \le \sigma_{r+1}(\Delta\theta_i) \tag{15}$$

where $\sigma_{r+1}(\Delta\theta_i)$ is the r+1-st singular value of $\Delta\theta_i$. By Johnson-Lindenstrauss Lemma [54], for any $\epsilon > 0$, there exists a mapping $f : \mathbb{R}^d \to \mathbb{R}^k$, where $k = O(log(n)/\epsilon^2)$, such that for any $x, y \in \mathbb{R}^d$. In our case, LoRA can be regarded as such a degenerate mapping. Assume $||\Delta\theta_i - \Delta\theta_j||_F \le \delta$ and according to the trigonometric inequality:

$$||L_i - L_j||_F \le ||L_i - \Delta\theta_i||_F + ||\Delta\theta_i - \Delta\theta_j||F + ||\Delta\theta_j - L_j||F \le \sigma_{r+1}(\Delta\theta_i) + \delta + \sigma_{r+1}(\Delta\theta_j), \tag{16}$$

Let $\varepsilon' = \sigma_{r+1}(\Delta\theta_i) + \sigma_{r+1}(\Delta\theta_j)$, we get:

$$||L_i - L_j||_F \le \delta + \varepsilon'. \tag{17}$$

For any polynomial-time attacker Adv,its ability to distinguish between $L_i$ and $L_j$ is restricted to the difference in their Frobenius paradigms. We can define a function $f$ such that:

$$|Pr[\texttt{Adv}(L(\mathcal{M}_i)) = 1] - Pr[\texttt{Adv}(L(\mathcal{M}_j)) = 1]| \le f(||L_i - L_j||_F) \tag{18}$$

where $f$ is a monotonically increasing function that represents the attacker's capability with respect to the matrix difference. Combining Eq. 16 and Eq. 17:

$$|Pr[\texttt{Adv}(L(\mathcal{M}_i)) = 1] - Pr[\texttt{Adv}(L(\mathcal{M}_j)) = 1]| \le \epsilon \tag{19}$$

when $\delta$ and $\varepsilon'$ are small enough, we can make sure that the right-hand side is smaller than the intended $\varepsilon$. This means that an attacker cannot reverse-engineer personalised local parameters and data to ensure privacy through the low-rank matrix of communication across clients. □

# Appendix D   Additional Finding & Experiment & Discussion

In this section, we explore and discuss potential research findings and questions for our LM-WEATHER via conducting additional experiments. These potential research questions are as follows:

- **RQ1.** How does LM-WEATHER compare to Personalized Federated Learning (PFL) baselines in terms of trade-offs in personalization and global model performance?
- **RQ2.** How does LM-WEATHER perform compared to centralized and local-only training modes?
- **RQ3.** How does the pre-trained language model contribute in LM-WEATHER?
- **RQ4.** What is the resource utilization and training & inference cost of LM-WEATHER?
- **RQ5.** Can LM-WEATHER be used for other tasks?

## D.1   Trade-offs in Personalization and Global Model Performance (RQ1)

Our LM-WEATHER builds on the assumption that the foundation model already exists, treating pre-trained language models (PLMs) as such and broadcasting it to each client to achieve local updates. Our aim is to employ device information-specific (e.g., geographic/atmospheric patterns) adapter, to promote the local PLM in achieving cross-domain knowledge transfer from language to meteorological sequences. This approach yields highly customized models for individual devices while achieving global knowledge to avoid data silo, thereby supporting diverse analyses of heterogeneous weather data. Alternative PFL methods do not match the efficiency and flexibility of our personalized adapter in this context, making them less suitable. By incorporating PFL baselines (Per-FedAvg [43], APPLE [55], FedPer [56], and FedALA [57]), we provide quantitative results that substantiate our claims, experiment setting is consistent with the manuscript on ODW1T.

**PFL Baseline.** A brief description of PFL baselines used in this section of the experiment is as follows.

- **Per-FedAvg**: Allowing for personalized model updates for each client by adding client-specific parameters to the global model and optimizing them during FL training.
- **APPLE**: Tackling statistical heterogeneity in FL by automatically capturing information required by clients from global models using adaptive local aggregation methods.
- **FedPer**: Dividing the model into a base layer and a personalization layer, only the base layer is uploaded during aggregation while keeping the personalization layer to combat statistical heterogeneity.
- **FedALA**: Tackling statistical heterogeneity in FL by automatically capturing information required by clients from global models using adaptive local aggregation methods.

Table 18: Comparison on personalized performance between our LM-WEATHER and PFL baselines under forecasting (multivariate-multivariate) and imputation (50% masking rate), where *Avg.* denotes the average performance of four periods [96, 192, 336, 720], **Bold** means the best.

| Task / Method | LM-WEATHER (Ours) | Per-FedAvg | APPLE | FedALA | FedPer |
|---|---|---|---|---|---|
| Forecasting (*Avg.*) | **45.4/74.6** | 48.6/76.7 | 51.7/79.0 | 50.4/80.0 | 52.1/79.0 |
| Imputation (*Avg.*) | **23.1/42.4** | 27.7/50.1 | 26.5/49.6 | 34.4/57.2 | 30.2/54.0 |
| Forecasting (*Avg.*, 5% Few-Shot) | **89.7/96.7** | 101.4/197.3 | 99.6/186.2 | 117.3/214.0 | 104.9/202.8 |
| Imputation (*Avg.*, 5% Few-Shot) | **62.3/124.4** | 89.2/177.8 | 92.1/199.2 | 82.4/153.7 | 84.9/159.7 |

**Personalized Performance Comparison.** The performance quantification of our LM-WEATHER and PFL baselines under personalized performance for different tasks and scenarios is shown in Table 18. Our LM-WEATHER outperform other PFL baselines across different tasks (forecasting/imputation) and scenarios (regular/few-shot learning) by a wide margin. This supports our finding that in the scenario of on-device weather variable modeling, PFL methods is not appropriate.

**Global Model Performance Comparison.** The comparison on global model performance across client between our LM-WEATHER and PFL baselines are shown in Table 18. Our LM-WEATHER outperforms PFL baselines in terms of global model performance, as demonstrated by the fact that its global model performs more stable across client with heterogeneous data.

Table 19: Comparison on global model performance across client between LM-Weather and PFL baselines on multivariate-multivariate forecasting tasks (OWD1T dataset, MAE/RMSE report), Red denotes the original LM-WEATHER's performance, and **Bold** means the best among global performance.

| Client ID | LM-WEATHER (Ours) | LM-WEATHER (Global) | Per-FedAvg (Global) | APPLE (Global) | FedALA (Global) | FedPer (Global) |
|---|---|---|---|---|---|---|
| 1 | 44.8/73.9 | 42.5/69.4 | 50.3/78.2 | 53.4/77.1 | 51.7/76.3 | 52.8/82.5 |
| 2 | 46.1/75.4 | 56.8/84.7 | 61.8/90.1 | 64.7/88.6 | 63.2/88.1 | 68.3/99.8 |
| 3 | 45.2/74.3 | 49.3/75.2 | 54.2/82.6 | 57.2/81.4 | 55.9/80.7 | 57.1/87.2 |
| 4 | 47.1/79.0 | 61.2/89.6 | 64.9/93.4 | 67.8/92.1 | 66.5/91.6 | 73.6/105.3 |
| 5 | 43.1/70.6 | 45.7/72.1 | 52.1/80.3 | 55.1/78.9 | 53.6/78.2 | 54.9/84.6 |
| 6 | 43.3/73.7 | 59.1/87.3 | 63.4/91.7 | 66.3/90.3 | 64.8/89.5 | 70.2/102.1 |
| 7 | 44.8/74.1 | 47.6/73.8 | 55.9/84.5 | 58.9/83.2 | 57.3/82.4 | 59.5/89.9 |
| 8 | 48.1/77.2 | 53.4/80.5 | 59.6/88.5 | 62.6/87.4 | 61.1//86.6 | 65.7/96.4 |
| 9 | 42.6/71.7 | 44.1/70.7 | 50.9/78.8 | 53.8/77.6 | 52.3/76.9 | 53.6/83.3 |
| 10 | 46.0/75.3 | 57.9/86.2 | 64.2/92.6 | 67.1/91.5 | 65.7/90.6 | 71.9/103.7 |
| 11 | 45.3/74.4 | 50.2/76.3 | 54.7/83.2 | 58.0/82.3 | 56.6/81.3 | 60.8/91.6 |
| 12 | 45.6/74.9 | 55.7/82.9 | 60.8/89.4 | 63.9/88.5 | 62.5/87.7 | 67.4/98.2 |
| 13 | 48.7/77.8 | 43.3/68.5 | 49.6/77.5 | 56.3/76.3 | 50.9/75.5 | 51.7/81.1 |
| 14 | 49.2/75.5 | 60.5/88.4 | 65.7/94.2 | 68.7/92.8 | 67.4/92.4 | 72.5/104.5 |
| 15 | 41.1/73.2 | 48.7/74.4 | 53.4/81.9 | 56.3/80.2 | 54.5/79.4 | 58.2/88.5 |
| Total (Avg.) | 45.4/74.6 | **51.2/78.7** | 57.0/84.9 | 59.9/83.2 | 58.4/82.8 | 61.4/92.7 |

**Personalization and Global Model Performance Trade-offs.** We consider the trade-off between personalization performance and global model performance for our LM-WEATHER and PFL baselines, the results are shown in Table 20. Compared with PFL baselines, our LM-WEATHER maintains the best trade-off between personalization performance and global model performance, *i.e.*, the personalization performance does not significantly exceed the global model performance while the performance far exceeds PFL methods, which means that the LM-WEATHER can be flexibly applied to different practical scenarios, including personalised analysis of regional weather trends as well as comprehensive analysis of weather trends over large-scale regions. This means that LM-WEATHER can be flexibly applied to different practice scenarios, including the personalised analysis of regional weather trends and the comprehensive analysis of weather trends over large scale areas.

Table 20: Comparison of LM-Weather between personalized performance and global model performance, results are obtained on the multivariate-multivariate forecasting task on OWD1T (MAE/RMSE report), **Bold** means the best, ↑ represents the improvement (gap) in the personalization performance of the method relative to the global model performance.

| Method | Personalized Performance | Global Model Performance | Ave. Variation (Personalized vs. Global) |
|---|---|---|---|
| Per-FedAvg | 48.6/76.7 | 57.0/84.9 | ↑ 13.99% |
| APPLE | 51.7/79.0 | 59.9/83.2 | ↑ 10.58% |
| FedALA | 50.4/80.0 | 58.4/82.8 | ↑ 9.68% |
| FedPer | 52.1/79.0 | 61.4/92.7 | ↑ 17.59% |
| LM-WEATHER (Ours) | **45.4/74.6** | **51.2/78.7** | ↑ **9.16%** |

**Performance and Adapter Updating Trade-offs.** Furthermore, we investigated the effect of varying the number of local update rounds in adapters across clients on the performance of LM-WEATHER regrading personalization. The results are presented in **Table 21**. We observed that increasing the local update rounds from the default five to fifteen leads to smoother and enhanced personalization performance across heterogeneous clients. However, this increase in local update rounds also incurs additional computational and communication costs, which, in our assessment, do not justify the modest performance improvements.

Table 21: Performance of each client under the multivariate-multivariate forecasting task on ODWT1 with different adapter local update epoch (MAE/RMSE report), where $E = 5/10/15$ represent the 5, 10, and 15 local training rounds, respectively.

| Client ID | LM-WEATHER ($E = 5$) | LM-WEATHER ($E = 10$) | LM-WEATHER ($E = 15$) |
|---|---|---|---|
| 1 | 44.8/73.9 | 41.3/70.5 | 41.6/69.7 |
| 2 | 46.1/75.4 | 47.9/75.8 | 46.3/74.5 |
| 3 | 45.2/74.3 | 43.5/72.1 | 43.1/71.1 |
| 4 | 47.1/79.0 | 46.2/74.6 | 45.7/73.6 |
| 5 | 43.1/70.6 | 42.7/71.3 | 42.4/70.4 |
| 6 | 43.3/73.7 | 45.5/74.2 | 44.9/73.0 |
| 7 | 44.8/74.1 | 44.1/73.0 | 43.5/71.7 |
| 8 | 48.1/77.2 | 48.6/76.3 | 47.2/75.2 |
| 9 | 42.6/71.7 | 40.9/70.1 | 41.2/69.3 |
| 10 | 46.0/75.3 | 46.8/75.4 | 45.3/74.1 |
| 11 | 45.3/74.4 | 43.2/71.8 | 42.8/70.8 |
| 12 | 45.6/74.9 | 45.7/74.5 | 44.5/72.7 |
| 13 | 48.7/77.8 | 42.1/70.9 | 41.9/70.1 |
| 14 | 49.2/75.5 | 47.3/75.1 | 46.8/74.9 |
| 15 | 41.1/73.2 | 44.8/73.4 | 43.8/72.3 |
| Total (Avg.) | 45.4/74.6 | 44.6/73.2 | 44.0/72.2 |

## D.2 Centralised and Local-only Training (RQ2)

The ordinary centralised training strategy (all data were aggregated into a single server) exhibits learning efficiency that an ordinary distributed learning strategy. The ultimate goal of FL is to achieve performance close to that of centralised training and to ensure privacy across data sources. **Table 22** illustrates that our LM-WEATHER achieves comparable effectiveness to Non-FL (centralised) training, with only a 2.04% disparity. Compared to LM-WEATHER-Local, which lacks interaction between devices, LM-WEATHER performs better due to overcoming data silos.

## D.3 Contributions of Pre-trained Language Model in LM-WEATHER (RQ3)

Our LM-WEATHER significantly outperforms time series-specific models trained from scratch under centralised setup. Centralised training aims to acquire an excellent pre-trained model, where PLMs possess inherent advantages due to their prior sequence modeling capabilities. Moreover, various parameter-efficient fine-tuning (PEFT) strategies enable PLMs to adapt to new domain knowledge cost-effectively. FL-based aggregation facilitates a stable on-device fine-tuning process, with LM-Weather enabling highly customized on-device fine-tuning of PLMs with greater efficiency. This highlights the substantial contribution of PLMs in this task.

Table 22: Comparison of LM-WEATHER's multivariate-multivariate performance in the FL and the Non-FL (centralised) setups, LM-WEATHER-Local is the setting in which LM-WEATHER is trained locally at each device without communication, and disparity is the difference in performance relative to Non-FL.

| Dataset | Length | Non-FL (centralised) | LM-WEATHER (Ours) | LM-WEATHER-Ave | LM-WEATHER-Local |
|---|---|---|---|---|---|
| ODW1T | 96 | 41.7/70.3 | 42.3/71.1 | 44.1/74.8 | 45.9/72.9 |
| | 192 | 43.5/71.6 | 44.4/73.6 | 46.3/77.5 | 46.7/74.3 |
| | 336 | 45.2/72.9 | 45.8/75.2 | 47.9/79.3 | 48.2/77.0 |
| | 720 | 46.8/73.6 | 49.2/78.5 | 51.8/83.0 | 50.5/80.5 |
| | Avg. | 44.3/72.1 | 45.4/74.6 | 47.5/78.7 | 47.8/76.2 |
| ODW1V | 96 | 42.3/68.7 | 42.3/69.6 | 42.7/69.5 | 44.5/68.6 |
| | 192 | 43.9/69.9 | 44.4/71.7 | 45.5/72.6 | 46.1/70.4 |
| | 336 | 45.4/71.2 | 46.0/72.4 | 47.2/74.3 | 48.8/73.2 |
| | 720 | 46.8/72.6 | 49.7/74.0 | 51.2/78.2 | 53.2/79.4 |
| | Avg. | 44.6/70.6 | 45.6/71.9 | 46.6/73.6 | 48.2/72.9 |
| Communication Param. # | | None | 0.38 M | 10.38 M | Not applicable |
| Disparity | | 0 | ↓ **2.04%** | ↓ 4.37% | ↓ 5.67% |

Table 23: Comparison between fine-tuning PLM with Adapter (LM-WEATHER) and training from scratch using non-PLM architecture (Pyraformer, Reformer, PatchTST, DLinear, and LightTS) on multivariate-multivariate forecasting tasks (MAE/RMSE report), **Bold** means the best.

| Dataset | Length | LM-WEATHER (Ours) | Pyraformer | Reformer | PatchTST | Dlinear | LightTS |
|---|---|---|---|---|---|---|---|
| ODW1T | 96 | 43.0/75.2 | 66.0/84.2 | 68.2/89.6 | 44.7/76.1 | 48.6/77.6 | 52.1/83.3 |
| | 192 | 44.7/78.4 | 68.7/87.4 | 69.3/89.9 | 46.5/78.4 | 51.0/79.8 | 57.5/87.0 |
| | 336 | 47.2/80.9 | 71.3/90.9 | 71.4/91.7 | 48.2/80.3 | 52.3/84.1 | 62.3/94.2 |
| | 720 | 50.4/83.8 | 76.8/92.3 | 74.5/93.9 | 54.2/84.5 | 55.4/86.3 | 69.5/98.4 |
| | Avg. | **46.3/79.6** | 70.7/88.7 | 70.9/91.3 | 48.4/79.8 | 51.8/82.0 | 60.4/90.7 |
| ODW1V | 96 | 45.4/71.3 | 60.4/69.1 | 45.7/71.8 | 47.2/73.4 | 48.5/74.7 | 50.1/75.3 |
| | 192 | 46.9/72.9 | 61.9/73.2 | 51.9/75.5 | 48.4/74.2 | 49.4/76.6 | 52.7/78.8 |
| | 336 | 49.0/75.5 | 64.4/76.9 | 53.4/76.9 | 48.9/76.7 | 53.7/78.3 | 54.0/80.4 |
| | 720 | 53.7/79.1 | 68.2/82.7 | 56.5/84.9 | 54.1/77.4 | 57.2/82.2 | 57.7/84.8 |
| | Avg. | **48.8/74.9** | 63.7/75.5 | 51.9/77.3 | 49.7/75.4 | 52.2/78.0 | 53.6/79.8 |
| 1$^{st}$ Count | | **16** | 1 | 0 | 3 | 0 | 0 |

## D.4 No Free Lunch in Performance Improvement (RQ4)

The remarkable capabilities of cutting-edge DL models across various domains and tasks, such as LLMs, and VLMs, can be attributed to their extensive parameters and training on large datasets. Currently, a perfect balance among performance, model size, and cost does not exist. Despite numerous studies focusing on reducing training and inference costs while maintaining superior performance, there is no one-size-fits-all solution. This is also true for our LM-WEATHER, which demonstrates exceptional performance across diverse tasks and scales on real-world datasets with significant heterogeneity, significantly outperforming comparable DL methods. In this context, we analyze the costs associated with training and inference for LM-WEATHER and its baselines, exploring and discussing the trade-offs between cost-effectiveness and performance in practical applications.

Table 24: Comparison of training/inference costs based on ODW1T with $N = 192$ under multivariate-multivariate forecasting tasks (MAE/RMSE report), where **Bold** denotes the best, 'Comm.' and 'Perf.' denote communication and performance, respectively.

| Method | Ave. Training Time (per round) | Inference Time (per client) | Training Memory (per client) | Inference Memory (per client) | Comm. Time | Perf. |
|---|---|---|---|---|---|---|
| FL-DLinear | **4 s** | **1 s** | **7.93 MB** | **4.21 MB** | 1.44 s | 52.3/81.8 |
| FL-LightTS | 4 s | 2 s | 30.94 MB | 17.24 MB | 9.93 s | 59.5/90.6 |
| FL-PatchTST | 124 s | 10 s | 1260.01 MB | 660.02 MB | 59.62 s | 47.3/79.8 |
| FL-Transformer | 121 s | 11.0 s | 1700.95MB | 841.09 MB | 36.44 s | 52.1/84.0 |
| FL-iTransformer | 9 s | 1.6 s | 705.36 MB | 372.28 MB | 22.40 s | 51.8/84.3 |
| FL-Informer | 110 s | 10 s | 1890.00 MB | 897.21 MB | 42.81 s | 52.9/84.6 |
| FL-Reformer | 145 s | 17.7 s | 786.98 MB | 421.23 MB | 16.72 s | 75.1/98.3 |
| FL-Pyraformer | 80 s | 8.1 s | 1880.41 MB | 950.77 MB | 119.66 s | 70.0/90.9 |
| FL-GPT4TS | 108 s | 21.2 s | 3640.11 MB | 1900.98 MB | 10.03 s | 48.6/81.3 |
| LM-WEATHER | 91 s | 14 s | 3014.82 MB | 1500.81 MB | **0.29 s** | **44.4/73.6** |

The quantification and comparison of computational costs against LM-WEATHER and baseline are shown in **Table 24**. We discuss this results from two perspectives as follow.

**Communication and Performance.** LM-WEATHER outperforms baseline in these two key metrics, which is critical for practical meteorological variable modeling and analysis, a bandwidth-sensitive and high accuracy demanding application.

**Trade-offs between Resource Consumption and Performance.** In terms of training and inference time and memory usage, LM-WEATHER is less efficient than lightweight baselines such as FL-DLinear, LightTS, and iTransformer, due to its use of a Pretrained Language Model (PLM) as a backbone. Although LM-WEATHER demands more resources, the trade-off is justified by its cost-effective performance gains. Its slightly increased memory requirements for training and inference are manageable on most devices. In the context of weather analysis, where precision is critical, prioritizing performance improvements over minimal resource consumption is essential. Additionally, LM-WEATHER capitalizes on the knowledge-rich PLM and requires only minimal, low-cost fine-tuning on devices to achieve superior performance. This strategy not only enhances performance but also reduces the frequency of future model updates, thereby lowering long-term costs compared to developing a baseline model from scratch.

Table 25: Comparison between LM-Weather and baseline in terms of model size on the device and performance of forecasting (multivariate-to-multivariate), and imputation (50% masking rate) on ODW1T (MAE/RMSE report), where **Bold** and Underline denote the best and the second best.

| Method | Size | Forecasting | Imputation |
|---|---|---|---|
| FL-DLinear | **0.28 M** | 53.3/82.8 | 28.5/49.9 |
| FL-LightTS | 1.10 M | 62.7/93.4 | 26.1/45.7 |
| FL-PatchTST | 37.61 M | 48.6/81.0 | 45.4/73.5 |
| FL-Transformer | 45.55 M | 52.8/84.7 | 57.6/82.3 |
| FL-iTransformer | 25.19 M | 53.7/63.7 | 27.6/48.2 |
| FL-Informer | 52.31 M | 53.4/85.2 | 61.4/85.9 |
| FL-Reformer | 45.59 M | 78.2/98.7 | 69.8/92.5 |
| FL-GPT4TS | 321.7 M | 49.9/82.5 | 25.1/46.2 |
| LM-WEATHER | 304.1 M | **45.4/74.6** | **23.1/42.4** |

In addition, we further provide a comparison of model sizes and performance between LM-WEATHER and baseline, as shown in **Table 25**. The difference in model size between our LM-WEATHER and baseline can be deemed acceptable for the following reasons.

**Trade-offs between Performance and Size.** While LM-WEATHER may not be as compact in terms of model size or resource efficiency as lightweight baselines, it offers significant advantages in various analysis tasks. Its high performance is particularly valuable in practical applications. Moreover, with a model size of 304.19 M, LM-WEATHER is still accessible for devices with limited resources. This contrasts sharply with many large foundation models, which typically comprise several hundred million parameters. The trade-off between performance and size is justified, especially considering the critical nature of accurate weather data analysis.

**Efficient Parameter Update and Communication.** LM-WEATHER implements efficient on-device fine-tuning of the pretrained language model. Unlike baselines that require training from scratch, LM-WEATHER only needs fine-tuning of a relatively small number of parameters (10.38 M) on each device, with minimal device-to-server communication overhead (0.38 M). This approach facilitates highly personalized cross-domain knowledge transfer, significantly reducing the ongoing costs associated with processing the ever-changing streams of weather data. These aspects highlight the pragmatic considerations that have shaped the design of LM-WEATHER. The model's capabilities to deliver exceptional performance, combined with its efficient parameter tuning and communication strategies, offer a cost-effective solution for advanced weather data analysis in resource-constrained environments.

### D.5   Additional Tasks for Potential Applications (RQ5)

Given datasets we proposed in this paper focus on forecasting and imputation tasks, we broaden its scope briefly to explore its potential application by integrating anomaly weather detection tasks. This involves relabeling the dataset to identify intervals with anomalous meteorological variables as instances of abnormal weather processes. Specifically, we label original datasets via Isolation Forecast [58], the main process as follows: (1) We set the cut length to 100, using this metric to segment each channel (variable) and construct several random trees that collectively form a forest. (2) The "isolation" degree of each data point is quantified by the average path length from the root node to the terminal node. (3) Data points with shorter path lengths are more easily isolated and

thus more likely to be outliers. (4) We establish a threshold based on the average path length; data points falling below this threshold are classified as anomalies.

**Evaluation Metrics.**   We used Precision (P), Recall (R), and F1-Score (F1) to simply quantify the performance of LM-WEATHER and baselines on the weather anomaly detection task, these can be formulated as:

$$P = \frac{TP}{TP + FP}, \qquad R = \frac{TP}{TP + FN}, \qquad F1 = 2 \times \frac{P \times R}{P + R}, \qquad (20)$$

where TP (True Positives), FP (False Positives), and FN (False Negatives) represent the number of samples correctly labeled as anomalous, the number of samples incorrectly labeled as anomalous, and the number of samples that were not labeled as anomalous by the model but were actually anomalous, respectively.

**Experiments and Results.**   We set the input time series length to 100, and other settings (e.g., baselines, hyper-parameters, local updating steps and federated communication rounds, etc.) are consistent with those in the main text, and we conduct experiments on OWD1T and OWD2T to briefly show the results. The performance quantification of our proposed LM-WEATHER and baseline on weather anomaly detection tasks is shown in **Table 26** (ODW1T results) and **Table 27** (ODW2T results).The findings underscore LM-WEATHER's robust applicability and its Moreover, LM-WEATHER's superior performance over baselines in both regular and few-shot tasks reaffirms its effectiveness and overall superiority. Moreover, LM-WEATHER's superior performance over baselines in both normal and few-shot tasks reaffirms its effectiveness and overall superiority.

Table 26: Results of LM-WEATHER and baseline for weather anomaly detection tasks on ODW1T, including regular and few-shot scenarios, where 5% means that 5% of the data is used in training, **Bold** and Underline denote the best and the second best.

| Scenario | Regular | | | Few-Shot | | |
|---|---|---|---|---|---|---|
| Method/Metrics | P | R | F1 | P | R | F1 |
| FL-DLinear | 82.33 | 78.54 | 80.34 | 69.72 | 71.22 | 70.48 |
| FL-LightTS | 86.11 | 72.89 | 78.76 | 70.46 | 70.86 | 70.54 |
| FL-PatchTST | 88.94 | 82.57 | 85.55 | 73.47 | 71.52 | 72.41 |
| FL-Transformer | 74.23 | 76.61 | 75.36 | 64.27 | 66.89 | 65.65 |
| FL-iTransformer | 83.68 | 82.96 | 83.17 | 67.8 | 74.87 | 71.21 |
| FL-Informer | 76.87 | 78.32 | 77.32 | 69.24 | 71.23 | 70.2 |
| FL-Reformer | 78.21 | 79.76 | 78.87 | 70.22 | 67.43 | 68.55 |
| FL-Pyraformer | 80.28 | 83.66 | 81.71 | 67.46 | 69.05 | 68.06 |
| FL-GPT4TS | 89.72 | 85.43 | 87.44 | 76.34 | 77.31 | 76.76 |
| LM-WEATHER-Ave | 90.21 | 88.14 | 89.18 | 80.41 | 82.68 | 81.46 |
| LM-WEATHER (Ours) | **92.00** | **90.45** | **91.15** | **84.25** | **86.23** | **85.09** |

Table 27: Results of LM-WEATHER and baseline for weather anomaly detection tasks on ODW2T, including regular and few-shot scenarios, where 5% means that 5% of the data is used in training, **Bold** and Underline denote the best and the second best.

| Scenario | Regular | | | Few-Shot | | |
|---|---|---|---|---|---|---|
| Method/Metrics | P | R | F1 | P | R | F1 |
| FL-DLinear | 80.21 | 75.88 | 78.32 | 72.56 | 73.75 | 72.98 |
| FL-LightTS | 84.30 | 81.74 | 82.70 | 68.68 | 70.24 | 69.56 |
| FL-PatchTST | 80.65 | 81.36 | 80.91 | 74.23 | 70.06 | 72.29 |
| FL-Transformer | 76.45 | 78.11 | 77.18 | 68.97 | 65.34 | 67.02 |
| FL-iTransformer | 81.76 | 69.53 | 75.07 | 67.32 | 70.74 | 69.02 |
| FL-Informer | 78.07 | 78.38 | 78.04 | 68.09 | 68.84 | 68.15 |
| FL-Reformer | 74.47 | 84.26 | 79.04 | 72.36 | 77.46 | 74.61 |
| FL-Pyraformer | 77.48 | 80.23 | 78.86 | 66.78 | 70.90 | 68.83 |
| FL-GPT4TS | 87.37 | 83.85 | 85.32 | 78.48 | 80.22 | 79.32 |
| LM-WEATHER-Ave | 89.99 | 89.20 | 89.55 | 86.24 | 84.33 | 85.10 |
| LM-WEATHER (Ours) | **90.49** | **95.45** | **92.98** | **88.37** | **87.47** | **87.64** |

# Appendix E  Full Experiment Results

In this section, we provide the full experimental results not included in the main manuscript. This includes the main experiments (**Appendix E.1**), few-shot learning experiments (**Appendix E.2**), and ablation studies (**Appendix E.3**), as well as extra analysis of our framework, covering hyperparameter sensitivity (**Appendix E.4**) and its performance with different PLMs.

## E.1  Full Main Results

In this section, we show detailed and full experimental results including:

- Forecasting (**Tab. 28**) and imputation (**Tab. 29**) across different scenes and settings on the **ODW1T** dataset.
- Forecasting (**Tab. 30**) and imputation (**Tab. 31**) across different scenes and settings on the **ODW1V** dataset.
- Forecasting (**Tab. 32**) and imputation (**Tab. 33**) across different scenes and settings on the **ODW2T** dataset.
- Forecasting (**Tab. 34**) and imputation (**Tab. 35**) across different scenarios and settings on the **ODW2V** dataset.

Note that we only show the comparison between the proposed LM-WEATHER and the time series-specific baseline in the full experimental results. Our LM-WEATHER outperforms specialized time-series analysis models on on-device weather datasets across various environments. Unlike these models, our method doesn't require training from scratch but only minor adjustments to a small number of parameters. This validates the effectiveness and superiority of our proposed framework in on-device weather modeling practice.

Table 28: Comparison of the performance of LM-WEATHER and baselines on the **ODW1T** under forecasting tasks. **Bold**: the best, Underline: the second best.

| Method | | LM-WEATHER-AVE | | LM-WEATHER | | FL-Reformer | | FL-Pyraformer | | FL-DLinear | | FL-PatchTST | | FL-iTransformer | | FL-LightTS | | FL-Transformer | | FL-Informer | |
|---|---|---|---|---|---|---|---|---|---|---|---|---|---|---|---|---|---|---|---|---|---|---|
| Variable | Length | MAE | RMSE | MAE | RMSE | MAE | RMSE | MAE | RMSE | MAE | RMSE | MAE | RMSE | MAE | RMSE | MAE | RMSE | MAE | RMSE | MAE | RMSE |
| Temperature | 96 | 27.8 | 36.6 | **26.6** | **34.7** | 58.8 | 72.3 | 51.1 | 62.8 | 27.9 | 36.9 | 30.2 | 39.4 | 32.7 | 42.3 | 28.2 | 37.2 | 38.1 | 50.2 | 41.2 | 51.9 |
| | 192 | 31.9 | 38.4 | **30.4** | **36.5** | 73.5 | 89.5 | 59.8 | 72.7 | 31.8 | 41.6 | 34.1 | 44.1 | 34.8 | 44.9 | 32.1 | 41.9 | 39.7 | 50.5 | 43.0 | 54.1 |
| | 336 | 33.3 | 40.1 | **31.6** | **38.0** | 83.7 | 101.5 | 66.1 | 79.7 | 34.8 | 45.0 | 36.5 | 46.6 | 37.1 | 47.4 | 35.2 | 45.4 | 40.7 | 51.5 | 44.7 | 55.9 |
| | 720 | 39.2 | 48.0 | **37.2** | **45.4** | 89.9 | 114.2 | 67.1 | 80.9 | 42.9 | 53.8 | 46.0 | 57.5 | 47.0 | 58.6 | 43.1 | 54.1 | 51.5 | 64.1 | 51.5 | 64.0 |
| | Avg. | 33.0 | 40.8 | **31.5** | **38.7** | 76.5 | 94.4 | 61.0 | 74.0 | 34.3 | 44.3 | 36.7 | 46.9 | 37.9 | 48.3 | 34.6 | 44.6 | 42.5 | 54.1 | 45.1 | 56.5 |
| Humidity | 96 | 52.0 | 67.4 | **49.9** | **64.0** | 75.8 | 92.0 | 70.7 | 85.2 | 53.3 | 69.5 | 57.9 | 75.4 | 59.3 | 77.2 | 53.6 | 69.8 | 69.4 | 87.1 | 73.6 | 91.7 |
| | 192 | 55.9 | 71.6 | **53.0** | **67.8** | 80.1 | 96.6 | 74.8 | 89.5 | 57.5 | 73.9 | 60.4 | 79.1 | 61.9 | 80.3 | 57.6 | 74.3 | 68.4 | 86.7 | 75.7 | 94.1 |
| | 336 | 58.6 | 74.0 | **55.6** | **70.1** | 85.0 | 102.2 | 79.2 | 94.6 | 60.4 | 76.4 | 75.1 | 93.3 | 63.6 | 81.8 | 60.3 | 77.0 | 68.6 | 86.8 | 77.2 | 95.4 |
| | 720 | 62.1 | 77.0 | **59.0** | **73.1** | 88.1 | 105.1 | 81.9 | 97.2 | 64.1 | 79.5 | 72.7 | 89.9 | 66.6 | 84.5 | 65.2 | 80.2 | 79.5 | 97.4 | 79.5 | 97.4 |
| | Avg. | 56.7 | 72.5 | **54.4** | **68.8** | 83.0 | 99.0 | 76.7 | 91.6 | 58.8 | 74.9 | 66.5 | 84.5 | 62.8 | 80.9 | 59.2 | 75.3 | 71.5 | 89.5 | 76.5 | 94.6 |
| Wind speed | 96 | 68.4 | 88.5 | **65.7** | **84.0** | 80.3 | 105.0 | 74.8 | 97.1 | 69.8 | 91.1 | 72.5 | 95.1 | 73.8 | 96.1 | 69.6 | 91.0 | 80.5 | 102.6 | 79.8 | 102.5 |
| | 192 | 69.5 | 89.7 | **66.7** | **85.0** | 83.0 | 108.0 | 77.3 | 99.7 | 71.0 | 92.4 | 73.0 | 96.7 | 74.4 | 97.0 | 70.8 | 92.3 | 80.4 | 103.4 | 80.5 | 103.4 |
| | 336 | 70.1 | 90.4 | **67.2** | **85.7** | 84.5 | 109.8 | 78.7 | 101.3 | 71.6 | 93.2 | 74.9 | 98.0 | 75.3 | 98.2 | 71.5 | 93.2 | 81.1 | 104.0 | 81.0 | 104.2 |
| | 720 | 70.9 | 91.4 | **68.0** | **86.5** | 85.7 | 232.3 | 79.8 | 215.0 | 72.6 | 94.3 | 76.4 | 99.9 | 76.0 | 99.4 | 72.5 | 94.2 | 81.8 | 105.5 | 81.8 | 105.5 |
| | Avg. | 69.7 | 90.0 | **66.9** | **85.3** | 83.4 | 138.8 | 77.6 | 128.3 | 71.2 | 92.8 | 74.2 | 97.4 | 74.9 | 97.7 | 71.1 | 92.7 | 80.9 | 103.9 | 80.8 | 103.9 |
| Surface Temperature | 96 | 28.1 | 39.2 | **27.0** | **37.4** | 55.7 | 72.9 | 52.0 | 67.6 | 28.1 | 39.9 | 31.4 | 43.0 | 33.0 | 44.9 | 28.3 | 40.0 | 56.9 | 72.0 | 56.4 | 72.0 |
| | 192 | 30.2 | 42.1 | **29.0** | **39.9** | 63.0 | 81.1 | 58.7 | 75.0 | 31.3 | 43.8 | 33.9 | 46.2 | 34.2 | 46.6 | 31.5 | 43.9 | 57.8 | 73.1 | 57.8 | 73.1 |
| | 336 | 32.4 | 44.7 | **31.1** | **42.4** | 69.6 | 88.9 | 64.7 | 82.1 | 33.9 | 46.7 | 57.4 | 75.9 | 36.3 | 49.1 | 34.3 | 47.1 | 44.0 | 58.9 | 58.8 | 74.4 |
| | 720 | 38.6 | 50.4 | **36.8** | **47.9** | 73.2 | 93.3 | 68.0 | 86.2 | 40.3 | 53.5 | 43.3 | 57.4 | 42.4 | 55.8 | 40.7 | 53.9 | 62.2 | 78.9 | 61.7 | 78.9 |
| | Avg. | 32.3 | 44.1 | **31.0** | **41.9** | 65.4 | 84.1 | 60.9 | 77.7 | 33.4 | 46.0 | 41.5 | 55.6 | 36.5 | 49.1 | 33.7 | 46.2 | 51.1 | 66.3 | 58.7 | 74.6 |
| All | 96 | 44.1 | 74.8 | **42.3** | **71.1** | 70.7 | 92.9 | 67.2 | 86.1 | 49.7 | 78.6 | 45.0 | 77.0 | 48.4 | 80.2 | 54.8 | 85.6 | 50.7 | 82.1 | 51.9 | 83.2 |
| | 192 | 46.3 | 77.5 | **44.4** | **73.6** | 75.1 | 98.3 | 70.0 | 90.9 | 52.3 | 81.8 | 47.3 | 79.8 | 51.8 | 84.3 | 59.5 | 90.6 | 52.1 | 84.0 | 52.9 | 84.6 |
| | 336 | 47.9 | 79.3 | **45.8** | **75.2** | 79.8 | 100.5 | 74.1 | 92.8 | 53.9 | 83.7 | 49.0 | 81.7 | 54.5 | 87.3 | 64.0 | 94.6 | 52.9 | 85.2 | 53.5 | 85.6 |
| | 720 | 51.8 | 83.0 | **49.2** | **78.5** | 87.1 | 102.9 | 80.5 | 95.2 | 57.2 | 87.3 | 53.3 | 85.6 | 60.1 | 93.1 | 72.4 | 102.7 | 55.4 | 87.6 | 55.3 | 87.4 |
| | Avg. | 47.5 | 78.7 | **45.4** | **74.6** | 78.2 | 98.7 | 73.0 | 91.3 | 53.3 | 82.8 | 48.6 | 81.0 | 53.7 | 63.7 | 62.7 | 93.4 | 52.8 | 84.7 | 53.4 | 85.2 |
| 1st Count | | | 0 | | 50 | | 0 | | 0 | | 0 | | 0 | | 0 | | 0 | | 0 | | 0 |

Table 29: Comparison of the performance of the proposed method and the baseline method on the **ODW1T** under the imputation task, where **bold** indicates the optimal results and underline indicates the sub-optimal results.

| Method | | LM-WEATHER-AVE | | LM-WEATHER | | FL-Reformer | | FL-Pyraformer | | FL-DLinear | | FL-PatchTST | | FL-iTransformer | | FL-LightTS | | FL-Transformer | | FL-Informer | |
|---|---|---|---|---|---|---|---|---|---|---|---|---|---|---|---|---|---|---|---|---|---|---|
| Ratio | Length | MAE | RMSE | MAE | RMSE | MAE | RMSE | MAE | RMSE | MAE | RMSE | MAE | RMSE | MAE | RMSE | MAE | RMSE | MAE | RMSE | MAE | RMSE |
| 25% | 96 | 20.8 | 39.8 | **20.2** | **38.6** | 39.0 | 57.7 | 38.2 | 56.0 | 23.6 | 43.8 | 25.1 | 48.9 | 21.0 | 40.3 | 20.5 | 38.9 | 56.3 | 82.6 | 40.2 | 60.1 |
| | 192 | **20.3** | 38.5 | 20.7 | 37.0 | 39.2 | 56.5 | 38.3 | 54.9 | 23.4 | 43.7 | 41.9 | 69.4 | 22.6 | 39.1 | 22.0 | 41.0 | 57.4 | 82.2 | 40.5 | 58.9 |
| | 336 | 21.4 | 39.9 | **21.0** | **38.4** | 54.5 | 77.0 | 53.0 | 74.8 | 23.2 | 43.3 | 60.7 | 96.3 | 24.5 | 45.7 | 22.6 | 41.8 | 58.0 | 80.7 | 56.6 | 80.2 |
| | 720 | 22.3 | 40.9 | **21.5** | **39.3** | 69.1 | 93.1 | 67.1 | 90.3 | 23.2 | 43.2 | 67.1 | 107.3 | 27.7 | 47.8 | 22.7 | 42.1 | 61.6 | 84.0 | 71.7 | 97.0 |
| | Avg. | 21.2 | 39.8 | **21.1** | **38.3** | 50.5 | 71.1 | 49.2 | 69.0 | 23.4 | 43.5 | 48.7 | 80.5 | 23.9 | 43.2 | 22.0 | 40.9 | 58.3 | 82.4 | 52.3 | 74.0 |
| 35% | 96 | 21.8 | 41.2 | **21.1** | **39.8** | 69.2 | 95.2 | 67.5 | 92.5 | 25.9 | 46.6 | 28.9 | 53.6 | 22.2 | 42.1 | 21.9 | 40.6 | 57.3 | 81.7 | 71.8 | 99.2 |
| | 192 | 22.0 | 39.9 | **21.3** | **38.3** | 70.3 | 94.5 | 68.5 | 91.7 | 25.6 | 46.3 | 25.5 | 49.4 | 26.6 | 43.7 | 23.6 | 42.8 | 58.4 | 82.7 | 42.4 | 61.4 |
| | 336 | 22.8 | 41.2 | **22.1** | **39.5** | 71.3 | 96.2 | 69.4 | 93.2 | 25.5 | 45.7 | 37.9 | 64.0 | 29.6 | 44.2 | 24.5 | 43.7 | 56.1 | 80.4 | 36.7 | 53.8 |
| | 720 | 23.9 | 41.5 | **23.2** | **39.8** | 77.8 | 99.5 | 75.6 | 96.6 | 25.2 | 45.4 | 49.2 | 70.2 | 33.2 | 45.2 | 24.6 | 44.2 | 64.2 | 88.4 | 73.3 | 98.5 |
| | Avg. | 22.6 | 41.0 | **21.9** | **39.4** | 72.2 | 96.4 | 70.3 | 93.5 | 25.5 | 46.0 | 35.4 | 59.3 | 27.9 | 43.8 | 23.6 | 42.8 | 59.0 | 83.3 | 56.0 | 78.2 |
| 50% | 96 | 22.4 | 43.5 | **21.7** | **41.8** | 63.7 | 88.4 | 62.2 | 85.9 | 29.2 | 50.8 | 28.9 | 54.6 | 22.8 | 44.5 | 24.4 | 43.7 | 58.3 | 82.8 | 70.8 | 99.6 |
| | 192 | 23.4 | 43.7 | **22.6** | **42.0** | 67.2 | 91.2 | 65.5 | 88.5 | 28.7 | 50.2 | 47.5 | 77.3 | 23.8 | 44.1 | 25.7 | 45.3 | 57.3 | 82.4 | 66.3 | 92.1 |
| | 336 | 24.1 | 44.1 | **23.2** | **42.4** | 70.4 | 93.4 | 68.5 | 90.6 | 28.3 | 49.4 | 48.6 | 77.0 | 27.2 | 47.7 | 26.9 | 46.6 | 58.4 | 83.5 | 36.9 | 55.3 |
| | 720 | 26.0 | 45.1 | **24.9** | **43.3** | 77.9 | 96.8 | 75.8 | 93.9 | 28.0 | 49.0 | 56.6 | 85.1 | 36.5 | 56.2 | 27.2 | 47.4 | 56.6 | 80.4 | 71.7 | 96.7 |
| | Avg. | 24.0 | 44.1 | **23.1** | **42.4** | 69.8 | 92.5 | 68.0 | 89.7 | 28.5 | 49.9 | 45.4 | 73.5 | 27.6 | 48.2 | 26.1 | 45.7 | 57.6 | 82.3 | 61.4 | 85.9 |
| 1st Count | | | 1 | | 29 | | 0 | | 0 | | 0 | | 0 | | 0 | | 0 | | 0 | | 0 |

Table 30: Comparison of the performance of LM-WEATHER and baselines on the **ODW1V** under the long-term forecasting task. **Bold**: the best, Underline: the second best.

| Method | LM-WEATHER-ave | | LM-WEATHER | | FL-DLinear | | FL-PatchTST | | FL-iTransformer | | FL-LightTS | | FL-Transformer | | FL-Informer | | FL-Reformer | | FL-Pyraformer | |
|---|---|---|---|---|---|---|---|---|---|---|---|---|---|---|---|---|---|---|---|---|
| Variable / Length | MAE | RMSE | MAE | RMSE | MAE | RMSE | MAE | RMSE | MAE | RMSE | MAE | RMSE | MAE | RMSE | MAE | RMSE | MAE | RMSE | MAE | RMSE |
| Temperature 96 | 32.0 | 38.9 | **28.0** | **37.0** | 28.9 | 36.9 | 32.5 | 46.1 | 32.6 | 42.2 | 32.2 | 39.2 | 40.6 | 51.2 | 87.5 | 101.1 | 87.4 | 101.1 | 84.4 | 98.3 |
| Temperature 192 | 36.8 | 44.4 | **31.8** | **41.4** | 35.8 | 46.6 | 39.2 | 50.5 | 34.7 | 44.8 | 37.1 | 44.9 | 39.3 | 50.0 | 87.2 | 100.8 | 87.3 | 100.9 | 87.3 | 100.8 |
| Temperature 336 | 39.1 | 48.0 | **34.6** | **44.7** | 39.8 | 51.0 | 39.5 | 50.7 | 36.9 | 47.2 | 39.7 | 48.9 | 44.6 | 55.8 | 87.1 | 100.5 | 87.1 | 100.5 | 87.8 | 101.1 |
| Temperature 720 | 49.0 | 58.8 | **43.1** | **54.2** | 48.9 | 59.8 | 54.8 | 67.4 | 45.4 | 56.7 | 49.4 | 60.3 | 51.5 | 64.1 | 87.0 | 100.0 | 88.8 | 104.1 | 86.7 | 99.7 |
| Temperature Avg. | 39.2 | 47.5 | **34.3** | **44.3** | 38.3 | 48.6 | 41.5 | 53.7 | 37.4 | 47.7 | 39.6 | 48.3 | 44.0 | 55.3 | 87.2 | 100.6 | 87.6 | 101.6 | 86.6 | 100.0 |
| All 96 | 42.7 | 69.5 | **42.3** | 69.6 | 42.9 | 67.8 | 57.7 | **67.2** | 46.4 | 73.3 | 44.3 | 69.6 | 56.8 | 76.8 | 48.0 | 75.1 | 67.0 | 89.4 | 59.0 | 80.3 |
| All 192 | 45.5 | 72.6 | **44.4** | 71.7 | 48.4 | 75.4 | 59.2 | **69.4** | 47.9 | 75.1 | 46.8 | 72.1 | 55.0 | 75.0 | 49.1 | 79.2 | 69.9 | 93.0 | 61.2 | 82.8 |
| All 336 | 47.2 | 74.3 | **46.0** | 72.4 | 51.0 | 77.0 | 63.4 | 73.3 | 49.1 | 76.9 | 48.5 | 74.8 | 62.4 | 83.7 | 50.8 | 77.9 | 71.4 | 94.8 | 63.7 | 85.8 |
| All 720 | 51.2 | 78.2 | **49.7** | 74.0 | 54.5 | 82.3 | 67.3 | 76.1 | 52.5 | 80.3 | 54.3 | 79.1 | 72.1 | 96.2 | 54.7 | 82.7 | 76.2 | 87.3 | 68.4 | 91.8 |
| All Avg. | 46.6 | 73.6 | **45.6** | 71.9 | 49.2 | 75.6 | 61.9 | **71.5** | 49.0 | 76.4 | 48.5 | 73.9 | 58.1 | 85.0 | 50.7 | 78.7 | 71.1 | 91.1 | 63.1 | 85.2 |
| 1st Count | 0 | | 17 | | 0 | | 3 | | 0 | | 0 | | 0 | | 0 | | 0 | | 0 | |

Table 31: Comparison of the performance of LM-WEATHER and baselines on the **ODW1V** dataset under the imputation task. **Bold**: the best, Underline: the second best.

| Method | LM-WEATHER-ave | | LM-WEATHER | | FL-DLinear | | FL-PatchTST | | FL-iTransformer | | FL-LightTS | | FL-Transformer | | FL-Informer | | FL-Reformer | | FL-Pyraformer | |
|---|---|---|---|---|---|---|---|---|---|---|---|---|---|---|---|---|---|---|---|---|
| Ratio / Length | MAE | RMSE | MAE | RMSE | MAE | RMSE | MAE | RMSE | MAE | RMSE | MAE | RMSE | MAE | RMSE | MAE | RMSE | MAE | RMSE | MAE | RMSE |
| 25% 96 | 36.7 | 56.8 | 35.9 | 55.4 | 37.9 | 59.4 | **29.2** | **48.6** | 31.8 | 50.7 | 45.2 | 68.6 | 34.5 | 53.1 | 36.7 | 55.8 | 34.5 | 52.8 | 33.6 | 50.3 |
| 25% 192 | 38.8 | 59.8 | 37.9 | 58.2 | 40.3 | 62.6 | **31.4** | **51.1** | 39.2 | 60.3 | 50.4 | 74.3 | 34.8 | 53.4 | 38.9 | 59.8 | 34.7 | 53.2 | 41.7 | 58.5 |
| 25% 336 | 42.3 | 63.8 | 41.3 | 62.2 | 44.0 | 66.8 | **35.1** | **56.8** | 45.6 | 68.6 | 54.4 | 78.6 | 45.2 | 58.0 | 41.1 | 61.1 | 45.0 | 63.7 | 46.6 | 61.3 |
| 25% 720 | 44.5 | 66.1 | 40.2 | 60.1 | 46.3 | 69.3 | 42.2 | 65.5 | 51.8 | 76.0 | 58.2 | 82.2 | 45.5 | 64.3 | 48.3 | 66.3 | 48.4 | 65.9 | 56.4 | 72.5 |
| 25% Avg. | 40.6 | 60.1 | 40.2 | 60.1 | 42.1 | 64.5 | **34.5** | **55.5** | 42.1 | 63.9 | 52.1 | 75.9 | 40.0 | 57.2 | 41.3 | 60.8 | 40.6 | 58.9 | 44.6 | 60.7 |
| 35% 96 | 38.1 | 58.5 | 37.3 | 57.1 | 39.7 | 61.2 | 40.3 | 50.2 | 35.9 | 55.1 | 48.2 | 71.6 | 38.4 | 55.2 | 38.8 | 58.2 | 35.9 | 54.6 | 36.6 | 53.8 |
| 35% 192 | 40.2 | 61.2 | 39.3 | 59.8 | 41.9 | 64.2 | 42.5 | 52.6 | 43.1 | 64.5 | 53.0 | 76.9 | 46.8 | 65.7 | 39.1 | 59.4 | 36.1 | 54.9 | 44.4 | 61.6 |
| 35% 336 | 40.8 | 62.4 | 39.9 | 60.9 | 42.5 | 65.4 | 51.7 | 64.1 | 48.6 | 71.7 | 56.8 | 81.0 | 47.3 | 66.4 | 44.3 | 60.9 | 46.5 | 65.4 | 48.1 | 65.2 |
| 35% 720 | 43.1 | 65.1 | 42.1 | 63.5 | 44.9 | 68.2 | 56.9 | 70.4 | 53.7 | 77.9 | 60.1 | 84.2 | 60.7 | 76.9 | 49.5 | 69.0 | 50.4 | 69.0 | 56.5 | 74.2 |
| 35% Avg. | 40.6 | 62.9 | 39.7 | 61.4 | 42.3 | 64.7 | 47.9 | 59.3 | 45.3 | 67.3 | 54.5 | 78.4 | 48.3 | 66.1 | 42.9 | 61.9 | 42.2 | 61.0 | 46.4 | 63.7 |
| 50% 96 | 42.1 | 62.0 | 41.1 | 60.4 | 43.8 | 64.9 | 42.3 | 53.0 | 43.0 | 63.0 | 53.6 | 77.1 | 38.7 | 58.2 | 41.5 | 61.5 | 37.8 | 56.9 | 41.1 | 59.2 |
| 50% 192 | 43.9 | 64.5 | 42.8 | 62.8 | 45.8 | 67.6 | 44.7 | 56.2 | 49.3 | 71.2 | 57.5 | 81.5 | 49.3 | 68.9 | 41.9 | 62.0 | 44.1 | 57.4 | 48.8 | 66.8 |
| 50% 336 | 45.7 | 66.6 | 44.6 | 64.9 | 47.6 | 69.8 | 54.6 | 65.7 | 53.4 | 76.6 | 60.7 | 85.0 | 60.0 | 79.8 | 47.3 | 64.6 | 48.5 | 68.0 | 50.2 | 67.1 |
| 50% 720 | 47.5 | 68.7 | 46.3 | 66.9 | 49.6 | 72.0 | 59.2 | 73.5 | 56.8 | 80.7 | 63.3 | 87.4 | 61.6 | 80.4 | 52.5 | 72.9 | 52.7 | 70.1 | 60.3 | 77.2 |
| 50% Avg. | 44.8 | 65.5 | 43.7 | 63.8 | 46.7 | 68.6 | 50.2 | 62.1 | 50.6 | 72.9 | 58.8 | 82.7 | 52.4 | 71.8 | 45.8 | 65.3 | 45.8 | 63.1 | 50.1 | 67.6 |
| 1st Count | 0 | | 10 | | 0 | | 9 | | 1 | | 0 | | 1 | | 1 | | 4 | | 1 | |

Table 32: Comparison of the performance of LM-WEATHER and baselines on the **ODW2T** under the long-term forecasting task. **Bold**: the best, Underline: the second best.

| Method | LM-WEATHER-ave | | LM-WEATHER | | FL-Reformer | | FL-Pyraformer | | FL-DLinear | | FL-PatchTST | | FL-iTransformer | | FL-LightTS | | FL-Transformer | | FL-Informer | |
|---|---|---|---|---|---|---|---|---|---|---|---|---|---|---|---|---|---|---|---|---|
| Metrics / Ratio | MAE | RMSE | MAE | RMSE | MAE | RMSE | MAE | RMSE | MAE | RMSE | MAE | RMSE | MAE | RMSE | MAE | RMSE | MAE | RMSE | MAE | RMSE |
| Temperature 96 | 40.6 | 52.5 | **40.2** | **51.8** | 90.8 | 107.0 | 90.5 | 107.3 | 41.1 | 53.5 | 52.2 | 65.6 | 41.2 | 53.2 | 41.8 | 54.1 | 58.7 | 72.7 | 59.1 | 73.4 |
| Temperature 192 | 45.3 | 57.8 | **44.6** | **56.9** | 97.3 | 119.3 | 96.9 | 114.3 | 46.2 | 59.0 | 48.0 | 60.8 | 46.2 | 58.8 | 46.5 | 59.4 | 60.5 | 75.2 | 60.9 | 75.5 |
| Temperature 336 | 48.4 | 61.0 | **47.7** | **60.2** | 100.3 | 121.2 | 99.8 | 117.2 | 50.1 | 63.3 | 49.2 | 62.2 | 49.2 | 62.1 | 49.7 | 62.8 | 61.1 | 76.0 | 62.2 | 77.2 |
| Temperature 720 | 55.3 | 69.2 | **54.6** | **68.1** | 130.3 | 150.3 | 127.7 | 144.1 | 61.4 | 75.1 | 58.2 | 73.0 | 56.2 | 70.4 | 59.1 | 72.9 | 67.9 | 78.2 | 66.3 | 82.7 |
| Temperature Avg. | 47.4 | 57.1 | **46.8** | 59.3 | 104.7 | 124.4 | 103.7 | 120.7 | 49.7 | 62.7 | 51.9 | 65.4 | 48.2 | 61.1 | 49.3 | 62.3 | 62.0 | 75.5 | 62.1 | 77.2 |
| Humidity 96 | 67.3 | 85.3 | **66.2** | **83.1** | 88.4 | 106.0 | 85.1 | 103.1 | 67.9 | 84.7 | 70.2 | 88.1 | 68.6 | 86.5 | 68.4 | 85.4 | 85.0 | 103.0 | 84.7 | 102.7 |
| Humidity 192 | 70.3 | 88.4 | **69** | **85.9** | 91.3 | 110.3 | 89.7 | 107.8 | 71.4 | 88.1 | 72.2 | 90.7 | 71.1 | 88.9 | 71.9 | 88.9 | 85.0 | 103.0 | 84.9 | 102.8 |
| Humidity 336 | 71.4 | 89.3 | **70** | **86.6** | 94.3 | 111.2 | 92.2 | 110.2 | 73.0 | 89.5 | 73.0 | 91.9 | 71.8 | 89.6 | 73.7 | 90.5 | 82.6 | 100.5 | 84.8 | 102.9 |
| Humidity 720 | 72.8 | 90.9 | **71.3** | **88.2** | 96.1 | 114.1 | 94.0 | 112.2 | 76.1 | 92.9 | 75.1 | 93.3 | 72.9 | 91.0 | 76.7 | 93.7 | 84.1 | 105.1 | 85.4 | 103.8 |
| Humidity Avg. | 70.5 | 88.5 | **69.1** | **86.0** | 92.5 | 110.4 | 90.2 | 108.3 | 72.1 | 88.2 | 72.6 | 91.0 | 71.1 | 89.0 | 72.7 | 89.6 | 84.2 | 102.9 | 84.9 | 103.1 |
| All 96 | 64.3 | 88.2 | **62.8** | **85.5** | 100.3 | 126.3 | 95.0 | 120.3 | 67.9 | 84.7 | 70.2 | 88.1 | 68.6 | 86.5 | 68.4 | 85.4 | 85.0 | 103.0 | 84.7 | 102.7 |
| All 192 | 67.7 | 91.5 | **66.2** | **89.1** | 102.1 | 130.3 | 99.9 | 125.8 | 71.4 | 88.1 | 72.2 | 90.7 | 71.1 | 88.9 | 71.9 | 88.9 | 85.0 | 103.0 | 84.9 | 102.8 |
| All 336 | 69.5 | 93.7 | **67.9** | **91.1** | 104.2 | 130.0 | 102.0 | 128.5 | 73.0 | 89.5 | 73.0 | 91.9 | 71.8 | 89.6 | 73.7 | 90.5 | 82.6 | 100.5 | 84.8 | 102.9 |
| All 720 | 72.6 | 97.3 | **70.7** | **94.6** | 107.3 | 134.2 | 104.2 | 131.4 | 76.1 | 92.9 | 75.1 | 93.3 | 72.9 | 91.0 | 76.7 | 93.7 | 84.1 | 105.1 | 85.4 | 103.8 |
| All Avg. | 68.5 | 92.7 | **66.9** | **90.1** | 103.5 | 130.2 | 100.3 | 126.5 | 72.1 | 88.8 | 72.6 | 91.0 | 71.1 | 89.0 | 72.7 | 89.6 | 84.2 | 102.9 | 84.9 | 103.1 |
| 1st Count | 0 | | 14 | | 4 | | 4 | | 0 | | 0 | | 0 | | 0 | | 0 | | 0 | |

Table 33: Comparison of the performance of the proposed method and the baseline method on the **ODW2T** under the imputation task. **Bold**: the best, Underline: the second best.

| Method | FLAME-ave | | FLAME | | FL-Reformer | | FL-Pyraformer | | FL-DLinear | | FL-PatchTST | | FL-iTransformer | | FL-LightTS | | FL-Transformer | | FL-Informer | |
|---|---|---|---|---|---|---|---|---|---|---|---|---|---|---|---|---|---|---|---|---|
| Ratio / Length | MAE | RMSE | MAE | RMSE | MAE | RMSE | MAE | RMSE | MAE | RMSE | MAE | RMSE | MAE | RMSE | MAE | RMSE | MAE | RMSE | MAE | RMSE |
| 25% 96 | 32.9 | 50.3 | **32.1** | 48.9 | 48.9 | 68.8 | 95.4 | 120.8 | 34.0 | 52.4 | 41.3 | 61.9 | 33.6 | 51.9 | 33.4 | 51.7 | 52.0 | 71.9 | 42.4 | 62.2 |
| 25% 192 | 36.2 | 54.9 | 35.1 | 53.2 | 51.2 | 71.1 | 96.7 | 122.0 | 34.6 | 54.7 | 46.5 | 66.4 | 37.3 | 56.6 | **33.5** | 51.8 | 62.7 | 83.8 | 43.9 | 64.0 |
| 25% 336 | 41.6 | 60.0 | 40.4 | 58.2 | 54.4 | 73.5 | 99.1 | 127.4 | **37.0** | 59.1 | 68.8 | 90.2 | 42.9 | 61.9 | 42.9 | 58.9 | 54.3 | 75.1 | 42.0 | 62.3 |
| 25% 720 | 46.1 | 65.5 | 44.9 | 63.5 | 59.2 | 78.5 | 101.2 | 128.1 | 42.2 | 63.0 | 75.1 | 98.1 | 47.5 | 67.5 | 43.1 | 62.0 | 57.8 | 77.2 | **40.8** | **60.6** |
| 25% Avg. | 41.2 | 60.1 | 40.1 | 60.9 | 53.4 | 73.0 | 98.1 | 124.6 | **36.9** | 57.3 | 57.9 | 79.1 | 40.3 | 59.5 | 38.2 | 56.1 | 56.7 | 77.0 | 42.3 | 62.2 |
| 35% 96 | 34.9 | 52.8 | **33.8** | 51.2 | 48.7 | 68.1 | 95.3 | 117.2 | 36.7 | 55.3 | 40.8 | 61.0 | 35.9 | 54.4 | 35.6 | 54.2 | 54.1 | 73.7 | 45.4 | 65.5 |
| 35% 192 | 36.5 | 55.2 | **35.3** | 53.5 | 51.4 | 72.4 | 96.2 | 119.4 | 37.1 | 56.5 | 50.6 | 71.8 | 37.6 | 57.0 | 35.9 | 54.3 | 55.6 | 75.9 | 47.1 | 67.6 |
| 35% 336 | 41.9 | 60.3 | 40.6 | 58.5 | 53.9 | 74.0 | 98.9 | 124.5 | **39.4** | 63.7 | 68.7 | 90.0 | 43.2 | 62.2 | 43.3 | 63.4 | 56.1 | 76.5 | 46.1 | 66.9 |
| 35% 720 | 47.2 | 67.6 | 45.8 | 65.6 | 61.5 | 77.7 | 100.5 | 127.3 | 44.4 | 69.4 | 76.1 | 98.5 | 48.6 | 69.7 | 44.3 | 65.4 | 57.9 | 77.8 | 45.5 | 65.8 |
| 35% Avg. | 39.5 | 59.0 | 36.6 | 59.2 | 53.9 | 73.1 | 97.7 | 122.1 | 39.4 | 61.2 | 59.1 | 80.3 | 41.3 | 60.8 | 39.8 | 59.3 | 55.9 | 76.0 | 46.0 | 66.4 |
| 50% 96 | 38.0 | 56.6 | 36.9 | 54.9 | 50.3 | 70.3 | 95.4 | 120.8 | 40.8 | 60.0 | 38.4 | 58.6 | 39.1 | 58.3 | 38.8 | 57.8 | 65.5 | 86.6 | 51.7 | 72.0 |
| 50% 192 | 38.3 | 56.6 | 37.2 | 54.9 | 52.1 | 74.2 | 96.2 | 122.3 | 42.9 | 62.7 | 66.7 | 87.8 | 39.4 | 58.3 | 39.5 | 58.4 | 71.4 | 92.8 | 55.0 | 75.7 |
| 50% 336 | 43.5 | 65.5 | 42.2 | 63.5 | 56.6 | 78.9 | 97.8 | 125.5 | 46.0 | 67.1 | 68.7 | 90.1 | 44.8 | 67.5 | 47.8 | 65.3 | 66.8 | 88.8 | 51.5 | 72.8 |
| 50% 720 | 47.9 | 68.8 | 46.5 | 66.7 | 64.3 | 87.7 | 99.1 | 129.9 | 52.8 | 76.1 | 70.4 | 93.5 | 49.3 | 71.0 | 48.0 | 68.0 | 67.4 | 89.2 | 51.5 | 73.0 |
| 50% Avg. | 41.9 | 61.9 | 38.8 | 61.7 | 55.8 | 77.8 | 97.1 | 124.6 | 45.6 | 66.6 | 61.1 | 82.5 | 43.2 | 63.8 | 43.5 | 62.4 | 67.8 | 89.4 | 52.4 | 73.4 |
| 1st Count | 0 | | 20 | | 0 | | 0 | | 3 | | 0 | | 0 | | 5 | | 0 | | 2 | |

## E.2 Full Few-Shot Learning Experiments

In this section, we show detailed and full few-shot learning experimental results including:

Table 34: Comparison of the performance of LM-WEATHER and the baseline method on the **ODW2V** under the forecasting task. **Bold**: the best, Underline: the second best.

| Metho | | FLAME-ave | | FLAME | | FL-Reformer | | FL-Pyraformer | | FL-DLinear | | FL-PatchTST | | FL-iTransformer | | FL-LightTS | | FL-Transformer | | FL-Informer | |
|---|---|---|---|---|---|---|---|---|---|---|---|---|---|---|---|---|---|---|---|---|---|
| Variable | Length | MAE | RMSE | MAE | RMSE | MAE | RMSE | MAE | RMSE | MAE | RMSE | MAE | RMSE | MAE | RMSE | MAE | RMSE | MAE | RMSE | MAE | RMSE |
| Humidity | 96 | 79.7 | 97.5 | **66.7** | **84.6** | 88.5 | 106.0 | 85.7 | 103.5 | 78.2 | 94.8 | 87.7 | 92.8 | 84.3 | 93.2 | 90.3 | 110.2 | 83.7 | 101.5 | 97.8 | 116.4 |
| | 192 | 84.0 | 102.2 | **71.4** | **90.3** | 89.7 | 109.5 | 89.2 | 107.1 | 87.6 | 98.8 | 91.4 | 106.1 | 90.1 | 99.5 | 97.7 | 119.3 | 84.8 | 102.7 | 96.5 | 114.6 |
| | 336 | 84.4 | 102.6 | **70.6** | **88.5** | 93.3 | 111.3 | 91.6 | 109.3 | 93.0 | 106.5 | 93.9 | 108.2 | 94.4 | 104.4 | 100.7 | 122.9 | 85.9 | 104.0 | 98.9 | 117.4 |
| | 720 | 85.0 | 103.5 | **72.1** | **90.3** | 97.9 | 114.2 | 93.5 | 111.5 | 94.6 | 108.6 | 98.5 | 103.2 | 99.7 | 112.7 | 104.5 | 127.5 | 89.4 | 109.3 | 99.3 | 119.4 |
| | Avg. | 83.3 | 101.4 | **70.2** | **88.4** | 92.4 | 110.2 | 90.0 | 107.9 | 88.3 | 102.2 | 92.9 | 102.6 | 92.1 | 102.4 | 98.3 | 120.0 | 86.0 | 104.4 | 98.1 | 117.0 |
| All | 96 | 76.8 | 99.7 | **65.1** | **88.4** | 89.6 | 112.7 | 89.1 | 112.5 | 74.8 | 96.8 | 76.3 | 99.9 | 73.5 | 97.7 | 92.2 | 117.7 | 77.0 | 100.1 | 77.4 | 100.4 |
| | 192 | 77.9 | 100.8 | **68.3** | **91.4** | 90.5 | 114.2 | 96.4 | 120.1 | 76.6 | 98.9 | 79.9 | 103.3 | 78.8 | 103.6 | 100.5 | 128.1 | 78.3 | 101.8 | 78.0 | 101.1 |
| | 336 | 78.5 | 101.5 | **69.9** | **93.0** | 94.2 | 119.3 | 98.4 | 122.2 | 77.6 | 100.2 | 81.8 | 105.3 | 82.1 | 107.5 | 105.5 | 134.4 | 79.4 | 103.3 | 78.7 | 102.0 |
| | 720 | 79.9 | 103.6 | **72.9** | **96.5** | 97.4 | 120.4 | 100.5 | 125.0 | 79.6 | 103.0 | 86.2 | 100.2 | 86.2 | 112.7 | 111.0 | 141.3 | 86.1 | 112.3 | 81.3 | 105.6 |
| | Avg. | 78.3 | 101.4 | **69.0** | **92.3** | 92.9 | 116.6 | 96.1 | 120.0 | 77.2 | 99.7 | 81.1 | 102.2 | 80.2 | 105.4 | 102.3 | 130.4 | 80.2 | 104.4 | 78.8 | 102.2 |
| 1st Count | | 0 | | **20** | | 0 | | 0 | | 0 | | 0 | | 0 | | 0 | | 0 | | 0 | |

Table 35: Comparison of the performance of LM-WEATHER and the baseline method on the **ODW2V** under the imputation task. **Bold**: the best, Underline: the second best.

| Method | | LM-WEATHER-ave | | LM-WEATHER | | FL-Reformer | | FL-Pyraformer | | FL-DLinear | | FL-PatchTST | | FL-iTransformer | | FL-LightTS | | FL-Transformer | | FL-Informer | |
|---|---|---|---|---|---|---|---|---|---|---|---|---|---|---|---|---|---|---|---|---|---|
| Ratio | Length | MAE | RMSE | MAE | RMSE | MAE | RMSE | MAE | RMSE | MAE | RMSE | MAE | RMSE | MAE | RMSE | MAE | RMSE | MAE | RMSE | MAE | RMSE |
| 25% | 96 | 28.1 | 45.4 | **27.5** | **44.1** | 43.9 | 63.2 | 42.1 | 59.4 | 63.4 | 82.8 | 35.1 | 52.8 | 58.6 | 78.9 | 91.2 | 117.8 | 45.2 | 64.9 | 46.2 | 66.8 |
| | 192 | 28.6 | 45.3 | **28.0** | **44.0** | 44.3 | 63.4 | 39.7 | 56.8 | 68.0 | 87.9 | 39.7 | 57.8 | 69.3 | 91.1 | 95.6 | 123.0 | 45.9 | 65.7 | 46.8 | 67.6 |
| | 336 | 30.1 | 46.6 | **29.3** | **45.2** | 47.6 | 68.7 | 38.3 | 55.9 | 70.5 | 90.8 | 44.5 | 63.2 | 76.2 | 99.3 | 100.2 | 128.4 | 46.1 | 66.0 | 47.0 | 67.6 |
| | 720 | 34.2 | 48.1 | **33.2** | **46.7** | 51.4 | 76.2 | 44.3 | 65.4 | 71.4 | 91.7 | 53.5 | 73.2 | 79.5 | 102.7 | 99.3 | 126.0 | 49.4 | 67.5 | 46.2 | 65.9 |
| | Avg. | 30.2 | 46.4 | **29.5** | **45.0** | 46.8 | 67.9 | 41.1 | 59.4 | 68.3 | 88.3 | 43.2 | 61.7 | 70.9 | 93.0 | 96.6 | 123.8 | 46.6 | 66.0 | 46.5 | 67.0 |
| 35% | 96 | 28.1 | 45.3 | **27.5** | **44.0** | 46.4 | 66.0 | 44.5 | 62.0 | 66.3 | 85.8 | 36.7 | 54.7 | 64.2 | 85.2 | 93.5 | 120.4 | 48.2 | 68.2 | 49.5 | 70.8 |
| | 192 | 28.7 | 45.4 | **28.1** | **44.1** | 46.9 | 66.3 | 41.0 | 59.1 | 70.6 | 90.5 | 41.4 | 59.8 | 73.2 | 95.5 | 97.0 | 124.5 | 49.1 | 69.4 | 50.4 | 71.9 |
| | 336 | 39.7 | 55.4 | **38.6** | **53.8** | 49.9 | 72.4 | 40.4 | 58.4 | 72.8 | 93.2 | 46.6 | 65.5 | 78.7 | 102.0 | 100.8 | 128.8 | 49.4 | 69.8 | 50.7 | 72.2 |
| | 720 | 41.2 | 59.1 | **40.0** | **57.3** | 54.5 | 78.6 | 47.8 | 69.0 | 73.4 | 93.6 | 55.5 | 75.4 | 80.8 | 103.9 | 99.1 | 125.5 | 51.4 | 71.5 | 50.0 | 70.7 |
| | Avg. | 34.4 | 51.3 | **33.6** | **49.8** | 49.4 | 70.8 | 43.4 | 62.1 | 70.8 | 90.8 | 45.1 | 63.8 | 74.2 | 96.6 | 97.6 | 124.8 | 49.5 | 69.7 | 50.1 | 71.4 |
| 50% | 96 | 28.1 | 45.3 | **27.5** | **44.0** | 50.3 | 70.3 | 53.2 | 72.4 | 72.1 | 92.0 | 39.8 | 58.4 | 72.7 | 94.7 | 96.4 | 123.5 | 52.7 | 73.2 | 54.8 | 76.9 |
| | 192 | 28.6 | 45.3 | **28.0** | **44.0** | 51.0 | 71.1 | 46.1 | 65.2 | 75.7 | 95.9 | 44.9 | 63.7 | 79.1 | 102.0 | 98.6 | 125.8 | 53.9 | 74.7 | 56.2 | 78.8 |
| | 336 | 33.7 | 49.8 | **32.7** | **48.4** | 54.2 | 76.6 | 74.2 | 97.3 | 77.3 | 97.8 | 50.9 | 70.1 | 82.6 | 106.1 | 101.2 | 128.8 | 54.4 | 75.4 | 56.8 | 79.7 |
| | 720 | 37.1 | 53.1 | **36.0** | **51.5** | 59.4 | 81.7 | 82.4 | 100.9 | 77.1 | 97.3 | 59.2 | 79.3 | 83.0 | 106.0 | 98.5 | 124.3 | 55.4 | 77.5 | 56.4 | 78.6 |
| | Avg. | 31.9 | 48.4 | **31.1** | **47.0** | 53.7 | 74.9 | 64.0 | 84.0 | 75.5 | 95.8 | 48.7 | 67.9 | 79.4 | 102.2 | 98.7 | 125.6 | 54.1 | 75.2 | 56.0 | 78.5 |
| 1st Count | | 0 | | **30** | | 0 | | 0 | | 0 | | 0 | | 0 | | 0 | | 0 | | 0 | |

- Forecasting (**Table. 36** for 5% training data, **Table. 37** for 15% training data) and imputation (**Table. 38** for 5% training data, **Table. 39** for 15% training data) across different scenes and settings on the **ODW1T** dataset.

- Forecasting (**Table. 40** for 5% training data, **Table. 41** for 15% training data) and imputation (**Table. 42** for 5% training data, **Table. 43** for 15% training data) across different scenes and settings on the **ODW1V** dataset.

- Forecasting (**Table. 44** for 5% training data, **Table. 45** for 15% training data) and imputation (**Table. 46** for 5% training data, **Table. 47** for 15% training data) across different scenes and settings on the **ODW2T** dataset.

- Forecasting (**Table. 48** for 5% training data, **Table. 49** for 15% training data) and imputation (**Table. 50** for 5% training data, **Table. 51** for 15% training data) across different scenarios and settings on the **ODW2V** dataset.

Table 36: Comparison of the performance of LM-WEATHER with the baseline method on the **ODW1T** dataset under the long-term forecasting task in a scenario where the proportion of training data is set to be 5% in the few-shot learning. **Bold**: the best, Underline: the second best, "-" denotes insufficient training data.

| Method | | LM-WEATHER-ave | | LM-WEATHER | | FL-Reformer | | FL-Pyraformer | | FL-Dlinear | | FL-PatchTST | | FL-iTransformer | | FL-Lights | | FL-Transformer | | FL-Informer | |
|---|---|---|---|---|---|---|---|---|---|---|---|---|---|---|---|---|---|---|---|---|---|
| Metrics | Length | MAE | RMSE | MAE | RMSE | MAE | RMSE | MAE | RMSE | MAE | RMSE | MAE | RMSE | MAE | RMSE | MAE | RMSE | MAE | RMSE | MAE | RMSE |
| Temperature | 96 | 91.6 | 102.1 | **84.6** | **93.9** | 298.5 | 328.2 | 295.9 | 327.3 | 94.0 | 118.9 | 85.6 | 103.8 | 91.6 | 117.0 | 304.2 | 362.4 | 147.8 | 179.4 | 103.6 | 130.4 |
| | 192 | 92.7 | 108.9 | **85.1** | **99.2** | 300.4 | 330.0 | 301.0 | 332.2 | 99.1 | 125.2 | 93.5 | 112.1 | 102.3 | 130.0 | 311.6 | 376.7 | 128.9 | 159.7 | 107.8 | 135.8 |
| | 336 | 97.0 | 110.2 | **88.6** | **101.0** | 302.6 | 331.8 | 304.4 | 335.7 | 103.9 | 130.2 | 108.3 | 130.0 | 111.1 | 140.1 | 315.0 | 379.0 | 114.7 | 143.9 | 114.0 | 143.0 |
| | 720 | - | - | - | - | - | - | - | - | - | - | - | - | - | - | - | - | - | - | - | - |
| | Avg. | 93.8 | 107.0 | 86.1 | 98.0 | 300.5 | 330.0 | 300.5 | 331.7 | 99.0 | 124.8 | 95.8 | 115.3 | 101.7 | 129.0 | 310.3 | 372.7 | 130.5 | 161.0 | 108.4 | 136.4 |
| Humidity | 96 | 84.7 | 108.0 | 78.1 | 99.4 | 111.0 | 130.1 | 105.6 | 124.9 | 90.7 | 99.8 | 78.8 | 95.0 | 81.0 | 102.8 | 115.3 | 140.2 | 111.5 | 137.0 | 87.8 | 109.5 |
| | 192 | 90.9 | 115.4 | 83.3 | 105.5 | 105.5 | 123.4 | 109.1 | 129.2 | 93.9 | 103.9 | 82.3 | 98.4 | 87.9 | 110.9 | 117.9 | 143.9 | 88.7 | 109.4 | 87.6 | 108.3 |
| | 336 | 96.2 | 117.1 | 88.1 | 107.1 | 106.7 | 125.3 | 112.5 | 134.2 | 96.2 | 106.3 | 92.0 | 106.0 | 92.0 | 108.2 | 118.4 | 144.3 | 88.0 | 112.0 | 90.3 | 111.8 |
| | 720 | - | - | - | - | - | - | - | - | - | - | - | - | - | - | - | - | - | - | - | - |
| | Avg. | 90.6 | 113.5 | 83.2 | 104.0 | 107.7 | 126.3 | 109.1 | 129.4 | 93.6 | 103.3 | 84.4 | 100.5 | 87.2 | 109.9 | 117.2 | 142.8 | 96.1 | 119.5 | 88.5 | 109.9 |
| Wind speed | 96 | 90.4 | 112.3 | 83.1 | 103.3 | 93.1 | 120.3 | 96.2 | 125.8 | 90.0 | 115.9 | 89.1 | 113.6 | 98.9 | 127.5 | 107.2 | 138.6 | 118.3 | 150.7 | 93.2 | 120.8 |
| | 192 | 94.9 | 114.5 | 87.0 | 105.0 | 92.9 | 120.5 | 97.4 | 127.1 | 91.3 | 117.4 | 90.8 | 116.9 | 101.5 | 130.6 | 108.7 | 140.2 | 92.7 | 120.6 | 92.5 | 118.9 |
| | 336 | 96.4 | 117.4 | 88.3 | 107.5 | 94.6 | 123.0 | 100.7 | 131.2 | 92.2 | 118.5 | 94.9 | 123.0 | 103.2 | 132.7 | 109.0 | 140.6 | 95.1 | 122.6 | 95.6 | 122.5 |
| | 720 | - | - | - | - | - | - | - | - | - | - | - | - | - | - | - | - | - | - | - | - |
| | Avg. | 93.9 | 114.7 | 86.1 | 105.3 | 93.6 | 121.2 | 98.1 | 128.1 | 91.2 | 117.2 | 91.6 | 117.8 | 101.2 | 130.3 | 108.3 | 139.8 | 102.0 | 131.3 | 93.8 | 120.8 |
| Surface Temperature | 96 | 83.5 | 94.2 | 80.2 | 93.2 | 176.9 | 204.5 | 179.6 | 208.8 | 82.9 | 107.0 | 83.6 | 102.2 | 74.1 | 97.5 | 188.2 | 229.5 | 98.0 | 126.6 | 91.8 | 115.9 |
| | 192 | 92.3 | 101.3 | 92.7 | 99.6 | 180.1 | 207.5 | 183.4 | 212.7 | 88.2 | 112.9 | 89.6 | 109.4 | 87.8 | 114.0 | 194.8 | 239.3 | 98.3 | 128.1 | 93.1 | 119.4 |
| | 336 | 95.1 | 102.7 | 96.1 | 100.6 | 182.7 | 210.2 | 185.6 | 215.6 | 92.0 | 117.2 | 103.1 | 118.0 | 99.1 | 127.6 | 196.9 | 240.9 | 98.6 | 129.4 | 94.8 | 122.2 |
| | 720 | - | - | - | - | - | - | - | - | - | - | - | - | - | - | - | - | - | - | - | - |
| | Avg. | 90.3 | 99.4 | 89.7 | 97.8 | 179.9 | 207.4 | 182.9 | 212.4 | 87.7 | 112.3 | 92.1 | 109.9 | 87.0 | 113.0 | 193.3 | 236.6 | 98.3 | 128.1 | 93.2 | 119.1 |
| All | 96 | 88.1 | 95.1 | 87.3 | 93.9 | 166.9 | 296.0 | 173.6 | 299.2 | 92.4 | 187.5 | 85.1 | 182.7 | 103.3 | 204.8 | 185.8 | 328.3 | 93.7 | 193.5 | 91.1 | 190.0 |
| | 192 | 90.2 | 98.4 | 89.6 | 96.5 | 166.9 | 297.3 | 176.0 | 303.0 | 94.4 | 192.4 | 90.7 | 191.6 | 106.7 | 210.7 | 188.1 | 336.0 | 96.8 | 200.1 | 93.5 | 195.9 |
| | 336 | 94.2 | 101.7 | 92.2 | 99.7 | 168.9 | 297.5 | 177.6 | 303.0 | 95.9 | 193.2 | 96.5 | 197.4 | 108.7 | 211.8 | 188.4 | 334.6 | 100.2 | 203.3 | 99.3 | 201.0 |
| | 720 | - | - | - | - | - | - | - | - | - | - | - | - | - | - | - | - | - | - | - | - |
| | Avg. | 90.8 | 98.4 | 89.7 | 96.7 | 167.7 | 296.9 | 175.7 | 301.7 | 94.2 | 191.0 | 90.7 | 190.6 | 106.3 | 209.1 | 187.5 | 333.0 | 96.9 | 199.0 | 94.6 | 195.6 |
| 1st Count | | 5 | | **21** | | 0 | | 0 | | 3 | | 7 | | 3 | | 0 | | 1 | | 0 | |

Experimental results indicate that our LM-WEATHER significantly outperforms the baseline in resource-constrained situations, such as few-shot learning environments with limited training data. This suggests that LM-WEATHER effectively leverages PLMs for sequential data modeling and achieves commendable performance without requiring extensive data for training.

Table 37: Comparison of the performance of LM-WEATHER with baselines on the **ODW1T** under the forecasting task in a scenario where the proportion of training data is set to be 15% in the few-shot learning. **Bold**: the best, Underline: the second best, "-" denotes insufficient training data.

| Method | | LM-WEATHER-AVE | | LM-WEATHER | | FL-Reformer | | FL-Pyraformer | | FL-DLinear | | FL-PatchTST | | FL-iTransformer | | FL-Lights | | FL-Transformer | | FL-Informer | |
|---|---|---|---|---|---|---|---|---|---|---|---|---|---|---|---|---|---|---|---|---|---|
| Variable | Length | MAE | RMSE | MAE | RMSE | MAE | RMSE | MAE | RMSE | MAE | RMSE | MAE | RMSE | MAE | RMSE | MAE | RMSE | MAE | RMSE | MAE | RMSE |
| Temperature | 96 | 53.9 | 70.6 | **52.3** | **66.6** | 181.8 | 205.4 | 168.4 | 191.7 | 74.4 | 93.2 | 52.7 | 69.6 | 59.9 | 77.6 | 143.1 | 170.2 | 102.2 | 124.9 | 80.8 | 102.1 |
| | 192 | 60.2 | 78.4 | **57.4** | **73.9** | 192.7 | 216.7 | 184.7 | 208.7 | 77.3 | 97.5 | 60.3 | 78.7 | 69.2 | 89.0 | 185.6 | 222.9 | 106.0 | 130.2 | 90.5 | 114.2 |
| | 336 | 66.2 | 84.5 | **62.4** | **79.4** | 200.4 | 223.9 | 202.9 | 226.6 | 80.6 | 101.1 | 66.4 | 108.1 | 77.6 | 98.5 | 210.4 | 252.0 | 99.5 | 122.8 | 95.5 | 119.3 |
| | 720 | 69.7 | 86.2 | **65.7** | **80.9** | 220.2 | 240.9 | 220.7 | 243.6 | 94.2 | 116.7 | 83.7 | 104.7 | 95.6 | 119.7 | 236.1 | 277.8 | 103.6 | 127.0 | 108.7 | 133.8 |
| | Avg. | 62.5 | 79.9 | **59.5** | **75.2** | 198.8 | 221.7 | 194.2 | 217.6 | 81.6 | 102.1 | 65.8 | 90.3 | 75.6 | 96.2 | 193.8 | 230.7 | 102.8 | 126.2 | 93.9 | 117.3 |
| Humidity | 96 | 65.3 | 85.1 | 63.7 | **82.7** | 96.2 | 114.2 | 89.8 | 107.9 | 74.7 | 93.0 | **63.2** | 83.5 | 69.7 | 90.7 | 91.3 | 112.4 | 87.1 | 108.0 | 86.0 | 106.0 |
| | 192 | 68.4 | 89.9 | **66.6** | **87.4** | 95.0 | 113.4 | 93.7 | 112.3 | 81.0 | 100.7 | 68.9 | 90.3 | 77.3 | 99.3 | 99.5 | 122.8 | 90.3 | 111.2 | 88.5 | 108.5 |
| | 336 | 70.7 | 91.5 | **69.7** | **88.8** | 94.6 | 112.9 | 95.7 | 114.7 | 85.1 | 105.0 | 72.5 | 93.7 | 84.3 | 106.6 | 102.8 | 126.7 | 88.7 | 108.8 | 89.9 | 110.2 |
| | 720 | 72.9 | 93.8 | **71.6** | **91.0** | 97.5 | 115.2 | 97.6 | 117.4 | 89.7 | 109.7 | 78.2 | 98.9 | 93.9 | 117.2 | 107.1 | 131.6 | 89.9 | 109.7 | 91.3 | 112.1 |
| | Avg. | 69.3 | 90.1 | **67.9** | **87.5** | 95.8 | 113.9 | 94.2 | 113.0 | 82.6 | 102.1 | 70.7 | 91.6 | 81.3 | 103.4 | 100.2 | 123.4 | 89.0 | 109.4 | 88.9 | 109.2 |
| Wind speed | 96 | 79.1 | 102.8 | **77.4** | **100.7** | 86.3 | 113.0 | 86.7 | 113.1 | 84.1 | 108.6 | 78.9 | 103.8 | 87.3 | 113.3 | 97.2 | 125.7 | 88.0 | 115.9 | 87.3 | 113.5 |
| | 192 | 81.8 | 109.9 | 81.3 | 107.0 | 86.8 | 113.3 | 88.3 | 114.8 | 86.6 | 111.5 | **81.3** | **106.7** | 92.1 | 118.9 | 100.2 | 129.2 | 87.5 | 115.5 | 87.9 | 113.9 |
| | 336 | 85.0 | 113.3 | 83.1 | 109.2 | 87.1 | 113.6 | 89.5 | 116.4 | 88.1 | 113.2 | **83.2** | **109.0** | 95.6 | 123.1 | 101.2 | 130.5 | 87.2 | 114.7 | 88.1 | 114.1 |
| | 720 | 87.2 | 114.9 | **85.6** | **111.8** | 90.2 | 115.5 | 91.0 | 118.2 | 89.6 | 115.1 | 87.5 | 113.9 | 99.3 | 127.5 | 102.5 | 131.8 | 88.1 | 115.8 | 89.2 | 115.7 |
| | Avg. | 83.3 | 110.2 | **81.9** | **107.2** | 87.6 | 113.8 | 88.9 | 115.6 | 87.1 | 112.1 | 82.7 | 108.3 | 93.6 | 120.7 | 100.3 | 129.3 | 87.7 | 115.5 | 88.1 | 114.3 |
| Surface Temperature | 96 | 43.2 | 59.6 | 42.7 | 58.5 | 135.0 | 162.6 | 122.6 | 148.9 | 60.4 | 80.6 | **42.4** | **58.9** | 51.0 | 68.0 | 128.3 | 157.5 | 83.9 | 108.6 | 80.9 | 102.7 |
| | 192 | 47.9 | 65.4 | **47.3** | **64.1** | 140.0 | 167.6 | 136.5 | 164.6 | 69.6 | 90.9 | 47.3 | 64.9 | 58.2 | 77.2 | 150.3 | 185.9 | 83.9 | 109.3 | 83.6 | 108.2 |
| | 336 | 50.7 | 68.4 | **50.0** | **67.2** | 143.8 | 171.0 | 147.6 | 175.5 | 76.0 | 97.9 | 52.0 | 70.1 | 67.6 | 88.7 | 161.0 | 198.3 | 82.7 | 107.8 | 85.3 | 110.0 |
| | 720 | 52.2 | 70.4 | **51.9** | **68.2** | 160.2 | 181.6 | 156.4 | 183.9 | 84.9 | 109.0 | 64.2 | 84.2 | 86.4 | 111.5 | 172.7 | 210.2 | 85.8 | 111.1 | 90.2 | 115.8 |
| | Avg. | 48.5 | 65.9 | **48.0** | **64.5** | 144.8 | 170.7 | 140.8 | 168.2 | 72.7 | 94.6 | 51.5 | 69.5 | 65.8 | 86.3 | 153.1 | 188.0 | 84.1 | 109.2 | 85.0 | 109.2 |
| All | 96 | 58.4 | 99.4 | **57.7** | **97.8** | 118.1 | 175.0 | 123.3 | 180.3 | 70.2 | 109.5 | 59.2 | 100.2 | 68.4 | 109.9 | 114.4 | 171.9 | 68.1 | 111.9 | 68.1 | 109.2 |
| | 192 | 65.7 | 102.5 | **64.7** | **100.4** | 119.3 | 177.5 | 127.5 | 184.8 | 71.0 | 111.3 | 63.4 | 105.5 | 73.7 | 116.8 | 125.2 | 190.8 | 71.0 | 114.1 | 69.8 | 112.1 |
| | 336 | 66.5 | 104.6 | **65.5** | **101.8** | 120.9 | 179.1 | 130.6 | 187.3 | 72.2 | 112.3 | 66.4 | 108.1 | 77.0 | 120.1 | 130.2 | 198.7 | 72.1 | 114.7 | 71.3 | 113.4 |
| | 720 | 70.7 | 111.2 | **69.2** | **108.2** | 127.8 | 184.2 | 133.6 | 189.0 | 77.1 | 117.8 | 74.5 | 116.5 | 83.4 | 127.0 | 136.6 | 206.0 | 76.7 | 118.8 | 76.2 | 118.1 |
| | Avg. | 65.3 | 104.4 | **64.3** | **102.1** | 121.5 | 179.0 | 128.8 | 185.4 | 72.6 | 112.7 | 65.9 | 107.6 | 75.6 | 118.4 | 126.6 | 191.8 | 72.4 | 114.9 | 71.3 | 113.2 |
| 1st Count | | 0 | | **44** | | 0 | | 0 | | 0 | | 7 | | 0 | | 0 | | 0 | | 0 | |

Table 38: Comparison of the performance of LM-WEATHER with the baseline on the **ODW1T** under the imputation task in a scenario where the proportion of training data is set to be 5% in the few-shot learning. **Bold**: the best, Underline: the second best, "-" denotes insufficient training data.

| Method | | LM-WEATHER-AVE | | LM-WEATHER | | FL-Reformer | | FL-Pyraformer | | FL-DLinear | | FL-PatchTST | | FL-iTransformer | | FL-LightTS | | FL-Transformer | | FL-Informer | |
|---|---|---|---|---|---|---|---|---|---|---|---|---|---|---|---|---|---|---|---|---|---|
| Ratio | Length | MAE | RMSE | MAE | RMSE | MAE | RMSE | MAE | RMSE | MAE | RMSE | MAE | RMSE | MAE | RMSE | MAE | RMSE | MAE | RMSE | MAE | RMSE |
| 25% | 96 | 56.1 | 111.4 | **52.6** | **109.2** | 137.4 | 242.2 | 133.6 | 227.9 | 87.5 | 176.7 | 56.8 | 134.9 | 98.8 | 198.1 | 175.2 | 317.1 | 130.8 | 248.0 | 133.8 | 253.5 |
| | 192 | 60.1 | 114.3 | **57.6** | **112.0** | 142.0 | 248.7 | 145.6 | 245.1 | 89.4 | 173.4 | 69.5 | 148.4 | 104.5 | 197.9 | 176.3 | 310.6 | 140.1 | 251.7 | 141.1 | 254.5 |
| | 336 | - | - | - | - | - | - | - | - | - | - | - | - | - | - | - | - | - | - | - | - |
| | 720 | - | - | - | - | - | - | - | - | - | - | - | - | - | - | - | - | - | - | - | - |
| | Avg. | 58.1 | 112.9 | **55.1** | **110.6** | 139.7 | 245.5 | 139.6 | 236.5 | 88.5 | 175.0 | 63.1 | 141.7 | 101.7 | 198.0 | 175.8 | 313.9 | 135.5 | 249.9 | 137.5 | 254.0 |
| 35% | 96 | 61.7 | 120.4 | **58.2** | **119.7** | 141.2 | 245.2 | 139.6 | 231.3 | 95.8 | 188.3 | 59.7 | 139.8 | 106.7 | 210.1 | 174.4 | 315.0 | 134.4 | 252.4 | 137.5 | 257.5 |
| | 192 | 64.5 | 126.5 | **60.9** | **122.3** | 142.6 | 247.1 | 141.3 | 234.8 | 97.0 | 184.3 | 72.4 | 156.8 | 110.7 | 207.7 | 174.7 | 307.1 | 143.5 | 255.6 | 144.6 | 257.9 |
| | 336 | - | - | - | - | - | - | - | - | - | - | - | - | - | - | - | - | - | - | - | - |
| | 720 | - | - | - | - | - | - | - | - | - | - | - | - | - | - | - | - | - | - | - | - |
| | Avg. | 63.1 | 123.5 | **59.6** | **121.0** | 141.9 | 246.2 | 140.5 | 233.1 | 96.4 | 186.3 | 66.1 | 148.3 | 108.7 | 208.9 | 174.6 | 311.0 | 138.9 | 254.0 | 141.1 | 257.7 |
| 50% | 96 | 63.1 | 122.4 | **62.1** | **119.4** | 147.4 | 261.3 | 149.5 | 256.4 | 110.0 | 209.1 | 64.2 | 147.0 | 119.0 | 228.5 | 173.0 | 310.8 | 140.8 | 260.0 | 143.9 | 264.7 |
| | 192 | 67.3 | 129.7 | **62.5** | **129.4** | 151.3 | 267.8 | 152 | 258.1 | 110.1 | 203.4 | 74.1 | 155.1 | 120.9 | 223.0 | 172.2 | 301.4 | 149.1 | 262.4 | 150.3 | 264.2 |
| | 336 | - | - | - | - | - | - | - | - | - | - | - | - | - | - | - | - | - | - | - | - |
| | 720 | - | - | - | - | - | - | - | - | - | - | - | - | - | - | - | - | - | - | - | - |
| | Avg. | 65.2 | 125.7 | **62.3** | **124.4** | 149.4 | 264.6 | 150.8 | 257.3 | 110.0 | 206.2 | 69.2 | 151.0 | 120.0 | 225.7 | 172.6 | 306.1 | 145.0 | 261.2 | 147.1 | 264.4 |
| 1st Count | | 0 | | **18** | | 0 | | 0 | | 0 | | 0 | | 0 | | 0 | | 0 | | 0 | |

Table 39: Comparison of the performance of the LM-WEATHER with the baseline on the **ODW1T** under the imputation task in a scenario where the proportion of training data is set to be 15% in the few-shot learning. **Bold**: the best, Underline: the second best, "-" denotes insufficient training data.

| Method | | LM-WEATHER-AVE | | LM-WEATHER | | FL-Reformer | | FL-Pyraformer | | FL-DLinear | | FL-PatchTST | | FL-iTransformer | | FL-LightTS | | FL-Transformer | | FL-Informer | |
|---|---|---|---|---|---|---|---|---|---|---|---|---|---|---|---|---|---|---|---|---|---|
| Ratio | Length | MAE | RMSE | MAE | RMSE | MAE | RMSE | MAE | RMSE | MAE | RMSE | MAE | RMSE | MAE | RMSE | MAE | RMSE | MAE | RMSE | MAE | RMSE |
| 25% | 96 | 30.8 | 58.8 | **30.1** | **54.2** | 76.3 | 121.9 | 74.1 | 120.3 | 62.6 | 96.6 | 31.6 | 60.0 | 63.5 | 101.6 | 113.3 | 173.7 | 72.4 | 115.3 | 73.3 | 117.0 |
| | 192 | 36.0 | 64.8 | **35.0** | **61.2** | 77.4 | 124.9 | 75.4 | 122.1 | 65.3 | 99.2 | 37.4 | 66.9 | 71.5 | 110.6 | 117.5 | 179.2 | 72.6 | 114.0 | 73.2 | 115.6 |
| | 336 | 42.0 | 71.2 | **41.0** | **68.6** | 80.2 | 130.2 | 79.9 | 126.0 | 67.3 | 100.5 | 43.7 | 73.8 | 76.3 | 115.0 | 121.1 | 181.8 | 72.5 | 112.0 | 72.8 | 113.4 |
| | 720 | 54.4 | 84.3 | **52.6** | **80.2** | 84.1 | 136.6 | 82.1 | 132.2 | 68.6 | 99.9 | 56.2 | 86.3 | 79.0 | 114.8 | 115.0 | 169.3 | 76.1 | 112.1 | 76.0 | 113.1 |
| | Avg. | 40.8 | 69.8 | **39.7** | **66.1** | 79.5 | 128.4 | 77.9 | 125.2 | 66.0 | 99.1 | 42.2 | 71.8 | 72.6 | 110.5 | 116.7 | 176.0 | 73.4 | 113.4 | 73.8 | 114.8 |
| 35% | 96 | 32.4 | 60.8 | **31.3** | **58.2** | 78.1 | 123.8 | 77.0 | 120.4 | 95.8 | 188.3 | 33.4 | 62.5 | 71.0 | 112.1 | 115.8 | 177.0 | 73.9 | 117.1 | 75.1 | 119.2 |
| | 192 | 38.0 | 67.3 | **36.8** | **62.5** | 79.4 | 125.9 | 78.1 | 122.4 | 70.7 | 106.2 | 39.3 | 69.4 | 77.1 | 118.7 | 118.6 | 180.3 | 74.2 | 115.9 | 75.1 | 117.9 |
| | 336 | 44.1 | 73.7 | **42.9** | **70.4** | 82.1 | 129.4 | 80.3 | 126.3 | 72.2 | 106.9 | 45.8 | 76.4 | 80.6 | 121.2 | 121.0 | 181.3 | 74.3 | 114.1 | 75.1 | 116.0 |
| | 720 | 55.9 | 85.6 | **54.0** | **82.5** | 85.0 | 134.6 | 83.2 | 130.4 | 72.3 | 104.8 | 57.7 | 88.1 | 81.5 | 118.8 | 113.9 | 167.4 | 78.8 | 115.3 | 79.1 | 116.7 |
| | Avg. | 42.6 | 71.9 | **41.3** | **68.4** | 81.2 | 128.4 | 79.7 | 124.9 | 77.8 | 126.6 | 44.0 | 74.1 | 77.6 | 117.7 | 117.3 | 176.5 | 75.3 | 115.6 | 76.1 | 117.4 |
| 50% | 96 | 35.4 | 65.6 | **34.3** | **62.5** | 81.4 | 128.3 | 80.9 | 123.2 | 79.0 | 118.1 | 36.7 | 67.2 | 82.4 | 128.0 | 118.8 | 180.4 | 76.3 | 119.9 | 78.3 | 123.3 |
| | 192 | 41.7 | 71.7 | **40.2** | **68.2** | 83.0 | 130.7 | 82.1 | 127.8 | 80.3 | 119.3 | 43.0 | 74.2 | 86.0 | 131.2 | 119.7 | 180.8 | 76.8 | 119.1 | 78.7 | 122.3 |
| | 336 | 48.0 | 78.6 | **46.5** | **75.9** | 86.2 | 134.2 | 84.9 | 130.2 | 80.8 | 118.7 | 49.7 | 81.0 | 87.5 | 131.1 | 120.4 | 179.6 | 77.5 | 117.9 | 79.2 | 121.0 |
| | 720 | 58.1 | 87.8 | **56.6** | **84.8** | 90.1 | 139.4 | 89.9 | 134.4 | 78.9 | 115.4 | 59.8 | 90.7 | 85.9 | 125.4 | 112.3 | 164.4 | 83.9 | 121.6 | 84.8 | 123.5 |
| | Avg. | 45.8 | 75.9 | **44.3** | **72.9** | 85.2 | 133.2 | 84.5 | 128.9 | 79.7 | 117.5 | 47.3 | 78.3 | 85.4 | 128.9 | 117.8 | 176.3 | 78.6 | 119.6 | 80.3 | 122.5 |
| 1st Count | | 0 | | **30** | | 0 | | 0 | | 0 | | 0 | | 0 | | 0 | | 0 | | 0 | |

Table 40: Comparison of the performance of LM-WEATHER with the baseline method on the **ODW1V** under the long-term forecasting task in a scenario where the proportion of training data is set to be 5% in the few-shot learning. **Bold**: the best, Underline: the second best, "-" denotes insufficient training data.

| Method | | LM-WEATHER-AVE | | LM-WEATHER | | FL-DLinear | | FL-PatchTST | | FL-iTransformer | | FL-LightTS | | FL-Transformer | | FL-Informer | | FL-Reformer | | FL-Pyraformer | |
|---|---|---|---|---|---|---|---|---|---|---|---|---|---|---|---|---|---|---|---|---|---|
| Metrics | Length | MAE | RMSE | MAE | RMSE | MAE | RMSE | MAE | RMSE | MAE | RMSE | MAE | RMSE | MAE | RMSE | MAE | RMSE | MAE | RMSE | MAE | RMSE |
| Temperature | 96 | 74.1 | 95.8 | 71.1 | 90.0 | 94.0 | 118.9 | 75.6 | 98.8 | 91.5 | 117.0 | 303.3 | 363.3 | 125.0 | 155.4 | 102.8 | 130.0 | 108.0 | 143.0 | 103.6 | 133.0 |
| | 192 | 81.8 | 104.9 | 77.3 | 98.6 | 99.1 | 125.2 | 83.5 | 108.1 | 102.2 | 130.0 | 309.6 | 372.7 | 134.3 | 165.5 | 108.8 | 137.2 | 114.3 | 150.9 | 107.2 | 140.3 |
| | 336 | 95.9 | 121.3 | 90.6 | 113.8 | 103.9 | 130.2 | 98.3 | 125.0 | 111.1 | 140.0 | 316.1 | 381.3 | 112.9 | 141.0 | 116.8 | 146.6 | 128.1 | 161.3 | 119.2 | 150.0 |
| | 720 | - | - | - | - | - | - | - | - | - | - | - | - | - | - | - | - | - | - | - | - |
| | Avg. | 83.9 | 107.3 | 79.7 | 100.8 | 99.0 | 124.8 | 85.8 | 110.6 | 101.6 | 129.0 | 309.7 | 372.4 | 124.0 | 154.0 | 109.5 | 138.0 | 116.8 | 151.7 | 110.0 | 141.1 |
| All | 96 | 79.6 | 104.3 | 75.7 | 98.1 | 101.5 | 130.2 | 81.6 | 107.5 | 98.8 | 127.4 | 327.6 | 392.4 | 135.0 | 168.3 | 111.0 | 141.6 | 116.6 | 155.8 | 111.5 | 144.9 |
| | 192 | 87.8 | 115.5 | 82.5 | 108.4 | 107.0 | 136.7 | 90.2 | 118.9 | 110.4 | 141.6 | 334.4 | 403.4 | 145.4 | 180.2 | 117.5 | 149.2 | 123.4 | 164.1 | 116.0 | 152.4 |
| | 336 | 103.9 | 133.4 | 98.7 | 125.4 | 113.2 | 142.4 | 106.1 | 137.5 | 120.0 | 153.2 | 341.6 | 413.7 | 122.1 | 153.5 | 126.3 | 159.7 | 133.6 | 161.3 | 123.2 | 167.4 |
| | 720 | - | - | - | - | - | - | - | - | - | - | - | - | - | - | - | - | - | - | - | - |
| | Avg. | 90.4 | 117.7 | 85.6 | 110.6 | 107.2 | 136.4 | 92.6 | 121.3 | 109.7 | 140.7 | 334.5 | 403.2 | 134.2 | 167.3 | 118.3 | 150.2 | 124.5 | 160.4 | 116.9 | 154.9 |
| 1st Count | | 0 | | 16 | | 0 | | 0 | | 0 | | 0 | | 0 | | 0 | | 0 | | 0 | |

Table 41: Comparison of the performance of LM-WEATHER with baselines on the **ODW1V** under the forecasting task in a scenario where the proportion of training data is set to be 15% in the few-shot learning. **Bold**: the best, Underline: the second best, "-" denotes insufficient training data.

| Method | | LM-WEATHER-AVE | | LM-WEATHER | | FL-DLinear | | FL-PatchTST | | FL-iTransformer | | FL-LightTS | | FL-Transformer | | FL-Informer | | FL-Reformer | | FL-Pyraformer | |
|---|---|---|---|---|---|---|---|---|---|---|---|---|---|---|---|---|---|---|---|---|---|
| Metrics | Length | MAE | RMSE | MAE | RMSE | MAE | RMSE | MAE | RMSE | MAE | RMSE | MAE | RMSE | MAE | RMSE | MAE | RMSE | MAE | RMSE | MAE | RMSE |
| Temperature | 96 | 52.2 | 67.5 | 51.1 | 65.5 | 74.4 | 93.2 | 52.7 | 69.6 | 59.9 | 77.6 | 143.2 | 170.7 | 83.9 | 105.1 | 75.6 | 95.8 | 78.7 | 102.0 | 80.3 | 105.1 |
| | 192 | 59.1 | 76.3 | 57.7 | 74.1 | 77.3 | 97.5 | 60.3 | 78.7 | 69.2 | 88.9 | 184.3 | 220.5 | 95.4 | 118.4 | 83.8 | 106.2 | 87.2 | 113.3 | 89.0 | 116.7 |
| | 336 | 64.5 | 82.4 | 62.8 | 80.1 | 80.6 | 101.1 | 66.1 | 84.9 | 77.5 | 98.4 | 211.0 | 253.0 | 92.2 | 114.6 | 88.2 | 110.8 | 92.1 | 117.9 | 94.0 | 121.4 |
| | 720 | 81.4 | 101.6 | 79.2 | 98.6 | 94.2 | 116.7 | 83.7 | 104.7 | 95.5 | 119.6 | 236.3 | 278.3 | 100.5 | 123.5 | 104.0 | 128.8 | 108.2 | 137.9 | 110.4 | 142.0 |
| | Avg. | 64.3 | 82.0 | 62.7 | 79.6 | 81.6 | 102.1 | 65.7 | 84.5 | 75.5 | 96.1 | 193.7 | 230.7 | 93.0 | 115.4 | 87.9 | 110.4 | 91.6 | 117.8 | 93.4 | 121.3 |
| All | 96 | 56.0 | 73.5 | 54.7 | 71.3 | 79.7 | 101.7 | 56.9 | 75.8 | 64.1 | 84.5 | 153.4 | 185.5 | 89.8 | 114.5 | 81.6 | 104.5 | 85.0 | 109.7 | 86.7 | 113.0 |
| | 192 | 63.4 | 83.2 | 61.7 | 80.7 | 83.1 | 106.2 | 65.1 | 85.8 | 74.2 | 97.0 | 197.4 | 240.1 | 102.4 | 129.1 | 90.5 | 115.9 | 94.3 | 124.1 | 96.2 | 127.8 |
| | 336 | 69.6 | 89.9 | 67.6 | 87.2 | 86.8 | 110.2 | 71.4 | 92.7 | 83.0 | 107.4 | 226.2 | 275.7 | 103.5 | 135.1 | 95.3 | 121.0 | 99.5 | 129.5 | 101.9 | 133.4 |
| | 720 | 88.3 | 110.7 | 85.9 | 107.4 | 101.3 | 127.2 | 90.4 | 114.1 | 102.2 | 130.4 | 253.9 | 303.5 | 108.0 | 134.6 | 112.3 | 140.5 | 116.9 | 150.3 | 119.5 | 154.8 |
| | Avg. | 69.3 | 89.3 | 67.5 | 86.7 | 87.7 | 111.3 | 71.0 | 92.1 | 80.9 | 104.8 | 207.7 | 251.2 | 100.9 | 128.3 | 94.9 | 120.5 | 98.9 | 128.4 | 101.1 | 132.3 |
| 1st Count | | 0 | | 20 | | 0 | | 0 | | 0 | | 0 | | 0 | | 0 | | 0 | | 0 | |

Table 42: Comparison of the performance of LM-WEATHER with baselines on the **ODW1V** under the imputation task in a scenario where the proportion of training data is set to be 5% in the few-shot learning. **Bold**: the best, Underline: the second best, "-" denotes insufficient training data.

| Method | | LM-WEATHER-AVE | | LM-WEATHER | | FL-DLinear | | FL-PatchTST | | FL-iTransformer | | FL-LightTS | | FL-Transformer | | FL-Informer | | FL-Reformer | | FL-Pyraformer | |
|---|---|---|---|---|---|---|---|---|---|---|---|---|---|---|---|---|---|---|---|---|---|
| Ratio | Length | MAE | RMSE | MAE | RMSE | MAE | RMSE | MAE | RMSE | MAE | RMSE | MAE | RMSE | MAE | RMSE | MAE | RMSE | MAE | RMSE | MAE | RMSE |
| 25% | 96 | 54.7 | 122.2 | 55.3 | 123.6 | 83.4 | 159.3 | 54.2 | 120.6 | 93.6 | 179.6 | 163.0 | 288.0 | 107.6 | 207.3 | 109.9 | 209.9 | 88.5 | 187.7 | 85.1 | 178.1 |
| | 192 | 67.1 | 134.7 | 67.8 | 136.3 | 84.5 | 155.3 | 66.4 | 132.9 | 98.4 | 177.5 | 164.3 | 280.1 | 114.3 | 205.1 | 112.9 | 205.2 | 91.6 | 180.2 | 87.2 | 171.0 |
| | 336 | - | - | - | - | - | - | - | - | - | - | - | - | - | - | - | - | - | - | - | - |
| | 720 | - | - | - | - | - | - | - | - | - | - | - | - | - | - | - | - | - | - | - | - |
| | Avg. | 60.9 | 128.5 | 55.3 | 130.0 | 84.0 | 157.3 | 60.3 | 126.8 | 96.0 | 178.5 | 163.6 | 284.0 | 111.0 | 206.2 | 111.4 | 207.5 | 90.1 | 184.0 | 86.2 | 174.6 |
| 35% | 96 | 60.9 | 128.3 | 61.5 | 129.8 | 90.9 | 171.0 | 57.0 | 125.3 | 100.8 | 190.9 | 162.3 | 285.2 | 111.1 | 211.5 | 113.4 | 215.2 | 91.3 | 189.8 | 87.6 | 180.3 |
| | 192 | 61.6 | 126.6 | 62.2 | 128.0 | 91.4 | 165.0 | 68.1 | 135.4 | 103.9 | 186.3 | 162.8 | 277.1 | 118.0 | 210.4 | 116.5 | 209.7 | 94.6 | 183.4 | 90.1 | 174.0 |
| | 336 | - | - | - | - | - | - | - | - | - | - | - | - | - | - | - | - | - | - | - | - |
| | 720 | - | - | - | - | - | - | - | - | - | - | - | - | - | - | - | - | - | - | - | - |
| | Avg. | 61.3 | 127.5 | 61.9 | 128.9 | 91.2 | 168.0 | 62.6 | 130.4 | 102.4 | 188.6 | 162.5 | 281.2 | 114.6 | 211.0 | 114.9 | 212.4 | 93.0 | 186.6 | 88.9 | 177.2 |
| 50% | 96 | 62.2 | 134.1 | 62.8 | 135.5 | 103.9 | 189.8 | 61.5 | 132.6 | 112.2 | 208.4 | 161.1 | 281.5 | 117.6 | 219.5 | 119.6 | 223.5 | 98.0 | 198.5 | 94.2 | 188.1 |
| | 192 | 71.4 | 140.5 | 72.2 | 142.1 | 103.3 | 182.4 | 70.6 | 138.8 | 113.3 | 200.6 | 160.5 | 272.4 | 124.5 | 218.9 | 122.7 | 217.8 | 101.8 | 191.3 | 96.7 | 181.5 |
| | 336 | - | - | - | - | - | - | - | - | - | - | - | - | - | - | - | - | - | - | - | - |
| | 720 | - | - | - | - | - | - | - | - | - | - | - | - | - | - | - | - | - | - | - | - |
| | Avg. | 66.8 | 137.3 | 67.5 | 138.8 | 103.6 | 186.1 | 66.1 | 135.7 | 112.7 | 204.5 | 160.8 | 276.9 | 121.0 | 219.2 | 121.2 | 220.7 | 101.8 | 194.9 | 95.5 | 184.8 |
| 1st Count | | 4 | | 0 | | 0 | | 14 | | 0 | | 0 | | 0 | | 0 | | 0 | | 0 | |

Table 43: Comparison of the performance of LM-WEATHER with baselines on the **ODW1V** under the imputation task in a scenario where the proportion of training data is set to be 15% in the few-shot learning. **Bold**: the best, Underline: the second best.

| Method | | LM-WEATHER-AVE | | LM-WEATHER | | FL-DLinear | | FL-PatchTST | | FL-iTransformer | | FL-LightTS | | FL-Transformer | | FL-Informer | | FL-Reformer | | FL-Pyraformer | |
|---|---|---|---|---|---|---|---|---|---|---|---|---|---|---|---|---|---|---|---|---|---|
| Ratio | Length | MAE | RMSE | MAE | RMSE | MAE | RMSE | MAE | RMSE | MAE | RMSE | MAE | RMSE | MAE | RMSE | MAE | RMSE | MAE | RMSE | MAE | RMSE |
| 25% | 96 | 30.1 | 56.7 | 29.7 | 55.4 | 60.2 | 91.2 | 30.4 | 57.4 | 60.4 | 95.9 | 107.1 | 164.1 | 66.3 | 108.4 | 69.7 | 113.2 | 72.6 | 117.7 | 54.3 | 90.3 |
| | 192 | 35.5 | 62.3 | 34.8 | 61.0 | 62.3 | 93.3 | 35.9 | 63.1 | 67.7 | 103.8 | 111.3 | 169.3 | 65.4 | 106.1 | 68.2 | 110.1 | 71.0 | 114.5 | 52.8 | 87.0 |
| | 336 | 41.6 | 69.4 | 40.6 | 67.8 | 64.2 | 95.5 | 42.1 | 70.3 | 74.8 | 110.2 | 115.1 | 174.4 | 64.7 | 104.9 | 66.5 | 107.8 | 69.3 | 112.2 | 51.3 | 84.5 |
| | 720 | 53.1 | 82.2 | 52.0 | 80.4 | 65.3 | 96.1 | 53.7 | 83.3 | 74.8 | 110.2 | 108.1 | 161.7 | 62.1 | 98.5 | 61.9 | 98.9 | 64.5 | 102.9 | 55.2 | 90.1 |
| | Avg. | 41.6 | 67.7 | 39.3 | 66.2 | 63.0 | 94.0 | 40.5 | 68.5 | 69.4 | 105.0 | 110.4 | 167.4 | 64.6 | 104.5 | 66.6 | 107.5 | 71.0 | 111.8 | 53.4 | 88.0 |
| 35% | 96 | 31.8 | 58.8 | 31.4 | 57.6 | 65.7 | 98.2 | 32.1 | 59.5 | 67.5 | 105.9 | 109.3 | 167.2 | 68.9 | 111.7 | 73.0 | 117.6 | 75.9 | 122.5 | 57.3 | 93.8 |
| | 192 | 37.2 | 64.5 | 36.6 | 63.0 | 67.3 | 99.9 | 37.6 | 65.3 | 73.0 | 111.6 | 112.3 | 170.4 | 68.2 | 109.6 | 71.5 | 114.5 | 74.4 | 119.3 | 55.4 | 89.9 |
| | 336 | 43.6 | 71.7 | 42.8 | 70.0 | 68.7 | 101.6 | 44.1 | 72.6 | 76.3 | 115.0 | 114.9 | 173.7 | 67.6 | 108.6 | 69.8 | 112.2 | 72.7 | 117.0 | 53.8 | 87.2 |
| | 720 | 54.5 | 83.9 | 53.3 | 82.0 | 68.7 | 101.6 | 55.1 | 85.0 | 76.9 | 113.8 | 107.1 | 159.9 | 65.3 | 102.8 | 65.2 | 103.5 | 67.9 | 107.8 | 57.4 | 92.1 |
| | Avg. | 41.8 | 73.4 | 41.0 | 68.2 | 67.6 | 100.3 | 42.2 | 70.6 | 73.4 | 111.6 | 110.9 | 167.8 | 67.5 | 108.2 | 69.9 | 111.9 | 73.6 | 114.7 | 56.0 | 90.8 |
| 50% | 96 | 34.8 | 62.9 | 34.1 | 61.4 | 75.6 | 111.6 | 35.2 | 63.7 | 78.3 | 120.0 | 112.1 | 170.3 | 72.9 | 116.2 | 78.3 | 124.5 | 81.4 | 129.7 | 62.8 | 100.6 |
| | 192 | 40.6 | 68.8 | 39.9 | 67.1 | 76.3 | 112.4 | 41.0 | 69.7 | 81.4 | 123.5 | 113.4 | 171.0 | 72.6 | 114.8 | 76.8 | 121.6 | 80.0 | 126.7 | 61.3 | 101.2 |
| | 336 | 47.1 | 75.9 | 46.0 | 74.1 | 76.7 | 112.8 | 47.7 | 76.9 | 80.7 | 119.9 | 114.3 | 172.0 | 72.4 | 114.6 | 75.3 | 119.5 | 78.4 | 124.6 | 58.4 | 92.5 |
| | 720 | 56.4 | 86.2 | 55.0 | 84.2 | 74.2 | 108.9 | 57.1 | 87.4 | 80.7 | 119.9 | 105.5 | 156.9 | 71.0 | 109.9 | 71.0 | 110.9 | 73.8 | 115.5 | 64.2 | 102.4 |
| | Avg. | 44.7 | 73.5 | 43.8 | 71.7 | 75.7 | 111.4 | 45.3 | 74.4 | 80.3 | 121.1 | 111.3 | 167.5 | 72.2 | 113.9 | 75.4 | 119.1 | 78.4 | 124.1 | 61.7 | 99.2 |
| 1st Count | | 0 | | 30 | | 0 | | 0 | | 0 | | 0 | | 0 | | 0 | | 0 | | 0 | |

Table 44: Comparison of the performance of LM-WEATHER with baselines on the **ODW2T** under the forecasting task in a scenario where the proportion of training data is set to be 5% in the few-shot learning. **Bold**: the best, Underline: the second best, "-" denotes insufficient training data.

| Method | | LM-WEATHER-AVE | | LM-WEATHER | | FL-Reformer | | FL-Pyraformer | | FL-DLinear | | FL-PatchTST | | FL-iTransformer | | FL-LightTS | | FL-Transformer | | FL-Informer | |
|---|---|---|---|---|---|---|---|---|---|---|---|---|---|---|---|---|---|---|---|---|---|
| Variable | Length | MAE | RMSE | MAE | RMSE | MAE | RMSE | MAE | RMSE | MAE | RMSE | MAE | RMSE | MAE | RMSE | MAE | RMSE | MAE | RMSE | MAE | RMSE |
| Temperature | 96 | 103.6 | 129.1 | **92.2** | **116.2** | 254.5 | 278.5 | 254.2 | 279.6 | 98.8 | 122.9 | 83.4 | 105.9 | 103.5 | 129.7 | 263.0 | 311.5 | 101.7 | 125.2 | 250.8 | 275.0 |
| | 192 | - | - | - | - | - | - | - | - | - | - | - | - | - | - | - | - | - | - | - | - |
| | 336 | - | - | - | - | - | - | - | - | - | - | - | - | - | - | - | - | - | - | - | - |
| | 720 | - | - | - | - | - | - | - | - | - | - | - | - | - | - | - | - | - | - | - | - |
| | Avg. | 103.6 | 129.1 | **92.2** | **116.2** | 254.5 | 278.5 | 254.2 | 279.6 | 98.8 | 122.9 | 83.4 | 105.9 | 103.5 | 129.7 | 263.0 | 311.5 | 101.7 | 125.2 | 250.8 | 275.0 |
| Humidity | 96 | 110.2 | 134.1 | 97.2 | 122.0 | 128.6 | 153.0 | 132.5 | 158.9 | 106.6 | 128.7 | **96.7** | **120.0** | 113.3 | 139.6 | 143.7 | 175.3 | 119.0 | 145.4 | 131.4 | 157.5 |
| | 192 | - | - | - | - | - | - | - | - | - | - | - | - | - | - | - | - | - | - | - | - |
| | 336 | - | - | - | - | - | - | - | - | - | - | - | - | - | - | - | - | - | - | - | - |
| | 720 | - | - | - | - | - | - | - | - | - | - | - | - | - | - | - | - | - | - | - | - |
| | Avg. | 110.2 | 134.1 | 97.2 | 122.0 | 128.6 | 153.0 | 132.5 | 158.9 | 106.6 | 128.7 | **96.7** | **120.0** | 113.3 | 139.6 | 143.7 | 175.3 | 119.0 | 145.4 | 131.4 | 157.5 |
| All | 96 | 111.0 | 159.4 | **99.0** | **135.5** | 158.3 | 241.2 | 173.3 | 247.1 | 107.1 | 152.8 | 101.2 | 147.9 | 115.9 | 166.3 | 183.6 | 273.3 | 142.3 | 199.6 | 158.8 | 201.3 |
| | 192 | - | - | - | - | - | - | - | - | - | - | - | - | - | - | - | - | - | - | - | - |
| | 336 | - | - | - | - | - | - | - | - | - | - | - | - | - | - | - | - | - | - | - | - |
| | 720 | - | - | - | - | - | - | - | - | - | - | - | - | - | - | - | - | - | - | - | - |
| | Avg. | 111.0 | 159.4 | **99.0** | **135.5** | 158.3 | 241.2 | 173.3 | 247.1 | 107.1 | 152.8 | 101.2 | 147.9 | 115.9 | 166.3 | 183.6 | 273.3 | 142.3 | 199.6 | 158.8 | 201.3 |
| 1st Count | | 0 | | **4** | | 0 | | 0 | | 0 | | 2 | | 0 | | 0 | | 0 | | 0 | |

Table 45: Comparison of the performance of LM-WEATHER with baselines on the **ODW2T** under the forecasting task in a scenario where the proportion of training data is set to be 15% in the few-shot learning. **Bold**: the best, Underline: the second best, "-" denotes insufficient training data.

| Method | | LM-WEATHER-AVE | | LM-WEATHER | | FL-Reformer | | FL-Pyraformer | | FL-DLinear | | FL-PatchTST | | FL-iTransformer | | FL-LightTS | | FL-Transformer | | FL-Informer | |
|---|---|---|---|---|---|---|---|---|---|---|---|---|---|---|---|---|---|---|---|---|---|
| Variable | Length | MAE | RMSE | MAE | RMSE | MAE | RMSE | MAE | RMSE | MAE | RMSE | MAE | RMSE | MAE | RMSE | MAE | RMSE | MAE | RMSE | MAE | RMSE |
| Temperature | 96 | 68.0 | 85.1 | 66.6 | **84.5** | 240.0 | 260.3 | 211.1 | 233.4 | 88.2 | 108.1 | **66.2** | 88.2 | 70.2 | 89.4 | 221.5 | 254.3 | 100.2 | 123.2 | 236.5 | 258.1 |
| | 192 | 73.7 | 93.3 | **71.7** | **90.7** | 234.9 | 255.2 | 225.6 | 247.8 | 86.1 | 106.7 | 78.5 | 94.8 | 79.2 | 100.0 | 239.2 | 281.8 | 120.3 | 144.3 | 241.9 | 263.4 |
| | 336 | 91.0 | **103.0** | 86.7 | 109.7 | 235.3 | 255.5 | 234.2 | 255.7 | 89.1 | 110.7 | 89.5 | 109.2 | **88.9** | 111.7 | 250.2 | 295.8 | 133.2 | 160.7 | 249.1 | 270.3 |
| | 720 | - | - | - | - | - | - | - | - | - | - | - | - | - | - | - | - | - | - | - | - |
| | Avg. | 77.5 | **93.8** | 76.7 | 97.1 | 236.7 | 257.0 | 223.6 | 245.6 | 87.8 | 108.5 | 78.1 | 97.4 | 79.5 | 100.3 | 237.0 | 277.3 | 117.9 | 142.7 | 242.5 | 263.9 |
| Humidity | 96 | 80.7 | 95.5 | 78.9 | 98.5 | 110.4 | 131.6 | 102.6 | 123.9 | 89.4 | 109.5 | 88.2 | 109.6 | 82.9 | 104.1 | 109.5 | 133.1 | 98.1 | 117.7 | 112.8 | 134.4 |
| | 192 | 103.5 | 117.8 | 100.4 | 114.0 | 109.1 | 130.0 | 108.0 | 129.7 | 98.4 | **111.4** | 99.2 | 118.9 | **89.6** | 111.4 | 118.3 | 144.2 | 105.2 | 128.4 | 113.0 | 134.6 |
| | 336 | 104.6 | 121.2 | 102.4 | **116.4** | 109.3 | 130.2 | 111.0 | 132.7 | 109.8 | 119.5 | 104.3 | 129.0 | **96.6** | 119.3 | 122.8 | 149.8 | 102.9 | 125.2 | 112.7 | 134.1 |
| | 720 | - | - | - | - | - | - | - | - | - | - | - | - | - | - | - | - | - | - | - | - |
| | Avg. | 96.3 | 111.5 | 93.9 | 109.6 | **109.6** | 130.6 | 107.2 | 128.8 | 99.2 | 113.5 | 97.2 | 119.2 | **89.7** | 111.6 | 116.9 | 142.4 | 102.1 | 123.7 | 112.8 | 134.4 |
| All | 96 | 84.2 | 124.7 | **82.3** | **121.0** | 142.2 | 167.9 | 146.7 | 209.7 | 97.1 | 136.6 | 85.5 | 123.7 | 91.3 | 132.9 | 153.1 | 223.5 | 112.5 | 143.8 | 174.4 | 200.4 |
| | 192 | 92.1 | 135.9 | 89.7 | **131.8** | 158.8 | 177.3 | 150.6 | 214.8 | 98.4 | 140.1 | **89.5** | 133.1 | 100.5 | 146.1 | 161.4 | 238.8 | 123.1 | 150.0 | 176.3 | 205.7 |
| | 336 | 95.4 | 141.3 | **92.9** | **137.1** | 182.5 | 190.2 | 153.8 | 218.6 | 101.0 | 146.4 | 98.7 | 146.8 | 107.4 | 156.9 | 166.7 | 246.9 | 130.4 | 158.5 | 188.2 | 211.1 |
| | 720 | - | - | - | - | - | - | - | - | - | - | - | - | - | - | - | - | - | - | - | - |
| | Avg. | 90.6 | 134.0 | **86.0** | **129.1** | 161.2 | 178.5 | 150.4 | 214.4 | 98.8 | 140.8 | 91.2 | 134.5 | 99.7 | 145.3 | 160.4 | 236.4 | 122.0 | 150.8 | 179.6 | 205.7 |
| 1st Count | | 2 | | **14** | | 1 | | 0 | | 1 | | 2 | | 4 | | 0 | | 0 | | 0 | |

Table 46: Comparison of the performance of LM-WEATHER with the baseline on the **ODW2T** under the imputation task in a scenario where the proportion of training data is set to be 5% in the few-shot learning. **Bold**: the best, Underline: the second best, "-" denotes insufficient training data.

| Method | | LM-WEATHER-AVE | | LM-WEATHER | | FL-DLinear | | FL-PatchTST | | FL-iTransformer | | FL-LightTS | | FL-Transformer | | FL-Informer | | FL-Reformer | | FL-Pyraformer | |
|---|---|---|---|---|---|---|---|---|---|---|---|---|---|---|---|---|---|---|---|---|---|
| Ratio | Length | MAE | RMSE | MAE | RMSE | MAE | RMSE | MAE | RMSE | MAE | RMSE | MAE | RMSE | MAE | RMSE | MAE | RMSE | MAE | RMSE | MAE | RMSE |
| 25% | 96 | 92.7 | 143.4 | **91.3** | 139.2 | 95.7 | **131.2** | 113.2 | 147.6 | 104.6 | 149.0 | 173.6 | 260.5 | 115.7 | 175.2 | 121.5 | 183.3 | 111.3 | 170.1 | 94.8 | 147.8 |
| | 192 | - | - | - | - | - | - | - | - | - | - | - | - | - | - | - | - | - | - | - | - |
| | 336 | - | - | - | - | - | - | - | - | - | - | - | - | - | - | - | - | - | - | - | - |
| | 720 | - | - | - | - | - | - | - | - | - | - | - | - | - | - | - | - | - | - | - | - |
| | Avg. | 92.7 | 143.4 | **91.3** | 139.2 | 95.7 | **131.2** | 113.2 | 147.6 | 104.6 | 149.0 | 173.6 | 260.5 | 115.7 | 175.2 | 121.5 | 183.3 | 111.3 | 170.1 | 94.8 | 147.8 |
| 35% | 96 | 95.6 | 147.2 | **94.1** | 144.2 | 103.2 | **141.8** | 116.3 | 151.2 | 112.0 | 160.5 | 173.4 | 259.1 | 119.9 | 180.8 | 126.0 | 189.5 | 115.9 | 176.3 | 98.0 | 151.7 |
| | 192 | - | - | - | - | - | - | - | - | - | - | - | - | - | - | - | - | - | - | - | - |
| | 336 | - | - | - | - | - | - | - | - | - | - | - | - | - | - | - | - | - | - | - | - |
| | 720 | - | - | - | - | - | - | - | - | - | - | - | - | - | - | - | - | - | - | - | - |
| | Avg. | 95.6 | 147.2 | **94.1** | 144.2 | 103.2 | **141.8** | 116.3 | 151.2 | 112 | 160.5 | 173.4 | 259.1 | 119.9 | 180.8 | 126.0 | 189.5 | 115.9 | 176.3 | 98.0 | 151.7 |
| 50% | 96 | 102.5 | 156.3 | **99.4** | **151.6** | 116.2 | 161.3 | 124.9 | 165.6 | 123.7 | 178.3 | 173.0 | 256.7 | 127.3 | 190.6 | 133.8 | 200.3 | 124.3 | 187.5 | 105.7 | 161.1 |
| | 192 | - | - | - | - | - | - | - | - | - | - | - | - | - | - | - | - | - | - | - | - |
| | 336 | - | - | - | - | - | - | - | - | - | - | - | - | - | - | - | - | - | - | - | - |
| | 720 | - | - | - | - | - | - | - | - | - | - | - | - | - | - | - | - | - | - | - | - |
| | Avg. | 102.5 | 156.3 | **99.4** | **151.6** | 116.2 | 161.3 | 124.9 | 165.6 | 123.7 | 178.3 | 173 | 256.7 | 127.3 | 190.6 | 133.8 | 200 | 124.3 | 188 | 105.7 | 161.0 |
| 1st Count | | 0 | | **8** | | 4 | | 0 | | 0 | | 0 | | 0 | | 0 | | 0 | | 0 | |

Table 47: Comparison of the performance of LM-WEATHER with the baselines on the **ODW2T** under the imputation task in a scenario where the proportion of training data is set to be 15% in the few-shot learning. **Bold**: the best, Underline: the second best, "-" denotes insufficient training data.

| Method | | LM-WEATHER-AVE | | LM-WEATHER | | FL-DLinear | | FL-PatchTST | | FL-iTransformer | | FL-LightTS | | FL-Transformer | | FL-Informer | | FL-Reformer | | FL-Pyraformer | |
|---|---|---|---|---|---|---|---|---|---|---|---|---|---|---|---|---|---|---|---|---|---|
| | Length | MAE | RMSE | MAE | RMSE | MAE | RMSE | MAE | RMSE | MAE | RMSE | MAE | RMSE | MAE | RMSE | MAE | RMSE | MAE | RMSE | MAE | RMSE |
| 25% | 96 | 72.5 | 105.3 | **71.7** | **103.7** | 81.1 | 111.1 | 85.1 | 118.7 | 76.7 | 111.8 | 137.7 | 204.5 | 89.9 | 140.3 | 94.4 | 146.7 | 85.6 | 134.9 | 76.8 | 122.8 |
| | 192 | 74.6 | 108.5 | **73.7** | **106.8** | 88.3 | 120.6 | 94.4 | 129.0 | 92.2 | 130.1 | 150.7 | 223.5 | 93.2 | 143.4 | 97.0 | 148.8 | 87.4 | 136.1 | 77.0 | 122.2 |
| | 336 | 75.1 | 117.5 | **74.2** | **115.6** | 91.7 | 125.1 | 97.8 | 133.7 | 102.0 | 142.3 | 160.1 | 236.5 | 95.7 | 145.4 | 98.4 | 149.5 | 88.2 | 136.0 | 77.3 | 121.1 |
| | 720 | - | - | - | - | - | - | - | - | - | - | - | - | - | - | - | - | - | - | - | - |
| | Avg. | 74.1 | 110.4 | **73.2** | **108.7** | 87.0 | 119.0 | 94.4 | 131.4 | 90.3 | 128.1 | 149.5 | 221.5 | 92.9 | 143.0 | 96.6 | 148.3 | 87.1 | 135.7 | 77.0 | 122.0 |
| 35% | 96 | 74.3 | 103.4 | **73.6** | **102.0** | 87.8 | 120.2 | 94.0 | 121.7 | 86.2 | 125.6 | 141.2 | 209.5 | 92.2 | 143.2 | 98.5 | 152.3 | 88.0 | 137.9 | 79.8 | 126.8 |
| | 192 | 77.6 | 110.9 | **76.6** | **109.4** | 94.6 | 129.3 | 100.8 | 138.4 | 99.5 | 141.0 | 151.6 | 224.3 | 96.2 | 147.5 | 101.2 | 154.9 | 90.4 | 140.0 | 79.3 | 125.3 |
| | 336 | 80.2 | 115.8 | 79.2 | 114.3 | 97.6 | 133.0 | 104.4 | 142.4 | 107.2 | 150.2 | 159.2 | 234.3 | 99.6 | 151.0 | 102.6 | 155.8 | 92.2 | 141.5 | **79.2** | 123.8 |
| | 720 | - | - | - | - | - | - | - | - | - | - | - | - | - | - | - | - | - | - | - | - |
| | Avg. | 77.4 | 110.0 | **76.5** | **108.6** | 93.3 | 127.5 | 94.0 | 134.2 | 97.6 | 138.9 | 150.7 | 222.7 | 96.0 | 147.2 | 100.8 | 154.3 | 90.2 | 139.8 | 79.4 | 125.3 |
| 50% | 96 | 74.9 | 105.3 | **74.1** | **103.7** | 100.0 | 138.0 | 105.5 | 147.7 | 101.2 | 147.1 | 145.4 | 214.9 | 96.1 | 147.7 | 105.4 | 161.7 | 92.2 | 143.1 | 85.0 | 133.7 |
| | 192 | 78.1 | 114.2 | **77.2** | **112.6** | 105.8 | 145.5 | 112.7 | 155.7 | 110.7 | 157.8 | 152.6 | 224.2 | 101.6 | 154.4 | 108.5 | 165.2 | 95.9 | 147.3 | 83.4 | 130.9 |
| | 336 | 84.1 | 119.0 | 83.1 | **117.3** | 108.0 | 148.1 | 115.6 | 158.5 | 115.6 | 163.2 | 157.6 | 230.5 | 106.6 | 160.6 | 110.9 | 166.2 | 99.5 | 151.6 | **82.7** | 128.7 |
| | 720 | - | - | - | - | - | - | - | - | - | - | - | - | - | - | - | - | - | - | - | - |
| | Avg. | 79.0 | 112.8 | **78.6** | **111.2** | 104.6 | 143.8 | 111.3 | 154.0 | 109.2 | 156.0 | 151.8 | 223.2 | 101.5 | 154.2 | 107.9 | 164.3 | 95.9 | 147.3 | 83.7 | 131.1 |
| 1st Count | | 0 | | **17** | | 0 | | 0 | | 0 | | 0 | | 0 | | 0 | | 0 | | 1 | |

Table 48: Comparison of the performance of LM-WEATHER with the baseline on the **ODW2V** under forecasting tasks in a scenario where the proportion of training data is set to be 5% in the few-shot learning. **Bold**: the best, Underline: the second best, "-" denotes insufficient training data.

| Method | | LM-WEATHER-AVE | | LM-WEATHER | | FL-Reformer | | FL-Pyraformer | | FL-Dlinear | | FL-PatchTST | | FL-iTransformer | | FL-LightTS | | FL-Transformer | | FL-Informer | |
|---|---|---|---|---|---|---|---|---|---|---|---|---|---|---|---|---|---|---|---|---|---|
| Variable | Length | MAE | RMSE | MAE | RMSE | MAE | RMSE | MAE | RMSE | MAE | RMSE | MAE | RMSE | MAE | RMSE | MAE | RMSE | MAE | RMSE | MAE | RMSE |
| Humidity | 96 | 109.6 | 133.5 | **99.7** | **124.3** | 130.7 | 155.2 | 130.1 | 155.8 | 109.6 | 138.7 | 116.7 | 140.0 | 113.3 | 139.7 | 146.2 | 178.9 | 118.3 | 144.0 | 128.4 | 153.3 |
| | 192 | - | - | - | - | - | - | - | - | - | - | - | - | - | - | - | - | - | - | - | - |
| | 336 | - | - | - | - | - | - | - | - | - | - | - | - | - | - | - | - | - | - | - | - |
| | 720 | - | - | - | - | - | - | - | - | - | - | - | - | - | - | - | - | - | - | - | - |
| | Avg. | 109.6 | 133.5 | **99.7** | **124.3** | 130.7 | 155.2 | 130.1 | 155.8 | 109.6 | 138.7 | 116.7 | 140.0 | 113.3 | 139.7 | 146.2 | 178.9 | 118.3 | 144.0 | 128.4 | 153.3 |
| All | 96 | 105.3 | 135.7 | **96.8** | **122.1** | 151.5 | 190.7 | 150.5 | 189.3 | 112.2 | 141.2 | 115.5 | 145.8 | 110.2 | 143.4 | 162.1 | 212.5 | 106.4 | 136.8 | 149.6 | 188.2 |
| | 192 | - | - | - | - | - | - | - | - | - | - | - | - | - | - | - | - | - | - | - | - |
| | 336 | - | - | - | - | - | - | - | - | - | - | - | - | - | - | - | - | - | - | - | - |
| | 720 | - | - | - | - | - | - | - | - | - | - | - | - | - | - | - | - | - | - | - | - |
| | Avg. | 105.3 | 135.7 | **96.8** | **122.1** | 151.5 | 190.7 | 150.5 | 189.3 | 112.2 | 141.2 | 115.5 | 145.8 | 110.2 | 143.4 | 162.1 | 212.5 | 106.4 | 136.8 | 149.6 | 188.2 |
| 1st Count | | 0 | | **8** | | 0 | | 0 | | 0 | | 0 | | 0 | | 0 | | 0 | | 0 | |

Table 49: Comparison of the performance of LM-WEATHER with baselines on the **ODW2V** under the long-term forecasting task in a scenario where proportion of training data is set to be 15% in the few-shot learning. **Bold**: the best, Underline: the second best, "-" denotes insufficient training data.

| Method | | LM-WEATHER-AVE | | LM-WEATHER | | FL-Reformer | | FL-Pyraformer | | FL-DLinear | | FL-PatchTST | | FL-iTransformer | | FL-LightTS | | FL-Transformer | | FL-Informer | |
|---|---|---|---|---|---|---|---|---|---|---|---|---|---|---|---|---|---|---|---|---|---|
| Variable | Length | MAE | RMSE | MAE | RMSE | MAE | RMSE | MAE | RMSE | MAE | RMSE | MAE | RMSE | MAE | RMSE | MAE | RMSE | MAE | RMSE | MAE | RMSE |
| Humidity | 96 | 79.1 | 99.9 | **75.7** | **94.9** | 111.1 | 132.5 | 100.7 | 121.0 | 87.4 | 102.4 | 96.2 | 96.6 | 83.0 | 104.2 | 110.3 | 134.1 | 100.5 | 124.1 | 112.1 | 133.3 |
| | 192 | 89.7 | 111.8 | **85.3** | **106.2** | 109.4 | 130.3 | 105.8 | 126.7 | 91.4 | 114.4 | 100.2 | 110.9 | 77.4 | 111.5 | 118.1 | 144.1 | 109.0 | 132.8 | 113.2 | 134.6 |
| | 336 | 105.7 | 120.8 | 100.4 | 114.8 | 109.3 | 130.0 | 109.8 | 131.0 | 97.8 | 118.5 | 102.3 | **113.0** | 106.6 | 119.4 | 122.3 | 149.2 | 106.5 | 129.8 | 113.5 | 135.3 |
| | 720 | - | - | - | - | - | - | - | - | - | - | - | - | - | - | - | - | - | - | - | - |
| | Avg. | 91.5 | 110.8 | **87.1** | **105.3** | 109.9 | 130.9 | 105.4 | 126.2 | 92.2 | 111.8 | 99.5 | 106.8 | 93.1 | 111.7 | 116.9 | 142.5 | 105.3 | 128.9 | 112.9 | 134.4 |
| All | 96 | 79.0 | 106.4 | **75.6** | **101.1** | 135.4 | 172.3 | 126.9 | 162.3 | 89.6 | 115.9 | 78.5 | 106.1 | 85.7 | 114.1 | 135.8 | 178.1 | 108.7 | 143.1 | 134.4 | 171.1 |
| | 192 | 87.8 | 116.6 | 83.6 | 110.8 | 133.7 | 169.9 | 131.9 | 167.6 | 90.8 | 117.4 | **82.6** | **110.5** | 92.7 | 122.1 | 141.9 | 187.2 | 107.4 | 141.5 | 134.4 | 170.8 |
| | 336 | 89.6 | 119.0 | **85.1** | **113.1** | 133.9 | 170.1 | 133.6 | 169.6 | 92.4 | 119.8 | 85.6 | 114.0 | 98.3 | 129.0 | 146.3 | 193.2 | 105.7 | 139.5 | 134.8 | 171.1 |
| | 720 | - | - | - | - | - | - | - | - | - | - | - | - | - | - | - | - | - | - | - | - |
| | Avg. | 85.5 | 114.0 | **81.4** | **108.3** | 134.3 | 170.8 | 130.8 | 166.5 | 90.9 | 117.7 | 82.2 | 110.2 | 92.2 | 121.7 | 141.3 | 186.2 | 107.3 | 141.4 | 134.6 | 171.0 |
| 1st Count | | 0 | | **12** | | 0 | | 0 | | 0 | | 0 | | 0 | | 0 | | 0 | | 0 | |

Table 50: Comparison of the performance of LM-WEATHER with baselines on the **ODW2V** under the imputation task in a scenario where the proportion of training data is set to be 5% in the few-shot learning. **Bold**: the best, Underline: the second best, "-" denotes insufficient training data.

| Method | | LM-WEATHER-AVE | | LM-WEATHER | | FL-DLinear | | FL-PatchTST | | FL-iTransformer | | FL-LightTS | | FL-Transformer | | FL-Informer | | FL-Reformer | | FL-Pyraformer | |
|---|---|---|---|---|---|---|---|---|---|---|---|---|---|---|---|---|---|---|---|---|---|
| Ratio | Length | MAE | RMSE | MAE | RMSE | MAE | RMSE | MAE | RMSE | MAE | RMSE | MAE | RMSE | MAE | RMSE | MAE | RMSE | MAE | RMSE | MAE | RMSE |
| 25% | 96 | 42.2 | 62.9 | **39.7** | **60.5** | 91.3 | 117.0 | 61.5 | 84.9 | 98.3 | 129.6 | 155.1 | 203.0 | 92.5 | 125.2 | 94.9 | 128.2 | 90.4 | 122.4 | 87.0 | 115.9 |
| | 192 | - | - | - | - | - | - | - | - | - | - | - | - | - | - | - | - | - | - | - | - |
| | 336 | - | - | - | - | - | - | - | - | - | - | - | - | - | - | - | - | - | - | - | - |
| | 720 | - | - | - | - | - | - | - | - | - | - | - | - | - | - | - | - | - | - | - | - |
| | Avg. | 42.2 | 62.9 | **39.7** | **60.5** | 91.3 | 117.0 | 61.5 | 84.9 | 98.3 | 129.6 | 155.1 | 203.0 | 92.5 | 125.2 | 94.9 | 128.2 | 90.4 | 122.4 | 87.0 | 115.9 |
| 35% | 96 | 41.5 | 63.0 | **38.5** | **61.2** | 96.8 | 123.5 | 65.0 | 88.9 | 104.2 | 136.7 | 154.7 | 201.8 | 95.7 | 129.1 | 98.9 | 133.3 | 93.7 | 126.6 | 88.1 | 117.1 |
| | 192 | - | - | - | - | - | - | - | - | - | - | - | - | - | - | - | - | - | - | - | - |
| | 336 | - | - | - | - | - | - | - | - | - | - | - | - | - | - | - | - | - | - | - | - |
| | 720 | - | - | - | - | - | - | - | - | - | - | - | - | - | - | - | - | - | - | - | - |
| | Avg. | 41.5 | 63.0 | **38.5** | **61.2** | 96.8 | 123.5 | 65.0 | 88.9 | 104.2 | 136.7 | 154.7 | 201.8 | 95.7 | 129.1 | 98.9 | 133.3 | 93.7 | 126.6 | 88.1 | 117.1 |
| 50% | 96 | 42.4 | 62.9 | **35.7** | 112.1 | 106.8 | 135.5 | 70.8 | **95.5** | 113.3 | 148.0 | 153.8 | 199.5 | 101.8 | 136.4 | 106.1 | 142.2 | 100.1 | 134.6 | 89.8 | 119.0 |
| | 192 | - | - | - | - | - | - | - | - | - | - | - | - | - | - | - | - | - | - | - | - |
| | 336 | - | - | - | - | - | - | - | - | - | - | - | - | - | - | - | - | - | - | - | - |
| | 720 | - | - | - | - | - | - | - | - | - | - | - | - | - | - | - | - | - | - | - | - |
| | Avg. | 42.4 | 62.9 | **35.7** | 112.1 | 106.8 | 135.5 | 70.8 | **95.5** | 113.3 | 148.0 | 153.8 | 199.5 | 101.8 | 136.4 | 106.1 | 142.2 | 100.1 | 134.6 | 89.8 | 119.0 |
| 1st Count | | 0 | | **10** | | 0 | | 2 | | 0 | | 0 | | 0 | | 0 | | 0 | | 0 | |

Table 51: Comparison of the performance of LM-WEATHER with baselines on the **ODW2V** under the imputation task in a scenario where the proportion of training data is set to be 15% in the few-shot learning. **Bold**: the best, Underline: the second best, "-" denotes insufficient training data.

| Method | | LM-WEATHER-ave | | LM-WEATHER | | FL-Dlinear | | FL-PatchTST | | FL-iTransformer | | FL-LightTS | | FL-Transformer | | FL-Informer | | FL-Reformer | | FL-Pyraformer | |
|---|---|---|---|---|---|---|---|---|---|---|---|---|---|---|---|---|---|---|---|---|---|
| Ratio | Length | MAE | RMSE | MAE | RMSE | MAE | RMSE | MAE | RMSE | MAE | RMSE | MAE | RMSE | MAE | RMSE | MAE | RMSE | MAE | RMSE | MAE | RMSE |
| 25% | 96 | 35.2 | 54.2 | **33.4** | **52.9** | 77.3 | 101.0 | 40.9 | 60.9 | 70.6 | 96.0 | 125.5 | 166.4 | 74.7 | 104.0 | 77.1 | 107.6 | 72.6 | 101.3 | 63.7 | 88.8 |
| | 192 | 36.7 | 56.3 | **34.3** | **53.4** | 83.9 | 108.2 | 47.5 | 68.1 | 86.1 | 114.0 | 134.8 | 179.1 | 77.4 | 106.9 | 80.5 | 114.6 | 74.6 | 103.5 | 64.9 | 90.3 |
| | 336 | 41.2 | 60.2 | **40.0** | **59.1** | 87.9 | 113.0 | 59.5 | 81.9 | 96.8 | 126.8 | 144.9 | 191.6 | 82.7 | 113.5 | 84.0 | 115.3 | 77.8 | 107.6 | 71.9 | 98.7 |
| | 720 | - | - | - | - | - | - | - | - | - | - | - | - | - | - | - | - | - | - | - | - |
| | Avg. | 37.7 | 56.9 | **35.9** | **55.1** | 83.0 | 107.4 | 49.3 | 70.3 | 84.5 | 112.3 | 135.1 | 179.0 | 78.3 | 108.1 | 80.5 | 122.5 | 75.0 | 104.1 | 66.8 | 92.6 |
| 35% | 96 | 36.4 | 55.8 | **36.0** | **55.1** | 82.7 | 107.2 | 42.7 | 63.1 | 78.8 | 106.3 | 128.2 | 169.5 | 77.0 | 106.7 | 80.4 | 111.7 | 74.5 | 103.4 | 122.4 | 157.9 |
| | 192 | 36.9 | 57.0 | **35.4** | **56.1** | 88.9 | 114.1 | 49.5 | 70.5 | 92.2 | 121.8 | 135.6 | 179.5 | 79.9 | 109.9 | 84.2 | 116.2 | 76.6 | 106.0 | 69.4 | 96.1 |
| | 336 | 43.8 | 64.6 | **41.2** | **63.5** | 92.4 | 118.5 | 62.3 | 84.9 | 100.9 | 132.1 | 144.1 | 189.9 | 86.0 | 117.8 | 87.9 | 120.5 | 80.7 | 111.2 | 73.4 | 100.4 |
| | 720 | - | - | - | - | - | - | - | - | - | - | - | - | - | - | - | - | - | - | - | - |
| | Avg. | 39.0 | 59.1 | **37.5** | **58.2** | 88.0 | 113.3 | 51.5 | 72.9 | 90.6 | 120.0 | 136.0 | 179.7 | 81.0 | 111.4 | 84.2 | 116.1 | 77.3 | 106.9 | 88.4 | 118.1 |
| 50% | 96 | 36.8 | 56.4 | **32.7** | **55.5** | 92.6 | 119.2 | 46.0 | 67.1 | 91.8 | 122.3 | 131.4 | 172.8 | 80.6 | 110.4 | 86.3 | 118.8 | 77.7 | 106.9 | 84.1 | 111.5 |
| | 192 | 37.4 | 57.9 | **36.7** | **56.8** | 97.7 | 124.9 | 53.6 | 75.4 | 101.6 | 133.5 | 136.3 | 179.3 | 84.3 | 115.1 | 90.8 | 124.7 | 80.6 | 110.7 | 83.8 | 112.5 |
| | 336 | 44.2 | 65.7 | **43.4** | **64.4** | 100.7 | 128.6 | 67.4 | 90.7 | 107.5 | 140.4 | 142.7 | 187.1 | 92.2 | 125.5 | 94.8 | 129.4 | 86.2 | 118.2 | 80.8 | 109.2 |
| | 720 | - | - | - | - | - | - | - | - | - | - | - | - | - | - | - | - | - | - | - | - |
| | Avg. | 39.5 | 60.0 | **37.6** | **58.9** | 97.0 | 124.2 | 55.7 | 77.7 | 100.3 | 132.1 | 136.8 | 179.7 | 85.7 | 117.0 | 90.6 | 124.3 | 81.5 | 112.0 | 82.9 | 111.1 |
| 1st Count | | 0 | | **24** | | 0 | | 0 | | 0 | | 0 | | 0 | | 0 | | 0 | | 0 | |

## E.3 Full Ablation Experiments

In this subsection, we show the results of the complete ablation experiment, both in the forecasting (**Table. 52**) and in imputation (**Table. 53**).

Table 52: Ablation experimental (forecasting) results for both the model composition level and the personalization mechanism level are included, where ↑ represent the degree of performance increase relative to the original LM-WEATHER, ↓ represent the degree of performance degradation, and the Comm. Param# represents the number of parameters transferred between client and server communication for the different variants. **Bold**: the best, Underline: the second best.

| Method | | Original | | Model Composition Perspective | | | | | | | | | | Personalized Perspective | | | | | |
|---|---|---|---|---|---|---|---|---|---|---|---|---|---|---|---|---|---|---|---|
| | | LM-WEATHER | | LM-WEATHER-A | | LM-WEATHER-B | | LM-WEATHER-C | | LM-WEATHER-D | | LM-WEATHER-E | | LM-WEATHER-F | | LM-WEATHER-G | | LM-WEATHER-H | |
| Variables | Length | MAE | RMSE | MAE | RMSE | MAE | RMSE | MAE | RMSE | MAE | RMSE | MAE | RMSE | MAE | RMSE | MAE | RMSE | MAE | RMSE |
| Temperature | 96 | 26.6 | 34.7 | 29.5 | 39.9 | 28.8 | 38.2 | 28.5 | 37.8 | 27.7 | 36.5 | 31.0 | 43.7 | 28.8 | 38.3 | 24.2 | 32.4 | 24 | 30.9 |
| | 192 | 30.4 | 36.5 | 34.1 | 42 | 33.2 | 40.2 | 32.9 | 39.6 | 32.0 | 38.2 | 35.9 | 45.8 | 33 | 40.2 | 28.0 | 32.5 | 27.9 | 32.6 |
| | 336 | 31.6 | 38.0 | 35.6 | 44 | 35 | 42.3 | 34.3 | 41.5 | 33.3 | 40.0 | 38.5 | 48.1 | 43.5 | 41.9 | 29.4 | 34.2 | 29.2 | 34.1 |
| | 720 | 37.2 | 45.4 | 42.1 | 52.9 | 41.3 | 50.8 | 40.7 | 50.2 | 39.3 | 48.6 | 44.2 | 57.8 | 40.5 | 50.2 | 35.1 | 42.0 | 33.8 | 42.6 |
| | Avg. | 31.5 | 38.7 | 35.3 | 44.7 | 34.6 | 42.9 | 34.1 | 42.3 | 33.1 | 40.8 | 37.4 | 48.9 | 36.5 | 42.7 | 29.2 | 35.3 | 28.7 | 35.1 |
| Humidity | 96 | 49.9 | 64.0 | 56.2 | 75.1 | 54.7 | 72.3 | 54.1 | 71.2 | 52.7 | 68.6 | 59.0 | 81.9 | 53.8 | 70.5 | 48.1 | 61.5 | 47.8 | 61.4 |
| | 192 | 53.0 | 67.8 | 60.5 | 80 | 59.3 | 78 | 58.3 | 75.8 | 57.6 | 73.9 | 63.6 | 87.4 | 58.4 | 75.0 | 51.7 | 65.3 | 51.3 | 64.7 |
| | 336 | 55.6 | 70.1 | 63.5 | 82.7 | 62.3 | 81.1 | 61.0 | 78.5 | 60.0 | 76.6 | 66.7 | 90.4 | 60.6 | 77.4 | 54.1 | 67.2 | 53.7 | 66.3 |
| | 720 | 59.0 | 73.1 | 67.3 | 86.7 | 66.1 | 85.2 | 64.8 | 82.0 | 63.5 | 80.0 | 70.7 | 94.8 | 64.3 | 82.0 | 57.4 | 71.1 | 56.7 | 69.4 |
| | Avg. | 54.4 | 68.8 | 61.9 | 81.1 | 60.6 | 79.2 | 59.6 | 76.9 | 58.5 | 74.8 | 65.0 | 88.6 | 59.3 | 76.2 | 52.8 | 66.3 | 52.4 | 65.5 |
| Wind speed | 96 | 65.7 | 84.0 | 73.8 | 99.7 | 72.2 | 96.6 | 71.5 | 94.6 | 70.8 | 92.1 | 78.1 | 108.9 | 70.8 | 92.6 | 64.2 | 81.8 | 63.9 | 81.4 |
| | 192 | 66.7 | 85.0 | 75.5 | 101.0 | 74.2 | 98.2 | 73.2 | 95.7 | 73.2 | 93.3 | 80.0 | 110.3 | 72.5 | 93.9 | 65.5 | 82.7 | 65.1 | 82.1 |
| | 336 | 67.2 | 85.7 | 76.3 | 101.7 | 75.1 | 99 | 74.0 | 96.4 | 74.0 | 94.0 | 81.0 | 9.3 | 73.4 | 94.7 | 66 | 83.2 | 65.6 | 82.5 |
| | 720 | 68.0 | 86.5 | 77.2 | 102.7 | 76.1 | 100 | 74.9 | 97.3 | 74.8 | 94.9 | 82.7 | 112.1 | 74.3 | 95.7 | 66.7 | 83.9 | 66.3 | 83.3 |
| | Avg. | 66.9 | 85.3 | 75.7 | 101.3 | 74.4 | 98.5 | 73.4 | 96.0 | 73.2 | 93.6 | 80.5 | 85.2 | 72.8 | 94.2 | 65.6 | 82.9 | 65.2 | 82.3 |
| Surface Temperature | 96 | 27.0 | 37.4 | 29.5 | 43.0 | 29.1 | 41.4 | 28.8 | 40.7 | 28.2 | 39.5 | 31.6 | 44.8 | 29.1 | 41.1 | 24.4 | 33.3 | 24.3 | 34.2 |
| | 192 | 29.0 | 39.9 | 31.7 | 46.0 | 32.4 | 43.6 | 31.0 | 43.8 | 31.3 | 42.6 | 34.1 | 50.2 | 31.4 | 44.2 | 26.3 | 36.1 | 26.1 | 37 |
| | 336 | 31.1 | 42.4 | 34.1 | 49.2 | 35.1 | 48.2 | 33.4 | 46.5 | 33.7 | 45.4 | 36.3 | 51.0 | 33.6 | 46.7 | 28.3 | 38.3 | 28.1 | 38.9 |
| | 720 | 36.8 | 47.9 | 41 | 55.3 | 41.4 | 54.5 | 39.9 | 52.7 | 39.7 | 51.0 | 43.1 | 57.9 | 40 | 52.7 | 34.5 | 44.5 | 33.2 | 44.8 |
| | Avg. | 31.0 | 41.9 | 34.1 | 48.4 | 34.5 | 46.9 | 33.3 | 45.9 | 33.2 | 44.6 | 36.3 | 51.0 | 33.5 | 46.2 | 28.4 | 38.1 | 27.9 | 38.7 |
| All | 96 | 42.3 | 71.1 | 47.0 | 83.2 | 46.2 | 81.2 | 45.7 | 79.4 | 45.2 | 77.7 | 50.0 | 90.5 | 45.7 | 78.3 | 40.0 | 68.1 | 39.8 | 68.3 |
| | 192 | 44.4 | 73.6 | 49.4 | 86.3 | 49.7 | 84.3 | 48.9 | 82.3 | 48.1 | 80.4 | 52.8 | 94.1 | 48.3 | 81.0 | 42.3 | 70.4 | 42 | 70.4 |
| | 336 | 45.8 | 75.2 | 51.4 | 88.6 | 52 | 86.7 | 51.3 | 84.5 | 50.4 | 82.6 | 54.5 | 96.7 | 50.0 | 82.9 | 43.6 | 71.9 | 43.2 | 71.8 |
| | 720 | 49.2 | 78.5 | 55.5 | 92.3 | 55.5 | 90 | 54.6 | 88.2 | 53.6 | 86.2 | 57.9 | 101.1 | 53.6 | 86.8 | 47.0 | 75.2 | 45.6 | 74.2 |
| | Avg. | 45.4 | 74.6 | 50.8 | 87.6 | 50.9 | 85.6 | 50.1 | 83.6 | 49.3 | 81.7 | 53.8 | 95.6 | 49.4 | 82.3 | 43.2 | 71.4 | 42.7 | 71.2 |
| Average | | 45.8 | 61.9 | 51.6 | 72.6 | 51.0 | 70.6 | 50.1 | 68.9 | 49.5 | 67.1 | 54.6 | 73.9 | 50.3 | 68.4 | 43.8 | 58.8 | 43.4 | 58.6 |
| Change | | - | - | 12.70%↓ | 17.3%↓ | 11.4%↓ | 14.1%↓ | 9.4%↓ | 11.3%↓ | 8.1%↓ | 8.4%↓ | 19.2%↓ | 19.4%↓ | 9.8%↓ | 10.5%↓ | 4.6%↑ | 5.3%↑ | 5.5%↑ | 5.6%↑ |
| Comm. Param. # | | 0.38 M | | 0.38 M | | 0.38 M | | 0.38 M | | 0.38 M | | 0.38 M | | 10.00 M | | 41.99 M | | 10.00 M | |

Table 53: Ablation experimental results (imputation) for both the model composition level and the personalization mechanism level are included, where ↑ represent the degree of performance increase relative to the original LM-WEATHER, ↓ represent the degree of performance degradation, and the Comm. Param# represents the number of parameters transferred between client and server communication for the different variants. **Bold**: the best, Underline: the second best.

| Method | | Original | | Model Composition Perspective | | | | | | | | | | Personalized Perspective | | | | | |
|---|---|---|---|---|---|---|---|---|---|---|---|---|---|---|---|---|---|---|---|
| | | LM-WEATHER | | LM-WEATHER-A | | LM-WEATHER-B | | LM-WEATHER-C | | LM-WEATHER-D | | LM-WEATHER-E | | LM-WEATHER-F | | LM-WEATHER-G | | LM-WEATHER-H | |
| Metrics | Length | MAE | RMSE | MAE | RMSE | MAE | RMSE | MAE | RMSE | MAE | RMSE | MAE | RMSE | MAE | RMSE | MAE | RMSE | MAE | RMSE |
| 25% | 96 | 20.2 | 38.6 | 21.8 | 43.4 | 22 | 42.8 | 21.4 | 41.8 | 21.0 | 41.5 | 22.0 | 42.7 | 23.1 | 47.2 | 19.4 | 36.9 | 19.2 | 35.8 |
| | 192 | 20.7 | 37.0 | 22.8 | 41.6 | 22.5 | 43.4 | 21.9 | 42.8 | 21.5 | 41.6 | 22.8 | 42.9 | 23.8 | 45.3 | 19.8 | 34.2 | 19.6 | 34.1 |
| | 336 | 21.0 | 38.4 | 23.2 | 43.0 | 23 | 42.6 | 22.2 | 43.8 | 21.8 | 41.2 | 23.2 | 42.5 | 24.2 | 46.8 | 20.2 | 37.1 | 20.0 | 35.6 |
| | 720 | 21.5 | 39.3 | 23.8 | 44.0 | 23.5 | 43.6 | 22.9 | 42.7 | 22.4 | 43.2 | 23.8 | 43.4 | 24.9 | 47.9 | 20.6 | 36.3 | 20.4 | 36.1 |
| | Avg. | 21.1 | 38.3 | 22.9 | 43.0 | 22.8 | 43.1 | 22.1 | 42.8 | 21.7 | 41.9 | 23.0 | 42.9 | 24.0 | 46.8 | 20.0 | 36.1 | 19.8 | 35.4 |
| 35% | 96 | 21.1 | 39.8 | 23.4 | 44.8 | 23.2 | 44.2 | 22..4 | 43.4 | 22.0 | 43.0 | 23.4 | 44.2 | 24.5 | 49.1 | 20.3 | 35.3 | 20.1 | 36.8 |
| | 192 | 21.3 | 38.3 | 23.6 | 45.2 | 23.4 | 42.6 | 22.7 | 41.6 | 22.2 | 44.1 | 23.6 | 43.0 | 24.7 | 46.7 | 20.4 | 36.7 | 20.3 | 37.1 |
| | 336 | 22.1 | 39.5 | 24.6 | 44.4 | 24.7 | 43.9 | 23.7 | 43.1 | 23.0 | 42.4 | 24.5 | 43.8 | 26.0 | 48.4 | 21.5 | 36.4 | 22.1 | 37.2 |
| | 720 | 23.2 | 39.8 | 25.9 | 44.8 | 25.6 | 45.4 | 25 | 43.4 | 24.2 | 42.8 | 25.7 | 44.2 | 27.6 | 28.8 | 22.4 | 37.0 | 22.2 | 36.8 |
| | Avg. | 21.9 | 39.4 | 24.4 | 44.8 | 24.2 | 44.0 | 23.8 | 42.9 | 22.9 | 43.1 | 24.3 | 43.8 | 25.7 | 46.3 | 21.2 | 36.4 | 21.2 | 37.0 |
| 50% | 96 | 21.7 | 41.8 | 24.3 | 47 | 23.7 | 46.4 | 23.2 | 45.3 | 22.8 | 44.9 | 23.9 | 46.3 | 25.9 | 51.2 | 21.0 | 38.6 | 20.8 | 38.8 |
| | 192 | 22.6 | 42 | 25.4 | 47.2 | 24.8 | 46.6 | 24.4 | 45.6 | 23.9 | 45.0 | 24.9 | 46.5 | 27.4 | 51.4 | 21.9 | 38.8 | 21.8 | 38.9 |
| | 336 | 23.2 | 42.4 | 26.2 | 47.7 | 25.6 | 47.1 | 25.3 | 46.2 | 24.6 | 45.4 | 25.7 | 47.0 | 28.4 | 52.0 | 22.5 | 39.1 | 22.4 | 39.3 |
| | 720 | 24.9 | 43.3 | 27.9 | 48.8 | 27.3 | 48.2 | 27.2 | 47.2 | 26.3 | 47.2 | 27.4 | 48.1 | 30.6 | 53.2 | 24.0 | 40.2 | 23.9 | 40.2 |
| | Avg. | 23.1 | 42.4 | 26.0 | 47.7 | 25.4 | 47.1 | 25.0 | 46.1 | 24.4 | 45.6 | 25.5 | 47.0 | 28.1 | 52.0 | 22.4 | 39.1 | 22.2 | 39.3 |
| Average | | 22.1 | 40.0 | 24.4 | 45.2 | 24.1 | 44.7 | 23.6 | 43.9 | 23.0 | 43.5 | 24.3 | 44.6 | 26.1 | 48.4 | 21.2 | 37.2 | 21.1 | 37.2 |
| variations | | - | - | 9.50%↓ | 13%↓ | 9.0%↓ | 11.8%↓ | 6.8%↓ | 9.8%↓ | 4.1%↓ | 8.8%↓ | 10.0%↓ | 11.5%↓ | 18.1%↓ | 21.0%↓ | 4.2%↑ | 7.5%↑ | 4.7%↑ | 7.5%↑ |
| Comm. Param. # | | 0.38 M | | 0.38 M | | 0.38 M | | 0.38 M | | 0.38 M | | 0.38 M | | 10.00 M | | 41.99 M | | 10.00 M | |

## E.4 Hyper-parameter Sensitivity

The impacts of rank on performance are detailed in **Table. 54**. As the rank goes up, there's a consistent improvement, reaching its best at $r = 8$. However, when $r = 12$, there's a drop in performance. This happens because a higher rank means the local model has more trainable parameters, which can improve performance empirically. While a higher rank can cause increased communication cost and introduce more uncertainty.

## E.5 Pre-trained Language Model Variants

We compare three representative PLM backbones with varying capacities, the result is shown in **Table. 55**. Under the proposed LM-WEATHER framework, it's evident that various PLM backbones maintain strong sequence modeling capabilities. Moreover, the lightweight personalized adapter in LM-WEATHER enhance the PLM's

Table 54: Results on parameter impact study, where **Length** refers to the length of weather sequences (that is, predicted horizons in forecasting and input sequence length in imputation). **Avg.** represents the average value of predicted horizons, encompassing $\{96, 192, 336, 720\}$.

| Rank | Length | Forecasting | | Imputation | | Param.# | |
| --- | --- | --- | --- | --- | --- | --- | --- |
| | | MAE | RMSE | MAE | RMSE | Trainable | Comm. |
| 2 | Avg. | 47.6 | 79.8 | 25.1 | 46.6 | **10.14 M** | **0.12 M** |
| 4 | Avg. | 46.5 | 78.4 | 24.2 | 45.1 | 10.25 M | 0.24 M |
| 6 | Avg. | 46.5 | 76.9 | 24.0 | 44.6 | 10.37 M | 0.35 M |
| 12 | Avg. | 45.9 | 76.1 | 23.4 | 43.7 | 10.72 M | 0.70 M |
| **8 (Ori.)** | Avg. | **45.4** | **74.6** | **23.1** | **42.4** | 10.49 M | 0.47 M |

Table 55: Performance statistics for the proposed LM-WEATHER with various PLM backbones are presented, recording only the average performance across all lengths for different datasets (namely, 96/192/336/720 prediction horizons). For the imputation task, results are documented solely for a random masking probability of 50%. **Bold**: the best, Underline: the second best.

| Variant | Dataset | Forecasting | Imputation (50%) |
| --- | --- | --- | --- |
| LM-WEATHER (GPT2, **Original**) | **ODW1T** | **45.4/74.6** | **23.1/42.4** |
| | **ODW1V** | **45.6/71.9** | **43.7/63.8** |
| | **ODW2T** | **66.9/90.1** | **38.8/61.7** |
| | **ODW2V** | **69.0/92.3** | **31.1/47.0** |
| LM-WEATHER (Bert, 5) | **ODW1T** | 49.3/82.9 | 25.0/47.2 |
| | **ODW1V** | 49.9/80.0 | 48.3/71.2 |
| | **ODW2T** | 73.6/100.4 | 42.4/68.7 |
| | **ODW2V** | 76.2/102.9 | 34.5/52.6 |
| LM-WEATHER (Llama, 4) | **ODW1T** | 47.2/77.6 | 24.0/44.5 |
| | **ODW1V** | 47.2/74.2 | 45.7/67.2 |
| | **ODW2T** | 69.5/94.4 | 40.5/65.0 |
| | **ODW2V** | 72.3/97.6 | 32.4/49.8 |

ability to transfer knowledge from natural language sequences to complex weather sequences. This further validates the superiority and versatility of our LM-WEATHER.

## Appendix F    Additional Statements

### F.1    Impact Statements

We highlight that the goal of this study to proposed LM-WEATHER is not to compete but instead to complement current on-device meteorological variable modeling framework. Today's climate foundation models are typically trained from scratch, utilizing exceptionally large datasets (nearly 100TB) and incurring substantial computational costs. We hope that LM-WEATHER offers a cost-effective alternative for modeling meteorological variables on-device, thereby enabling accurate regional weather trend analysis. In addition, the dataset we complied can be the important resource to provide exploring chances for this field, facilitating future research.

This research seeks to make on-device meteorological variable modeling more efficient and adaptable. By using a PLM as a foundation model instead of training large foundation models from scratch, it eliminates the need for large-scale real weather data and extensive computational resources. Additionally, it supports a variety of devices, enabling everything from advanced smartphones to basic IoT sensors to perform meteorological variable modeling. The method is also designed to be stable in environments with limited data and those outside of typical distribution ranges, providing credible analytical support for further weather trend analyses.

## F.2  Limitations

Although our LM-WEATHER significantly outperforms models trained from scratch for time series analysis across various tasks and scenarios with minimal parameter tweaks, it still faces two primary limitations:

- **Limited Dataset Scale**: Due to constraints on computational resources and operational costs, we evaluated the performance of LM-WEATHER using the real-world datasets that did not approach the scale of tens of terabytes often required for training large-scale meteorological models. This limitation does not affect LM-WEATHER to be extended as a general framework for regional weather trend analysis. This framework supports the analysis of on-device meteorological variables and can be further developed and adapted for additional applications.

- **Dependence on PLMs' Quality and Performance**: Although LM-WEATHER leverages PLMs to achieve high efficiency and customization on heterogeneous devices, this dependency means that the quality and the performance of LM-WEATHER are intrinsically tied to the underlying PLMs. Should there be inherent limitations or biases within the PLMs, these could translate to the meteorological modeling performance. Conversely, if conditions allow the use of a more powerful LLM, LM-WEATHER's performance can be significantly improved. This might give the community more opportunities to explore the future road-map.

## F.3  Future Works

In future work, we aim to broaden the use of LM-WEATHER across more on-device variable modeling applications. We also plan to incorporate additional types of data, including satellite and radar imagery, as well as textual weather descriptions, to advance towards a more generalized approach to on-device meteorological variable modeling.

