# OpenReview forum: "Personalized Adapter for Large Meteorology Model on Devices: Towards Weather Foundation Models"
_NeurIPS.cc/2024/Conference — NeurIPS 2024 poster_

### Official Review · Reviewer_C54F · 2024-07-02

**Soundness:** 3
**Presentation:** 2
**Contribution:** 3
**Rating:** 7
**Confidence:** 3

**Summary:**

This paper proposes an approach called LM-WEATHER that utilizes pre-trained language models (PLMs) as foundation models for on-device modeling of heterogeneous meteorological variables. LM-WEATHER enhances PLMs-equipped devices with local weather knowledge through a lightweight personalized adapter. Additionally, it leverages low-rank based transmission to fuse global knowledge among devices, enabling high-efficiency communication.This approach provides an effective solution for modeling real-world, heterogeneous, and continuously updated weather data, without requiring high resource demands. I believe this is a meaningful contribution to the field.

**Strengths:**

This paper extensively compares various time series models across different meteorological variables and weather station.

The appendix section of the paper is comprehensive.

**Weaknesses:**

My first concern is that since the personalized adapter is already being used to adapt to the modeling of heterogeneous weather data collected from different weather station devices, what is the point of using the averaging operations during inter-device communication? This requires further explanation.

The definitions and formulations of symbols and equations throughout the sections, from Section 3.1 Local Training to Section 4 Theorems, are disorganized and complex. Additionally, there are numerous instances of incorrect singular and plural word usage. It is recommended to rewrite these sections with increased clarity and accuracy, simplifying the content for better comprehension.

**Questions:**

This paper uses GPT-2 as the PLM, which is a very basic pre-trained language model. I am curious why more advanced PLMs like GPT-4 [1], Llama [2], and Vicuna [3] were not considered for better sequence modeling performance.

Reference:
[1] Achiam J, Adler S, Agarwal S, et al. Gpt-4 technical report[J]. arXiv preprint arXiv:2303.08774, 2023.
[2] Touvron H, Lavril T, Izacard G, et al. Llama: Open and efficient foundation language models[J]. arXiv preprint arXiv:2302.13971, 2023.
[3] Chiang W L, Li Z, Lin Z, et al. Vicuna: An open-source chatbot impressing gpt-4 with 90%* chatgpt quality, March 2023.

**Limitations:**

Yes

---

> ### Author Rebuttal · Authors · 2024-08-04
>
> Thank you for providing us with your valuable feedback. We have carefully considered your questions and would like to address them as below:
>
> ---
> **Response to Weaknesses**
>
> **W1**. Further explain the significance of using the averaging operation.
>
> We appreciate your concern and would like to clarify the importance of the averaging operation. Averaging is the most important operation in FL that knowledge interaction between clients, it serves two key purposes in our framework:
> * **Knowledge sharing:** By averaging learnable generic knowledge across devices, we prevent data silos and enable each device to benefit from the collective insights of the collaboration network, while maintaining local personalization and data privacy when sensitive data are involved.
>
> * **Trade-off between personalization and generalization**: The averaging operation allows devices to learn broader patterns from others, while preserving local adapter personalization. This interplay forms a robust framework for adapting to local characteristics and leveraging global knowledge, striking a balance between personalization and generalization.
>
> Our experiments (**Table 1 below**) show that LM-Weather with averaging outperforms LM-Weather-Local without it, highlighting the importance of averaging operation. We hope these explanations and additional experiments can address your concerns.
>
> *Table 1. Results of LM-Weather and LM-Weather-Local (no averaging operation, i.e., no communication only updating the respective models locally at each client) on forecasting, $\downarrow$ is the average performance degradation relative to LM-Weather.*
> | ODW1T | LM-Weather | LM-Weather-Local | ODW1V | LM-Weather | LM-Weather-Local |
> |:--:|:--:|:--:|:--:|:--:|:--:|
> | 96    | 42.3   | 45.9     | 96    | 42.3       | 44.5   |
> | 192   | 44.4  | 46.7 | 192   | 44.4       | 46.1  |
> | 336   | 45.8  | 48.2  | 336   | 46.0       | 48.8  |
> | 720   | 49.2   | 50.5  | 720   | 49.7       | 53.2  |
> | Avg.  | 45.4   | 47.8  | Avg.  | 45.6       | 48.2  |
> | **Disparity** | -   | **$\downarrow$ 5.29%** | **Disparity** | -   | **$\downarrow$ 5.70%** |
>
> **W2**. Recommended to rewrite Section 3.1-4  with increased clarity and accuracy, simplifying the content for better comprehension.
>
> We appreciate your careful review and constructive suggestions. We will thoroughly revise these content to streamline the presentation of symbol and equations and simplify the content for better clarity and readability.
>
> ---
> **Response to Questions**
>
> **Q.  This paper uses GPT-2 as the PLM, which is a very basic pre-trained language model. I am curious why more advanced PLMs like GPT-4, Llama, and Vicuna were not considered for better sequence modeling performance.**
>
> We acknowledge the potential benefits of using more advanced PLM. However, Our primary goal is to establish a *general framework* for on-device weather sequence modeling via tuning PLMs. We chose GPT-2 for stability and cost-effectiveness, allowing for a basic yet effective demonstration of our approach. While more advanced PLMs may offer better performance, they require significant computational and storage resources, which can be a challenge for deployment on low-resource weather devices. To explore the LM-Weather performance with more advanced PLMs, we conducted additional evaluations using Llama-3B and Vicuna-7B, which show that more advanced PLMs can indeed improve performance **(Tables 2-5 below)**. Although GPT-4 was not included due to the unavailability of its weights (we are fine-tuning based on weights), our results demonstrate the potential benefits of using more advanced PLMs. Nevertheless, the increased computational demands of these models must be carefully considered for practical deployment. We hope these explanations and additional experiments can address your concerns.
>
> *The following results report the average forecasting (period [96, 192, 336, 720]) and imputation (50%) performance.*
>
> *Table 2. Performance of LM-Weather with different PLM backbone (ODW1T).*
> | PLM Backbone  | Forecasting  | Imputation   |
> |--|--|--|
> | GPT2 (default)| 45.4/74.6    | 23.1/42.4    |
> | LLaMA-3b (4 layers)| 47.2/77.6 | 24.0/44.5 |
> | LLaMA-3b (6 layers)  | 45.0/74.3    | 22.9/42.4    |
> | LLaMA-3b (8 layers)  | 44.0/73.5    | 22.4/42.5    |
> | Vicuna-7b (4 Layers)| 45.2/74.0| 23.0/41.9 |
> | Vicuna-7b (6 Layers) | 44.6/73.9    | 22.7/41.8    |
> | Vicuna-7b (8 Layers) | 43.3/72.8    | 22.0/41.1    |
>
> *Table 3. Performance of LM-Weather with different PLM backbone (ODW1V).*
> | PLM Backbone  | Forecasting  | Imputation   |
> |--|--|--|
> | GPT2 (default)  | 45.6/71.9    | 43.7/63.8    |
> | LLaMA-3b (4 layers)  | 47.2/74.2    | 45.7/67.2    |
> | LLaMA-3b (6 layers)  | 45.5/71.1    | 43.2/63.5    |
> | LLaMA-3b (8 layers)  | 44.4/70.6    | 42.1/62.0    |
> | Vicuna-7b (4 Layers) | 45.0/71.5    | 43.0/63.6    |
> | Vicuna-7b (6 Layers) | 44.3/70.0    | 41.9/61.4    |
> | Vicuna-7b (8 Layers) | 42.5/69.7    | 40.2/60.0    |
>
> *Table 4. Performance of LM-Weather with different PLM backbone (ODW2T).*
> | PLM Backbone  | Forecasting  | Imputation   |
> |--|--|--|
> | GPT2 (default) | 66.9/90.1    | 38.8/61.7    |
> | LLaMA-3b (4 layers) | 69.5/94.4  | 40.5/65.0   |
> | LLaMA-3b (6 layers)  | 67.0/89.2    | 38.2/61.3    |
> | LLaMA-3b (8 layers)  | 65.4/87.5    | 36.7/59.9    |
> | Vicuna-7b (4 Layers) | 66.1/89.2    | 36.5/60.6    |
> | Vicuna-7b (6 Layers) | 64.9/87.1    | 35.4/59.2    |
> | Vicuna-7b (8 Layers) | 62.9/85.0    | 33.6/58.1    |
>
> *Table 5. Performance of LM-Weather with different PLM backbone (ODW2V).*
> | PLM Backbone  | Forecasting  | Imputation   |
> |--|--|--|
> | GPT2 (default)  | 69.0/92.3    | 31.1/47.0    |
> |LLaMA-3b (4 layers)  | 72.3/97.6    | 32.4/49.8    |
> | LLaMA-3b (6 layers)  | 68.4/91.3    | 30.9/46.2    |
> | LLaMA-3b (8 layers)  | 65.4/89.5    | 29.6/45.1    |
> | Vicuna-7b (4 Layers) | 68.8/90.4    | 30.8/46.7    |
> | Vicuna-7b (6 Layers) | 65.6/87.4    | 29.4/45.3    |
> | Vicuna-7b (8 Layers) | 63.2/85.5    | 28.1/44.1    |

---

> > ### Comment · Reviewer_C54F · 2024-08-13
> >
> > I appreciate the authors' response, which addresses most of my concerns. I am happy to raise my rating.

---

> > > ### Author Response · Authors · 2024-08-14
> > >
> > > Dear Reviewer C54F,
> > >
> > > Thank you for raising the rating of our paper. We are happy to have addressed your concerns.
> > >
> > > Best regards,
> > >
> > > Authors

---

### Official Review · Reviewer_tT2F · 2024-07-05

**Soundness:** 4
**Presentation:** 3
**Contribution:** 3
**Rating:** 7
**Confidence:** 4

**Summary:**

This paper proposes LM-WEATHER, which builds upon previous work to explore further the powerful capabilities of PLMs in modelling meteorological variables. By learning sequence modelling capabilities from natural databases and applying them to on-device heterogeneity meteorological variable modelling, a lightweight adapter was developed to provide weather pattern awareness. Additionally, two real-world meteorological station datasets ODW1 & ODW2 support regional weather forecasting. Abundant experiments were designed to verify the feasibility of LM-WEATHER in modelling meteorological variables.

The article work is more valuable to further validate the superiority of PLMs in modelling meteorological variables and the provision of weather station data that can be used as a baseline dataset for weather station forecasting.

**Strengths:**

1.This work is more valuable to further validate the superiority of PLMs in modelling meteorological variables.
2.Two real-world meteorological station datasets ODW1 & ODW2, which were collected and compiled from four real-world versatile datasets for on-device meteorological variable benchmark.
3. Exploring the spatio-temporal sequence modeling of weather pattern specificity with high distributional similarity to provide a viable solution paradigm for sparse and heterogeneous weather station data.
4. From the experimental results, this method can further enhance the possibility of maintaining confidence in its future development in the presence of unconstrained data and PLM models.

**Weaknesses:**

1.	Figure 1 does not intuitively reflect the contributions of the paper. The Fig.1F mentioned in line 68 seems to be missing; additionally, the explanation for Figure C should be Task Adapter Generation, which is hard to understand. It is recommended to keep the labels a, b, … in the figure consistent with A, B, … in the text.
2.	The paper discusses sequence modeling in the time dimension of stations. Can this method be validated for spatiotemporal modeling?
3.	The experiments in the paper mainly focus on the accuracy of forecasting time. Can further validation be done on the timeliness and stability of the forecasts?
4.	To better present the experimental data, some statistical charts can be drawn to visually demonstrate the advantages of LM-WEATHER in meteorological variable forecasting.

**Questions:**

1.	In the Zero-Shot Learning (Out of Distribution Modeling) experiments mentioned in line 282, the domain transfer experiments for regional forecasting on the OWD1 dataset showed that the transfer performance was not as good as the forecasting performance of GPT-4. Can you explain the reason for this?
2.	In Theorem 4.2, how is it demonstrated that Low-Rank Matrices ensure privacy? Are there any related research papers on this topic? Can you provide a detailed explanation?
3.	Currently, there are many sparse station forecasts. Can interpolation be performed based on sparse station data to further extend the model to spatial dimension forecasting and enhance its generalizability?
4.	LM-WEATHER seems to outperform current methods. Can it maintain stability in long-term forecasting, and what is the maximum forecasting time it can achieve?
5.	In Table 6, line 305, it appears that having more trainable parameters is better. If the comparison experiment groups had the same number of trainable parameters, what would the results be?

**Limitations:**

Regarding limitation 1, further validating the impact of increased data scale on model performance using large-scale ERA5 data would be beneficial. Utilizing ERA5 as pre-training data to model invariance in meteorological forecasting and learning fundamental meteorological variable patterns, followed by fine-tuning with real-world data, could be a highly valuable endeavor.

---

> ### Author Rebuttal · Authors · 2024-08-04
>
> We greatly appreciate your insightful comments. We've carefully considered and addressed your concerns as follows ***(additional results are in Global Rebuttal PDF file)***:
>
> ---
> **Response to Weaknesses**
>
> **W1. Some confusion about Figure 1.**
>
> We will revise Fig. 1 to clarify the explanation of panel C and unify labels to improve overall clarity and readability, and correct the typo referencing Fig. 1F to Fig. 1a.
>
> **W2. Effectiveness of the method in spatio-temporal modeling.**
>
> To further demonstrate the effectiveness of LM-Weather for spatio-temporal modeling, we conducted additional experiments with a variant, LM-Weather-ST, which aggregates site data for centralized modeling. We compared its performance to the original LM-Weather and classical spatio-temporal forecasting baselines (STGCN, ASTGCN, TCGN, and CLCRN). The results ***(Table 3, Global Author Rebuttal PDF file)*** show that LM-Weather-ST achieves optimal performance, while the original LM-Weather still significantly outperforms the baselines despite being trained in a decentralized manner. This suggests that LM-Weather has strong potential for spatio-temporal modeling tasks. Future work will explore incorporating server-side graphs to model relationships between devices, further enhancing LM-Weather's benefits for spatio-temporal modeling.
>
> **W3. Timeliness and stability of forecasts**
>
> * **Timeliness:** We provide inference times for various forecasting periods **(Table 4, Global Author Rebuttal PDF file)**. When combined with the communication time of 0.29 s, these results demonstrate that LM-Weather achieves efficient prediction times, ensuring timely delivery of forecasts.
> * **Stability:** All experiment have been conducted five times. We present the standard deviations of LM-Weather and the runner-up model ***(Table 5,  Global Author Rebuttal PDF file)***, which indicate that LM-Weather can maintain stable performance over multiple experiments. Furthermore, we assessed longer-term forecasting tasks (1080, 1440, 1800, and 2160 hours) with a fixed input length of 192 ***(Table 6, Global Author Rebuttal PDF file)***. LM-Weather not only achieves optimal performance but also exhibits relatively stable error variations compared to representative baselines, further validating its stability.
>
> **W4. It is recommended that statistical charts be added.**
>
> We will add visualizations to each of the core experiment results in the revision.
>
> ---
> **Response to Questions**
>
> **Q1.  Why region transfer performance on ODW1 is not as good as that of GPT4.**
>
> LM-Weather outperforms FL-GPT4TS in most zero-shot scenarios, but the specific case of OWD1T -> OWD2V highlights an interesting trade-off. The difference lies in the distinct local tuning and global communication strategies employed by each method. LM-Weather uses learnable low-rank parameters to balance generic sequence modeling with personalized regional insights, whereas FL-GPT4TS unfreezes LayerNorm layers for more direct adaptation of the PLM's knowledge. This allows FL-GPT4TS to update and communicate more parameters, potentially enabling greater knowledge sharing in modeling heterogeneous weather data, but at the cost of increased communication overhead. Additionally, significant non-iid between OWD1T and OWD2V contribute to this difference.
>
> **Q2 How the low rank matrix of Theorem 4.2 ensures privacy.**
>
> We provide additional description and proofs for this, which can be found in ***G1, Global Author Rebuttal***.
>
> **Q3. Can interpolation based on sparse stations be used to extend the model to spatial dimension prediction and improve generalizability?**
>
> While spatial interpolation on sparse stations can increase the scale of training data, it is not an effective approach for extending the model's forecasting range and enhancing its generalizability due to:
>  * Spatial interpolation of site data does not ensure the accuracy of the interpolated values, particularly when dealing with very sparse datasets like ODW2.
>  * Even when interpolating between geographically proximate stations, the biases introduced by geographical differences cannot be fully compensated by interpolation alone, especially when multiple variables are involved. The complex interactions between variables and the influence of regional meteorological features render spatial interpolation ineffective for sparse station data.
>
> To further validate our statement, we conducted experiments on OWD1T and OWD2T using three spatial interpolation algorithms (Kriging, IDW, Spline). The results ***(Tables 7&8, Global Author Rebuttal PDF file)*** show that spatial interpolation performs poorly in sparse stations and fails to improve performance.
>
> **Q4. Stability and maximum forecasting time.**
>
> Regarding stability, **please refer to our response for W3**. As for the maximum forecasting time, while it is theoretically possible to extend the forecast horizon indefinitely by increasing computational resources, when the input data length remains fixed, longer forecast periods introduce greater uncertainty, leading to decreased performance. **As discussed in W3**, we observed a significant decline in performance when extending the forecast period from 720 to 2160 hours. This degradation highlights that, although technically feasible, very long forecasting times may not be practically useful due to the increased uncertainty and reduced accuracy.
>
> **Q5. Results when parameters are the same in the ablation experiment.**
>
> While it's true that increasing trainable parameters often boosts performance, LM-Weather-G/H's 4.5% gain comes at a steep cost: a five-fold increase in parameters. This trade-off is suboptimal. Our goal is to strike a balance between performance and efficiency. Notably, when parameter of the experiment group are similar, the original LM-Weather consistently outperforms its variants ***(Table 9, Global Author Rebuttal PDF File)***, demonstrating its optimal trade-off between performance and efficiency.

---

### Official Review · Reviewer_mBbn · 2024-07-13

**Soundness:** 4
**Presentation:** 3
**Contribution:** 4
**Rating:** 8
**Confidence:** 5

**Summary:**

The paper aims to develop weather foundation models by leveraging the on-device data on many distributed sensors. Specifically, a federated learning and low-rank adaption mechanism have been applied to a time-series-based foundation model training on Meteorology data.

**Strengths:**

1) The paper proposes a new way to develop weather foundation models by leveraging many distributed devices with collected Meteorology data. It is worth noting that weather foundation models are an emerging research domain critically important to geoscience and tackling climate change.

2) The proposed solution is technique sounds. The procedure of time series processing, LoRA-based adaptation, and federated learning are seamlessly integrated to implement its goal.

3) The paper provides sufficient details with an appendix and open-sourced codes. It is an essential part to ensure the reproducibility of this work.

4) The paper conducted a comprehensive experiment for comparison and evaluation of the few/zero-shot learning scenarios on the foundation model.

**Weaknesses:**

1) The paper needs a strong justification to describe the motivation for using a pre-trained language model as a basis for the weather foundation model. Are there any texts in the Meteorology dataset?

2) The paper’s contents organization could be improved. For example, the figure 1 is too complex to understand. In Eq 2, it would be better to write the function as F(theta, D). Because F(theta | D) is easily confusing with the conditional distribution. Moreover, some technique details should not be placed in the appendix. It is a usual assumption that the main paper is self-contained so that the readers can understand the paper without reading the appendix.

3) In the contribution part (lines 73 - 94), the last two items look like the advantages of the proposed method rather than contributions.

**Questions:**

1) Does the proposed method only focus on time series data?

2) Would you please highlight the main research question in plain language? It seems there are many components mixed in the paper.

3) In Section 3, what’s the federated aggregation mechanism? Is there any pseudo code to describe the algorithm?

4) Can you please explain the token embedding for time series data? Is the token to be described as a special pattern of a subsequence of the time series? How to define the vocabulary of the tokens?

---

> ### Author Rebuttal · Authors · 2024-08-04
>
> Thank you for providing us with your valuable feedback. We have carefully considered your questions and would like to address them as below:
>
> -------
> **Response to Weaknesses**
>
> **W1. Justification for using PLM and clarity on weather dataset.**
>
> Although our dataset consists solely of time series data without any textual content, using a PLM as the basis of the weather foundation model offers substantial flexibility and an optimal trade-off between performance and costs. Our motivations can be summarized by:
> * *Limitations of Existing Climate Foundation Models*: Current climate foundation models, such as ClimaX [1] (73B parameters) and Pangu-Weather [2] (up to 100B parameters), demand vast computational resources and extensive reanalysis data volumes (over 100 TB) for pre-training, making them unsuitable for deployment on low-resource meteorological equipment due to prohibitive operational costs and computational demands.
> * *Sequence Modeling Capabilities of PLMs*: Despite being initially trained on text, their advanced sequence modeling capabilities, proven on time series analysis [3]. This eliminates the need for training models from scratch with high-quality weather observational data, which is often scarce and expensive to acquire as opposed to reanalysis data. Consequently, adopting a PLM not only employs its powerful sequence modeling abilities but also drastically cuts down on computational expenses and deployment costs when fine-tuning for weather sequence modeling.
>
> **W2. The paper's organization and clarity can be improved, e.g., notation, figure, and placement of technical details.**
>
> Thank you for your careful review and constructive comments. In our revision, we will recheck all notations in the paper to avoid confusion. Additionally, we will adjust the layout of the main text to incorporate important content from the appendix, enhancing readability. Your feedback will significantly improve the quality of our paper!
>
> **W3. The last two items in contribution part appear to be advantages rather than contributions.**
>
> We will adjust the contribution part to improve clarity of expression.
>
> --------
> **Response to Questions**
>
> **Q1: Does the proposed method only focus on time series data?**
>
> Yes, LM-Weather currently specializes in processing weather time series. Utilizing a PLM can bring significant benefits in terms of flexibility and cost-effectiveness, **as detailed in our response for W1**. To further enhance the generality and applicability of LM-Weather, we plan to expand our dataset to include multi-modal data such as descriptive weather texts, satellite cloud imagery, and radar data, ultimately developing a comprehensive multi-modal weather sequence analysis framework.
>
> **Q2. Highlight the main research question.**
>
> The main research question can be summarized as: **How can we adapt pre-trained language models to efficiently and effectively model weather sequences on local devices in a decentralized manner, while addressing the challenges of data diversity and limited computing resources?** This question seeks to explore:
> * Adaptation: How can complex models originally designed for language processing be tailored to understand weather sequences?
> * Personalization: Can these models provide tailored weather predictions that suit the specific conditions and data available at individual locations and devices?
> * On-device efficiency: Can this approach operate effectively on local devices, ensuring privacy and minimizing resource use despite the constraints of limited computational power and storage?
>
> **Q3. Federated aggregation mechanism and pseudo code.**
>
> Our aggregation mechanism is based on FedAvg, where only low-rank parameters are shared and averaged, weighted by each client's sample size ratio. We provide the pseudo code below, which will be added to the revision for clarity.
>
> *Algorithm 1*. low-rank parameter-based aggregation mechanism (server-side, all clients participate in the training).
> |---|
> *Input*: Low-rank parameters from each client {$\theta_{l,1}, \theta_{l,2}, ..., \theta_{l,k}$}, number of samples per client {$n_1, n_2, ..., n_k$}
> *Output*: Aggregated low-rank parameters $\theta_{l}$
>
> ---
> 1. Receiving low-rank parameters from clients
> 1. Calculate the total number of samples across clients $N = \sum_{i=1}^k n_i$
> 2. Aggregating low-rank parameters: $\theta_{l}' = \sum_{i=1}^k \frac{n_i}{N} \theta_{l, i}$
> 3. Broadcasting updated $\theta_{l}'$ to clients to continue next communication round.
> ---
>
> **Q4. Explain the token embedding? Is the token to be described as a special pattern of a subsequence of the time series? How to define the vocabulary of the tokens?**
>
> The token embedding module aims to transform subsequences of weather time series into a higher-dimensional space using a convolutional layer. Each transformed subsequence is represented as a continuous vector, known as a `token`. This process allows the model to capture localized patterns within a specific receptive field. Each token encapsulates patterns from a subsequence of the time series, capturing local dynamics. The kernel size determines the subsequence length, combining each point with its immediate neighbors to form a token. These tokens capture important temporal dependencies. Unlike traditional NLP, the `vocabulary` here is formed by the diverse outputs of the convolutional layer, representing a range of patterns in the input data. This vocabulary is continuously defined, shaped by the convolutional filters that learn to identify and abstract temporal features during training.
>
> ---
> **Reference**:
> 1. ClimaX: A foundation model for weather and climate. arXiv 2023.
> 2. Accurate medium-range global weather forecasting with 3D neural networks. Nature 2023.
> 3. "Large language models for forecasting and anomaly detection: A systematic literature review. arXiv 2024.

---

> > ### Comment · Reviewer_mBbn · 2024-08-11
> >
> > Many thanks for your rebuttals. After carefully checking it, my questions have been answered and  I will keep my score.

---

> ### Author Response · Authors · 2024-08-11
>
> Thank you for your confirmation. We are pleased that our rebuttals have addressed your concerns and appreciate your continued support for our paper.
>
> Best regards,
>
> Authors

---

### Official Review · Reviewer_Jjnz · 2024-07-13

**Soundness:** 3
**Presentation:** 2
**Contribution:** 3
**Rating:** 6
**Confidence:** 4

**Summary:**

This paper introduces LM-WEATHER, a framework leveraging pre-trained language models (PLMs) for on-device meteorological variable modeling. The framework integrates personalized adapters into PLMs, enhancing their ability to handle heterogeneous weather data efficiently. Key contributions include superior performance in forecasting and imputation tasks, minimal communication overhead, and maintaining privacy during client-server interactions. Experiments on real-world datasets demonstrate LM-WEATHER's effectiveness across various scenarios, including few-shot and zero-shot learning.

**Strengths:**

S1. LM-WEATHER introduces a novel approach by integrating personalized adapters into PLMs, allowing for efficient and accurate on-device meteorological modeling.

S2. The framework addresses heterogeneity in weather data and demonstrates robust performance in various tasks and scenarios.

S3.The approach offers a promising solution for personalized weather modeling on resource-constrained devices, potentially benefiting various applications in meteorology and related fields.

**Weaknesses:**

W1.In the communication section, it is mentioned that only the low-rank matrix parameters are transmitted between the client and server, but specific details about the transmission mechanism and strategy are missing. This includes transmission frequency, bandwidth consumption, and potential issues in practical applications. These missing details may prevent readers from fully understanding the feasibility and effectiveness of the communication strategy.

W2. In the parameter adapter section, it describes integrating the generated adapters with the original input through FFN to produce the final weather sequence predictions. However, the specific architecture of the FFN, how its parameters are configured, and why this particular architecture was chosen are not detailed. The lack of these key details might leave readers questioning the mechanism and effectiveness of the parameter adapter.

W3. In Section 4, while Theorems 4.1 and 4.2 propose the rationality of time series decomposition and privacy assurance through low-rank matrix exchange. There is no introduction and description about the background and application of these Theorems.  Also, the descriptions are too brief, making it difficult for readers to fully understand the theoretical basis.

W4. The paper's experimental results are heavily supplemented by a 37-page appendix, which can hinder readability and accessibility of the core content. Important experimental details and results should be better integrated into the main body of the paper to facilitate easier comprehension and evaluation by the readers.

**Questions:**

Q1.Could you provide more detailed proofs and theoretical validation for the decomposition rationality and privacy assurances mentioned in the theorems?

Q2.Could you clarify the potential confusion in the notations n¬i and nk in Section 2.2, and ensure the formulas are correctly presented?

Q3.Could you provide detailed information on the architecture and parameter configuration of the FFN used in the parameter adapter, and explain why this architecture was chosen?

Q4.Could you provide more detailed information about the datasets, particularly whether data from extreme climatic regions(e.g., tropical rainforests and deserts) are included? If not, how do you plan to expand these datasets to cover more global climatic conditions?

**Limitations:**

1 The authors acknowledge the limitations in the diversity of the datasets and the potential challenges in modeling specific climatic conditions not covered in the current study. Future work should focus on expanding dataset coverage and providing more detailed theoretical and empirical validation. Additionally, the practical applications and limitations of few-shot and zero-shot learning capabilities should be more thoroughly explored and documented.

2 It would be much better integrate the essential experimental details and results into the main body of the paper to improve readability and comprehension?

3 I have questioned whether there will be a powerful foundation model for time series prediction. If the assumption is not true, the application of the proposed solution is limited.

---

> ### Author Rebuttal · Authors · 2024-08-05
>
> Thank you for providing us with your valuable feedback. We have carefully considered your questions and would like to address them as below ***(additional results are in Global Rebuttal PDF file)***:
>
> ---
> **Response to Weaknesses and Question (& means joint response)**
>
> **W1. Details about the transmission strategy.**
>
> Our framework employs a synchronous federated learning strategy, where the client and server communicate at a fixed frequency (every 20 iterations). The process of transmission includes:
> * 1. client-side updating: each participant uploads low-rank parameters of their local model.
> * 2. server-side aggregation: the server aggregates uploaded parameters via federated averaging.
> * 3. low-rank parameter broadcasting: the server broadcasts aggregated parameters to each participant for the next round.
>
> By communicating only low-rank parameters (0.38M parameters), we minimize bandwidth consumption. We acknowledge potential issues in practical applications, including communication overhead, latency, and data privacy. Our method addresses these concerns by transmitting only low-rank parameters, thereby safeguarding privacy and minimizing communication overhead. We will clarify this key aspect of the transmission strategy in the revision to enhance understanding.
>
> **W2 & Q3. Detailed information of the FFN in parameter adapter.**
>
> Our parameter adapter is a straightforward FFN with a single linear layer, converting the PLM's output *(input channels, e.g., 768 for GPT2, 4096 for Llama-3B)* to align prediction horizons of [96, 192, 336, 720] *(output channels)*. We chose this configuration to optimize the balance between performance and computational efficiency. Our additional experiments ***(Table 1, Global Author Rebuttal PDF file)*** show that adding more layers only marginally improves performance (0.21% with two or three layers) and even decreases with four layers, while significantly increasing computational demands. Therefore, the single-layer setup is the most cost-effective, providing an optimal trade-off between efficiency and performance.
>
> **W3 & Q1. Introduction and description of backgournd, and more detailed proofs and theoretical validation for theorems.**
>
> We provide additional description and proofs for theorems, which can be found in **G1, Global Author Rebuttal**.
>
> **W4. Important details and results should be better integrated into the main body.**
>
> Thank you for your constructive feedback. We will revise the layout to highlight the core results in the main body.
>
> **Q2. Potential confusion in the notations $n_i$ and $n_k$.**
>
> We apologize for the confusion caused by our typo. The correct Eq. 2 is: $n_i$, $F(\theta) := argmin \sum_{i=1}^{N} \frac{n_i}{n} F_i(\theta_i; \lbrace D_i \rbrace)$, where $n_i$ represents the $i$-th client. We will carefully revise and clarify these notations to enhance the readability.
>
> **Q4. Does the dataset have detailed information on extreme climate regions, and if not, what are the plans for expanding the dataset?**
>
> Including more details about specific extreme climatic regions indeed benefits the expansion of our dataset's potential application scenarios. Current version comprises real observations from various cities, with limited access to detailed topographic or regional information. Recognizing the potential benefits of expanding this coverage, we are actively working to include a broader range of climatic conditions. Specifically, we plan to introduce multimodal weather data, such as radar echoes and satellite cloud images, into our dataset's corresponding regions. Additionally, by collaborating with climate experts, we will provide textual descriptions of weather trends to create a comprehensive multimodal dataset.
>
> -----------
> **Response to Limitations**
>
> **L1: Future work: expanding the dataset, deepening theoretical and empirical validation, and exploring the applications and limitations of few/shot-learning.**
>
> Thank you for your insightful comments, which have provided a clear direction for our future work. **As the response for Q4**, we plan to introduce additional data modalities to enhance the comprehensiveness and applicability of current datasets. In future work, we will conduct more detailed theoretical and empirical analyses based on the newly constructed multimodal dataset. Additionally, we will further explore the limitations of these observations in practical applications and few/zero-shot scenarios.
>
> **L2: It would be much better integrate the essential experimental details and results into the main body.**
>
> We will integrate essential details and results into the main body in the revision to enhance readability and adjust the layout accordingly.
>
> **L3: Whether there will be a powerful foundation model for time series prediction. If the assumption is not true, the application of the proposed solution is limited.**
>
> Our approach is indeed based on the assumption that there is already a good foundation model. We believe that this assumption is reasonable because there are already many well-established foundation models that can be used in time series forecasting tasks, e.g., MOMENT[1], Timer[2]. However, we also understand the your concern that without a powerful foundation model, our method may suffer. To address this concern, we used several PLMs that weak than GPT-2 (e.g., Bert [3], CTRL [4]) to evaluate the performance of LM-Weather, and the results ***(Table 2, Global Author Rebuttal PDF File)*** show that LM-Weather can achieve good performance when the assumption of a powerful foundation model is not hold, and significantly outperforms other regular model.
>
> ---
> Reference:
> 1. Moment: A family of open time-series foundation models. arXiv 2024.
> 2. Timer: Transformers for time series analysis at scale. arXiv 2024.
> 3. Bert: Pre-training of deep bidirectional transformers for language understanding. arXiv 2018.
> 4. Ctrl: A conditional transformer language model for controllable generation. arXiv 2019.

---

> > ### Comment · Reviewer_Jjnz · 2024-08-13
> >
> > Thanks a lot for your response and revisions. I have raised my score to reflect such improvement.

---

> > > ### Author Response · Authors · 2024-08-13
> > >
> > > Dear Reviewer Jjnz,
> > >
> > > Thank you for upgrading your score. We appreciate your constructive suggestions for improving our paper.
> > >
> > > Best,
> > >
> > > Authors

---

### Author Rebuttal · Authors · 2024-08-05

We thank the reviewers for their valuable feedback. ***Additional experiments are included in the attached PDF file***, indexed as follows:
* *Reviewer Jjnz:* Table 1, Table 2
* *Reviewer mBbn:* No additional experiments supplements.
* *Reviewer tT2F:* Table 3 - Table 9
* *Reviewer C54F:* Additional experiments are provided in the box corresponding to Reviewer C54F.

Below are our general replies to the general concerns from the reviewers:

**G1. *(Reviewer Jjnz, Reviewer tT2F)* Detailed introduction, description, and proof for Therorems.**

* **Introduction and Background for Therorem 4.1:** Decomposing time series into trend and seasonal components helps model underlying patterns. This theorem states that non-orthogonal trend and seasonal components cannot be fully separated by orthogonal bases. Since self-attention layers learn orthogonal transformations similar to PCA, applying attention to raw time series is ineffective at separating non-orthogonal components, highlighting the need for manual decomposition.

* **Introduction and Background for Theorem 4.2:** The theorem is built upon federated learning, where an external attacker can exploit client parameters to compromise sensitive information. We demonstrate that LM-Weather ensures privacy by limiting uploaded parameters to a small, low-rank subset, making it challenging for an attacker to reconstruct client data. This guarantees the protection of sensitive information.

**Additional Proof for Theorem 4.1 - Self-attention Layer is a Nonlinear PCA:** The Self-attention can be viewed as a non-linear PCA, with the computation process represented as: $Attention(Q, K, V) = softmax(QK^T / \sqrt{d}) V$.  This can be decomposed into: $Attention(Q, K, V) = \sum_i \alpha_i v_i$, where $\alpha_i$ is the weight coefficient, and $v_i$ is the $i^{th}$ component of the Value. The weight coefficient $\alpha_i$ can be computed as: $\alpha_i = \frac{\exp\left(\frac{QK_i^T}{\sqrt{d}}\right)}{\sum_j \exp\left(\frac{QK_j^T}{\sqrt{d}}\right)}$. This is similar to the SVD of PCA: $X = U\Sigma V^T = \sum_i \sigma_i u_i v_i^T$. where $X$ is the input data, $U$ is the left singular vector, $\Sigma$ is the singular value matrix, $V$ is the right singular vector, $\sigma_i$ is the singular value, and $u_i$ and $v_i$ are the $i^{th}$ components of the left and right singular vectors, respectively. Comparing the two formulas reveals the similarity between Self-Attention and PCA. Combined with Theorem 1 in [1], this suggests that self-attention layer in PLMs functions closely to PCA (only learn orthogonal transformation) and can't automatically decompose time series into trend and seasonal components (non-orthogonal) without manual intervention.

**Additional Proof for Theorem 4.2 - Model Indistinguishability:** Consider two clients with different local models $\mathcal{M}_i$ and $\mathcal{M}_j$ parameterized by $\theta_i$ and $\theta_j$. Let $L(\mathcal{M})$ denote the low-rank matrix from model $\mathcal{M}$, where $B \in \mathbb{R}^{d\times r}$ and $A \in \mathbb{R}^{r\times d}$, with $r\ll d$. Then, for any polynomial-time attacker $\texttt{Adv}$, the following holds: $|Pr[\texttt{Adv}(L(\mathcal{M}_i)) = 1] - Pr[\texttt{Adv}(L(\mathcal{M}_j)) = 1]| \leq \epsilon$, where $\epsilon$ is a small positive number. This implies that the attacker cannot distinguish between client models from a shared low-rank update.

*Proof*. LoRA aims to find the low-rank approximation: $ min ||\theta - \theta_0 - BA||_F$, where $\theta_0$ is the initial weight. Consider two models $\mathcal{M}_i$ and $\mathcal{M}_j$ with weights differing by $\Delta\theta = \theta_i - \theta_j$ from different clients. The corresponding LoRA matrix is:

$$L_i = B_iA_i \approx \Delta \theta_i = \theta_i - \theta_0,  L_j = B_jA_j \approx \Delta \theta_j = \theta_j - \theta_0.$$
According to matrix approximation theory, for the best approximation with rank $r$, the upper bound on the error is:

$||\Delta \theta_i - L_i||_ F  \leq \sigma_{r+1}(\Delta \theta_i)$

where $\sigma_{r+1}(\Delta \theta_i)$ is the $(r+1)$-st singular value of $\Delta \theta_i$. By Johnson-Lindenstrauss Lemma [2], for any $\epsilon > 0$, there exists a mapping $f: \mathbb{R}^d \rightarrow \mathbb{R}^k$ with $k = O(log(n)/ \epsilon^2)$, such that for any $x,y \in \mathbb{R}^d$. LoRA can be regarded as such a mapping. Assuming $||\Delta \theta_i - \Delta \theta_j||_F \leq \delta$ and using the trigonometric inequality:

$$||L_i - L_j||_F \leq ||L_i - \Delta \theta_i||_F + ||\Delta \theta_i - \Delta \theta_j||_F + ||\Delta \theta_j - L_j||_F$$

$$ \leq \sigma_{r+1}(\Delta \theta_i) + \delta + \sigma_{r+1}(\Delta \theta_j).$$

Let $\varepsilon' = \sigma_{r+1}(\Delta \theta_i) + \sigma_{r+1}(\Delta \theta_j)$, we get:

$$||L_i - L_j||_F \leq \delta + \varepsilon'$$

For any polynomial-time attacker $\texttt{Adv}$, its ability to distinguish between $L_i$ and $L_j$ is restricted to the difference in their Frobenius paradigms. We can define a function $f$ such that:

$$\left|Pr[\texttt{Adv}(L(\mathcal{M}_i)) = 1] - Pr[\texttt{Adv}(L(\mathcal{M}_j)) = 1] \right| \leq f(||L_i - L_j||_F),$$

where $f$ is a monotonically increasing function representing the attacker's capability. Finally, we get: $|Pr[\texttt{Adv}(L(\mathcal{M}_i)) = 1] - Pr[\texttt{Adv}(L(\mathcal{M}_j)) = 1]| \leq \epsilon$. When $\delta$ and $\varepsilon'$ are small enough, the right-hand side is smaller than the intended $\varepsilon$, ensuring that an attacker cannot reverse-engineer local parameters and data, thus preserving privacy through low-rank parameters-only communication.

*We hope that the above additions will address the reviewers' concerns. We will add these additional introductions and proofs to the theorems in our paper during the revision to improve readability.*

---
Reference:
1. One fits all: Power general time series analysis by pretrained lm. NeruIPS 2023.
2. On variants of the Johnson–Lindenstrauss lemma. Random Structures & Algorithms 2008.

---

> ### Comment · Reviewer_tT2F · 2024-08-11
>
> Thank you for your response. This approach offers an efficient solution for modeling heterogeneous and continuously updated real-world weather data, while minimizing resource demands. I believe this is a significant contribution to the field.

---

> > ### Author Response · Authors · 2024-08-11
> >
> > Dear Reviewer tT2F,
> >
> > We appreciate your recognition of our approach's efficiency and significance. Your thoughtful comments also provide us with valuable new insights, for which we are grateful.
> >
> > Best,
> >
> > Authors

---

### Decision · Program_Chairs · 2024-09-25

**Decision:**

Accept (poster)

**Comment:**

This paper proposes LM-Weather, which uses pre-trained LMs (PLMs) as foundation models for on-device meteorological variable modeling. LM-Weather integrates personalized adapters into PLMs, enhancing the ability to handle heterogenous weather data. Experiments are conducted on multiple real-world datasets showcasing the effectiveness of LM-Weather, outperforming SoTA results on forecasting and imputation tasks, and having low computation overhead.

Strengths:
Proposed LM-Weather is a novel approach that integrates personalized adapters into PLMs, enabling accurate on-device meteorological modeling. Sound experimentation on real-world data, results showcase robust performance of LM-Weather across various tasks & scenarios, as well as improvements over SoTA. Comparison of various time series models across different meteorological variables and weather stations.


Weaknesses:
Prior to the final submission, authors should address the feedback and questions of the reviewers, as well as clarify:
(1)  transmission mechanisms and strategy of the low-rank matrix
(2)  motivation behind using FFN architecture, details of the parameter configuration
(3) expanding on the theoretical aspect
(4) synthesizing the most important results and including them in the main part of the paper to help readers get the main contribution of the work right away